# Shared Lexical Task Representations Explain Behavioral Variability In LLMs

Zhuonan Yang [* 1]   Jacob Xiaochen Li [* 2]   Francisco Piedrahita Velez [* 1]
Eric Todd [3]   David Bau [3]   Michael L. Littman [1]   Stephen H. Bach [1]   Ellie Pavlick [1]

## Abstract

One of the most common complaints about large language models (LLMs) is their prompt sensitivity—that is, the fact that their ability to perform a task or provide a correct answer to a question can depend unpredictably on the way the question is posed. We investigate this variation by comparing two very different but commonly-used styles of prompting: *instruction-based* prompts, which describe the task in natural language, and *example-based* prompts, which provide in-context few-shot demonstration pairs to illustrate the task. We find that, despite large variation in performance as a function of the prompt, the model engages some common underlying mechanisms across different prompts of a task. Specifically, we identify task-specific attention heads whose outputs literally describe the task — which we dub *lexical task heads*—and show that these heads are shared across prompting styles and trigger subsequent answer production. We further find that behavioral variation between prompts can be explained by the degree to which these heads are activated, and that failures are at least sometimes due to competing task representations that dilute the signal of the target task. Our results together present an increasingly clear picture of how LLMs' internal representations can explain behavior that otherwise seems idiosyncratic to users and developers.

## 1. Introduction

Large Language Models (LLMs) demonstrate remarkable capabilities across complex tasks, yet their performance remains frustratingly sensitive to prompt phrasing (Errica

et al., 2025; Shafiei et al., 2025; Cao et al., 2026). This behavioral variance poses significant reliability and usability concerns, as minor linguistic shifts can lead to unpredictable outcomes or catastrophic failures. Understanding the internal mechanisms driving this variance is thus critical for building robust and predictable AI systems, as well as for improving our basic understanding of how and why LLMs behave the way they do.

We focus on two widely used prompting styles. *Instruction-based* prompts provide natural language descriptions of the task to the model, e.g., "Translate the following word into German: cat". In contrast, *example-based* prompts illustrate the task by showing desired input–output pairs, e.g., "hello → hallo; please → bitte; dog → Hund; cat →" (Brown et al., 2020). While in principle these prompts carry information about the same underlying task, in practice we can see wide variation in the model's accuracy at performing the task as a function of the particular prompt used (Fig. 1).

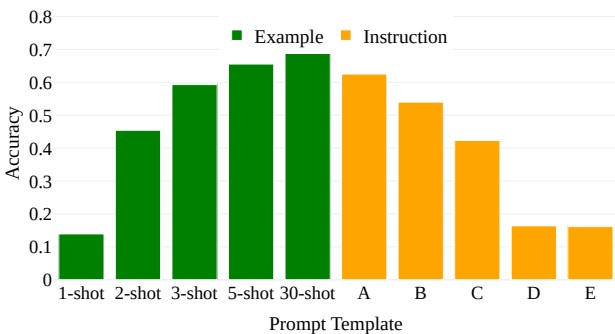

*Figure 1.* A representative result showing that outputs of Llama-3.1-8B vary in accuracy based on both prompt styles (example-based vs. instruction-based prompts) and prompt templates (number of examples, wording, etc.). A–E refer to specific wordings of instructions (see §A.3 for prompting details). Although the prompts all query the antonyms of a same set of words, accuracy can differ by a factor of 4. See Appendix B for results across different tasks and models.

In this work, we seek to explain this variation by investigating the underlying mechanisms invoked for each type of prompt. Previous literature has characterized mechanisms that are not shared between prompting styles (Todd et al., 2024; Hendel et al., 2023; Davidson et al., 2025), which could explain the behavior variations. Here, we investigate

---

[*]Equal contribution  [1]Brown University  [2]MIT  [3]Northeastern University.   Correspondence to:   Zhuonan Yang <zhuonan_yang@brown.edu>, Jacob Xiaochen Li <jacobli@mit.edu>, Francisco Piedrahita Velez <fpiedrah@brown.edu>.

*Proceedings of the $43^{rd}$ International Conference on Machine Learning*, Seoul, South Korea. PMLR 306, 2026. Copyright 2026 by the author(s).

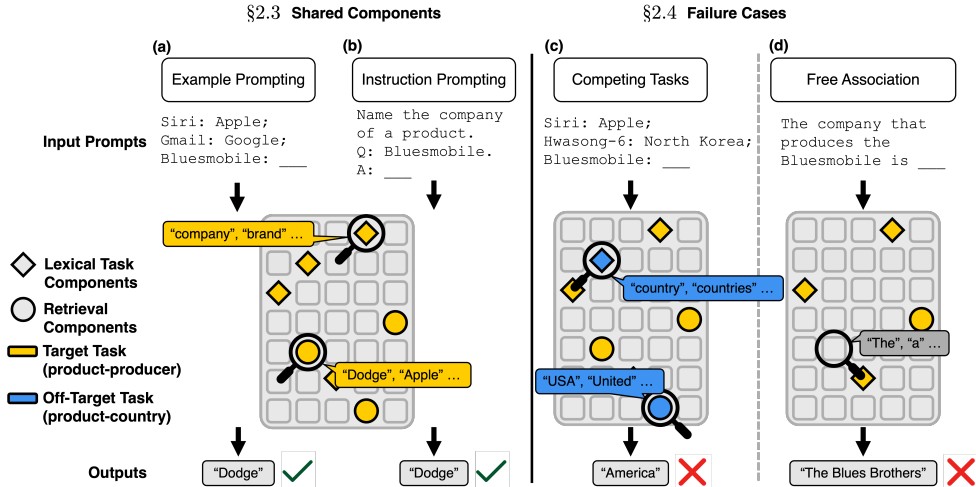

*Figure 2.* An overview of Section 2. In §2.2, we describe the key model components studied in this work, *lexical task components* (shown as diamonds) and *retrieval components* (shown as circles). In §2.3, we focus our study on the cases where the model gets *correct* results and find that these key components are shared across prompting styles. (a) example-based prompts and (b) instruction-based prompts for the same target task (e.g., Product–Producer) rely on shared lexical and retrieval components (shown in yellow) to produce the correct answer. In §2.4, we compare the *success and failure* cases and show how the shared components are activated differently, leading to incorrect outputs. For the correct prompts, lexical task representations are robust, guiding retrieval components to promote the correct answer. Conversely, the incorrect prompts exhibit weak activation of the target task and either (c) trigger mechanisms for competing tasks (e.g., Product–Country, shown in blue) when ambiguous examples are presented or (d) are not strong enough to overcome baseline behaviors (e.g., free association of "The Blues Brothers" movie to the "Bluesmobile" car).

the mechanisms that *are shared*, with variation explained by the strength of their activations. Specifically, we focus on mechanisms by which LLMs represent and execute tasks. Prior work in this area (Bakalova et al., 2025; Kharlapenko et al., 2025; Geva et al., 2023) use a set of simple tasks that extract a single attribute (e.g., capital city) of a given subject (e.g., Japan). In these tasks, previous work proposed that the model forms a **task representation** (some internal encoding of the Country–Capital task). The model then uses the task representation to **execute** the task (retrieve the output "Tokyo" for the input "Japan").

One kind of task representation that has been studied is function vectors (Todd et al., 2024; Hendel et al., 2023), which are representations that do not directly perform, but trigger the execution of, a task. Function vectors are produced by a set of attention heads when the model performs in-context-learning (ICL) tasks with input–output examples. Davidson et al. (2025) later found that the function vectors extracted from instruction-based prompts leverage a mostly distinct set of attention heads for task representation.

In this work, we identify a different task representation that is shared across prompting styles. We refer to it as a *lexical task representation* because it aligns with the literal description or definition of a task. To identify attention heads that produce lexical task representations, we project attention heads' outputs to the vocabulary space (Nostalgebraist, 2020; Dar et al., 2023). Doing so allows us to access the output of a given attention head and decode it

in words. The lexical task representations we study differ from token embedding representations (Dar et al., 2023), because the decoded words semantically match the meaning of the *task* itself rather than the final model outputs. For example, the decoded vocabularies of the *lexical task heads* for the Antonym task are "opposite", "reverse", etc., which represent the definition of the Antonym task, rather than providing the specific antonym of a queried word. Details of how we define and identify the attention heads that produce lexical task representations are provided in §2.2.

Most importantly, lexical task representations differ from function vectors (see Appendix I) in that they generalize across prompting styles, revealing a shared encoding that captures the core meaning of a task, rather than being specific to surface-level structure of a given prompting style. For the 17 tasks that we study, causal experiments on these lexical task heads account for at least 10% to as much as 90% of the task performance across prompts (see Appendix G.1).

For task execution, recent literature on factual recall (Geva et al., 2023; Elhelo & Geva, 2025; Chughtai et al., 2024) identified a set of attention heads that retrieve factual knowledge, which we call "retrieval heads". For example, to recall the capital city of Japan, retrieval heads extract attributes like "Tokyo", "Japanese", "kimonos", etc. of the subject "Japan". However, if and to what extent these heads are shared across prompts was not previously studied. Furthermore, how task-representation components modulate task-execution components was also not thoroughly explored.

In summary, we make the following contributions:

1. **Introduce lexical task representations**. We identify a distinct class of attention heads that encode a task in "lexical" format such that their outputs are interpretable after being projected to the vocabulary space (§2.2).

2. **Characterize shared mechanisms across prompting styles**. We show that lexical task heads and retrieval heads are shared across prompting styles (§2.3 & Appendix J.4).

3. **Explain behavioral variability with shared components**. We provide one mechanistic explanation of prompt sensitivity: semantically equivalent prompts trigger varying activation levels in lexical task heads, which subsequently influence how effectively retrieval heads extract the correct answer, resulting in varying performance across prompts (§2.4.1).

## 2. Experiments and Results

Our primary questions are: Are there shared internal mechanisms across prompting styles; and can these shared mechanisms explain behavioral variability in LLMs?

### 2.1. Experimental Setup

To investigate the internal mechanisms of LLMs, we use 13 autoregressive transformer language models across three model families, 17 simple tasks, three compositional tasks and one free-form code generation task, as well as diverse prompts across prompting styles (Appendix A). Results presented in the main paper are based on Llama-3.1-8B-Instruct (Grattafiori et al., 2024).

### 2.2. Identifying key components

#### 2.2.1. MOTIVATION

Our first exploratory analysis reveals some unique attention heads, where when their outputs are decoded into vocabulary space using the Logit Lens (Nostalgebraist, 2020; Dar et al., 2023), they appear to directly describe the task specified in the prompts. For example, some heads for the prompt *The capital city of Japan is __* decode to words like "cities".

#### 2.2.2. DEFINITIONS

Based on these observations, we propose the following definitions to aid in subsequent experiments.

**Lexical Task Heads**. We define lexical task heads as attention heads that encode interpretable task information in vocabulary space.

Lexical Task Head *Per-Prompt*: A head is classified as a lexical task head for a specific prompt if the task is verbalized in its output. In this work, our criterion is that at least $n$ of its top $k$ decoded tokens match a predefined set of task-descriptive terms (where $n = 1$ and $k = 10$). See Appendix A.2.1 for these terms.

We are interested in finding internal circuitry that is shared across all instances of a *task*, where we assume that the same task might be triggered by multiple *prompts*. We define the lexical task heads for a given set of prompts as follows:

Lexical Task Head *Per-Prompt-Style*: A head is considered a lexical task head for a given prompt style if it consistently represents a task in the vocabulary space across many prompts of the prompting style. In this work, the criteria is that it satisfies the per-prompt criteria for at least $p\%$ of prompts in a prompt style for a given task (where $p = 10$). See quantification of identified lexical task heads in Appendix D and sensitivity analyses for parameters $n$, $k$ and $p$

| Prompt | Head Index | Top Decoded Vocab |
|---|---|---|
| The capital city of Japan is __ | (L16, H25) | _cities, _Cities, 城市(city), _towns, cities |
| | (L17, H3) | _city, -city, _urban, city, _City |
| England: London; Korea: Seoul; Japan: | (L16, H25) | _cities, _Cities, _towns, cities, 城市(city) |
| | (L17, H4) | _cities, _town, _towns, cities, _Cities |
| The company that produces Mini E is | (L16, H25) | _companies, companies, Companies, _firms |
| | (L18, H24) | _company, _compan, _firm, _Company, 公司(company) |
| The antonym of push is __ | (L13, H22) | _opposite, op, _reverse, ful, _противоп (opposite) |
| | (L17, H20) | _opposite, _contrario, _ngược (reverse), _contrary, |
| dog: dogs; shoe: shoes; mouse: | (L13, H22) | _multiple, _Multiple, Multiple, extended, multiple |
| | (L18, H8) | _trio, _dozen, _multiple, tec, _multiples |

*Table 1.* We identify "lexical task heads", which decode to human-interpretable descriptions of the task in vocabulary space.

in Appendix E.

**Retrieval Heads.** Our initial explorations of decoding attention head outputs also reveal a different set of heads we name "retrieval heads". They differ from lexical task heads in that they decode to the correct answer (e.g., "Tokyo"), as opposed to the task (e.g., "capital city"). We hypothesize that lexical task heads provide task information to retrieval heads such that they can retrieve the task-relevant attribute (e.g., "Tokyo") of a given subject (e.g., "Japan"). The evidence supporting this hypothesis is described in §2.4.1, and the definition of retrieval heads is detailed in Appendix J.

### 2.3. Shared Task Mechanism for Different Prompts

Given the definitions above, we return to our initial question: Do different prompting styles activate shared circuits across prompting styles?

#### 2.3.1. DIRECT COMPONENT OVERLAP

We measure circuit reuse using the lexical task heads and retrieval heads as defined above (§2.2.2). For each task (e.g., country-capital, antonym, etc.), we collect a separate set of heads for each prompting style—that is, one set for example-based prompts and another for instruction-based prompts. We focus only on attention heads that are consistently activated by prompts that produce *correct* answers for a prompting style. We describe the results for lexical task heads below and provide the analysis for retrieval heads in Appendix J.4.

Fig. 3 shows the overlap between lexical task heads identified for the instruction-based vs. example-based prompting styles for each task. On average, 73% of lexical task heads for a given task are shared across the two prompting styles. This result suggests that some components are reused when processing different surface instantiations of the same underlying task.

Notably, while these components are largely invariant to different prompt forms of a task, they are specific to task types. Unrelated tasks (e.g., translation tasks) often recruit a distinct set of heads, whereas similar tasks (e.g., different knowledge retrieval tasks) share a significant portion of their lexical task heads. That is, while it is generally true that a given attention head can be reused by different tasks, it doesn't mean that it functions the same way across tasks, and in fact might often generate distinct outputs for each task. It is therefore interesting that lexical task heads are not just shared across prompts, but in fact generate the *same outputs* for different prompts of the same underlying task. For example, the attention head (L16, H25) in Llama-3.1-8B-Instruct outputs "_cities"-related tokens for both example-based and instruction-based prompts of the Country–Capital task, while it decodes to "_companies"-

related tokens for the Product–Producer task (see Table 1).

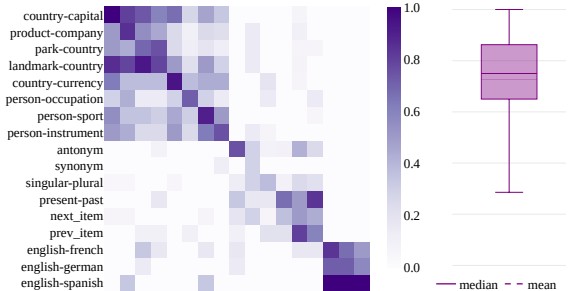

*Figure 3.* Lexical task heads are shared across prompting styles. Left: X-axis of the heatmap displays tasks for instruction-based prompts and Y-axis for example-based prompts. The darker the color, the more % of heads overlap for a given task across the two prompting styles. Right: Quantification of the overlap for each task (the diagonal of the heatmap). On average, 73% of the lexical task components of each task are shared across prompting styles. See Appendix F for results on other models.

#### 2.3.2. FUNCTIONAL EQUIVALENCE ACROSS PROMPTING STYLES

Given the above evidence of direct overlap between lexical task heads, we next ask whether these heads exhibit functional equivalence across prompting styles. That is, do lexical task heads generate *shared task representations*?

We use activation patching to determine whether these heads perform similar roles across different prompts. To do this, we activate lexical task heads by patching their activations from one prompt template into another one (see examples in Table 2) and measure the change in task accuracy induced by this patching intervention. We patch to zero-shot prompts so that we can test if lexical task heads can makeup the lack of task information in these prompts.

| Cache *average* activations from | Patch to |
|---|---|
| Instruction-based prompts (of target task): The capital city of Japan is ___ | |
| Example-based prompts (of target task): Korea: Seoul; Japan: ___ | Japan: ___ |
| Control prompts (of a related task): The currency used in Japan is ___ | |

*Table 2.* A table of sample prompts used to cache the activations from and patch to.

In Fig. 4, we show that lexical task heads generate outputs that are functionally equivalent across different prompts. Specifically, lexical task head activations from different prompts achieve similar causal effect, suggesting that lexical task heads produce a common task signal across prompts. We use average activations to make sure that generic task information, instead of specific answers are patched in. For each colored line, lexical task heads are activated by mul-

tiplying the average activations by a scalar, resulting in an increase in task accuracy above the baseline (gray solid line) performance of the zero-shot prompts that generate incorrect answers.

We conduct a control experiment to show that while a related task may share a subset of lexical task heads with the target task (Fig. 3), the activations from a related task are sufficiently distinct that they fail to exert the same causal influence on the target task (see gray dashed line). Thus, activations across prompt styles for the same task are more similar than activations across tasks that share lexical task heads. The causal effect of lexical task heads varies across tasks, where at least 10% to 90% of task performance is recovered across prompts (Appendix G.1).

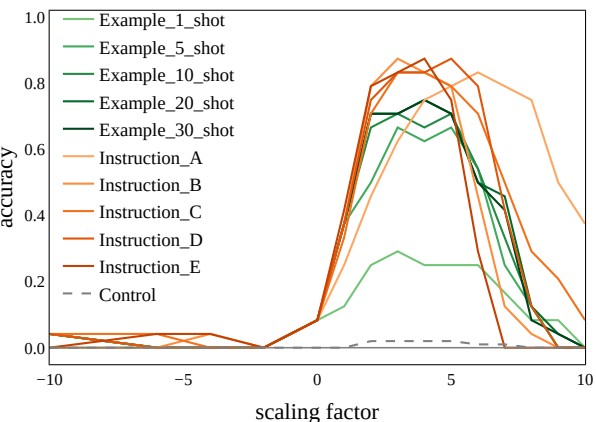

*Figure 4.* The shared lexical task heads produce functionally equivalent outputs across prompting styles. Each solid line represents an activation patching experiment. For all lines, the activations of a same set of heads are patched to same prompts. The only difference is the source of the patched activation, which is cached from different prompting templates and styles, represented by different colors. Instruction templates A-E are wording variations (details in Appendix A.3.1).

## 2.4. Analysis of Failure Cases

The above results demonstrate that LLMs create shared task representations across prompting styles, at least in the cases where the model answers correctly. We then investigate the next question: can the shared lexical task heads explain behavioral variability? That is, if the lexical task heads generate lexical task representations when models are able to output the correct answers for a task, do lexical task heads give rise to different representations when the models *fail*?

We hypothesize that failures stem from the correct components being insufficiently activated. As a result, the target task signal is too weak to overcome a circuit that does something else entirely, e.g., the background "noise" of just freely associating related words. For example, in the Product-

Producer task, while the LLM might be capable of retrieving the producer ("Dodge") for a given product ("Bluesmobile"), it sometimes outputs the name of the movie ("The Blues Brothers"), where "Bluesmobile" was prominently featured (Fig. 2).

### 2.4.1. LEXICAL TASK HEADS ARE INSUFFICIENTLY ACTIVATED

We first show that the model sometimes produces wrong answers due to insufficient activation of the correct components of the target task. We present four analyses below.

**Correlation with model correctness**. We first compare the lexical task heads in success and failure cases. We find a positive correlation between the properties of the lexical task heads and the correctness of the model's final outputs. Specifically, prompts that result in correct answers activate a higher number of shared heads compared to failing prompts. Beyond the number of activated heads, we also compare their activation strength by calculating the $L_2$ norm of their outputs and find the average norm across lexical task heads is also higher in correct prompts (Fig. 5). These results suggest that different activation levels of the shared heads can lead to different outputs (see detailed analysis in Appendix H.1.1).

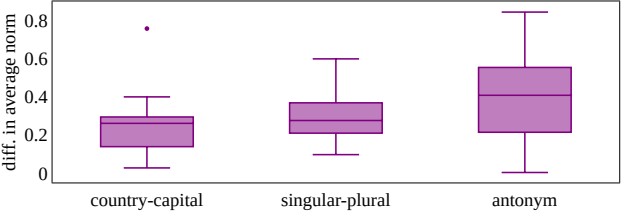

*Figure 5.* Prompts that generate correct answers activate lexical task heads more strongly. The y-axis is the difference of attention head output norms between correct and incorrect prompts.

**Explaining many-shot effect in In-Context Learning (ICL)**. To further study the correlation between the model's performance and its internal components, we tend to a well-documented "many-shot" phenomenon in LLMs: prompts with higher number of demonstration pairs (shots) in ICL (example-based prompting) have higher accuracy. Our analysis suggests that lexical task components could be a mechanism behind this phenomenon. We observe that as the number of shots increases, both the quantity of lexical task heads and their output norms rise accordingly (Fig. 6). This relationship holds across a broad spectrum of tasks. For 14 out of 17 evaluated tasks, the number or the output strength of lexical task heads positively correlates with task accuracy. These results suggest that the sufficient activation of lexical task heads is a critical mechanism for achieving higher performance (see detailed analysis in Appendix H.1.2).

**Increasing the activation of the lexical task components**

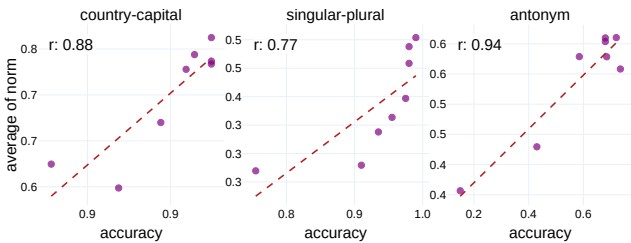

*Figure 6.* There is positive correlation between the accuracy and the magnitude of the outputs of lexical task heads. In each subplot for a task, each dot represents a given shot count. The plots for all the tasks are in Appendix H.1.2.

**fixes incorrect prompts**. Beyond correlation, we ask if increasing the activation of the target task can repair prompts that originally produce wrong answers. To do so, we conduct causal intervention experiments (activation patching). We first sample some incorrect prompts and some correct prompts. Then we test if we can fix the incorrect prompts by increasing the activation of the lexical task heads (of the target task). Specifically, in incorrect prompts, we replace lexical task heads' activation with the average activation cached from the correct prompts. We show that scaling up this average activation can fix a portion of the incorrect prompts (Fig. 7). This result provides causal evidence that activating lexical task heads to a sufficient level can indeed help the model to overcome the background noise in order to correctly perform a task.

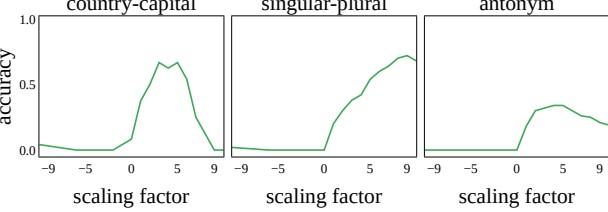

*Figure 7.* Scaling up the activation of lexical task heads can fix a portion of originally failed prompts. The average outputs of lexical task heads from correct prompts are patched to the incorrect prompts. The baseline accuracy is 0 for the incorrect prompts. See Appendix H.1.3 for results of all the tasks.

**Interaction between task representation and execution.** We further explore the mechanisms by which lexical task representations enable the model to generate correct answers. We hypothesize that these representations act as a directional filter for retrieval heads, narrowing the broad space of subject-related attributes down to the specific task-relevant answer. To test this, we measure the impact of manipulating lexical task head activations on retrieval heads' performance. Using the same intervention setup in Fig. 7, we find that strengthening the output of lexical task heads directly modulates retrieval heads: as lexical task head activations increases, a greater number of retrieval heads correctly identify the target answer. Furthermore, for each individual

retrieval head, the logits attributed to the correct answer show a corresponding increase (Fig. 8).

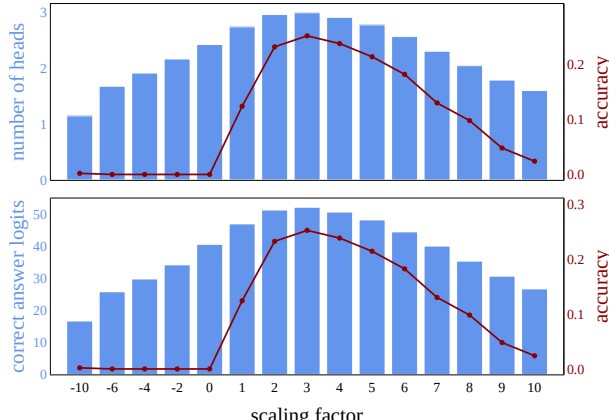

*Figure 8.* Lexical task heads modulate the output of retrieval heads. After scaling up lexical task head activations, more retrieval heads retrieve the correct answer (top), and the logits of the correct answer contributed by these heads also increase (bottom).

The four analyses presented above provide a mechanistic account of how differential activation of shared components dictates model output. Crucially, lexical task components do not generate the final answer directly; rather, they encode the task and modulate the retrieval circuitry to ensure the model extracts the precise attribute required by the prompt. When activation of these lexical task components is insufficient, the downstream retrieval heads fail to be triggered correctly, leading to incorrect outputs.

### 2.4.2. PROMPT AMBIGUITY TRIGGERS COMPETING TASKS

In previous section, we show that failures can stem from insufficient activation of the target task. This raises the question, then: what other circuits are triggered when the model produces an incorrect answer? In addition to generic errors like free association, we explore a specific scenario where some features of the prompt cause the model to trigger a competing task, diluting the signals for the target task.

The investigation stems from a curious pattern in the model's errors. We noticed that the ambiguity in certain prompts within the datasets could potentially mislead the model to perform a different task than the target task. Consider the following prompt, "Siri: Apple; Hwasong-6: North Korea; Bluesmobile: ___". The intended target task is Product–Producer, however, the relationship between "Hwasong-6" and "North Korea" could represent either a Product–Producer or a Product–Country mapping. As a result, the model incorrectly predicts the product's country ("America") rather than its producer ("Dodge"). We hypothesize that these kind of ambiguous prompts cause the model to trigger the internal circuits of a competing task (Fig. 2).

To test this hypothesis, we compare ambiguous prompts that imply a competing task against those that do not. The ambiguous prompts contain the same misleading demonstration pair (e.g., "Hwasong-6: North Korea"), and the query words are a set of different products. The control prompts contain the same number of demonstration pairs and the same set of query words, but the demonstration pairs are not ambiguous (at least to an average human, e.g., "Gmail: Google"). We measure the lexical task head activations of the intended target task (e.g., "Product-Producer") and the competing off-target task (e.g., "Product-Country").

Our analysis in Fig. 9 shows that ambiguous prompts tend to activate fewer lexical task heads for the target task, with diminished activation strength (see additional analysis in Appendix K.1). In contrast, ambiguous prompts have more strongly activated lexical task heads of the off-target task, compared to the control prompts. In addition, we replicate these ambiguity results in a different "number list" task (see Appendix K.2), which is not a knowledge retrieval task like the Product-Producer task. These exploratory analyses provide initial evidence that some prompts can inadvertently trigger a competing circuit. This "off-target" activation dilutes the signal for the intended task, ultimately steering the model toward the wrong output.

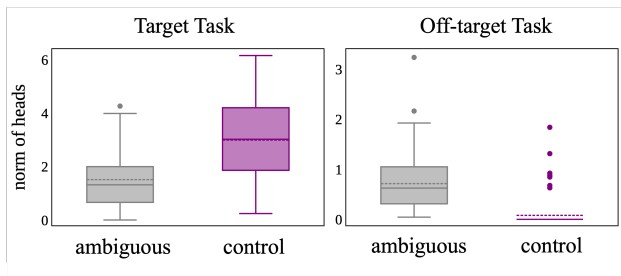

*Figure 9.* Ambiguous prompts trigger the internal circuits of a competing task, diluting the signals of the intended target task.

### 2.5. Generalization To More Complex Tasks

We investigate if lexical task representations generalize to more complex compositional or free-form generation tasks.

#### 2.5.1. COMPOSITIONAL TASKS

As a proof of concept, we test lexical task representation mechanisms on several two-hop compositional tasks studied in prior work (Khandelwal & Pavlick, 2025). We first identify lexical task heads for each of the two composing tasks using the same process as described in §2.2. For example, for the Park-Country-Capital task, we separately detect "country" and "capital city" heads. We then validate the causal effect of the lexical task heads of both composing tasks, using activation patching experiments as described in §2.3.2. We find that activating lexical task heads recovers

12%–50% of the performance across tasks (Fig. 10). See detailed analysis of compositional tasks in Appendix M.

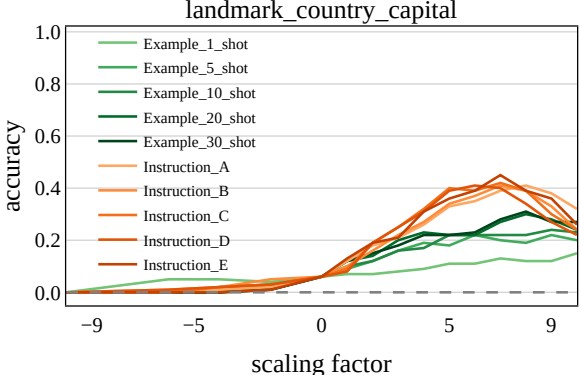

*Figure 10.* Quantification of the causal effect of lexical task heads.

#### 2.5.2. CODE GENERATION TASKS

For free form generation, we test lexical task representation mechanisms on a code generation task, using Python and JavaScript coding problems from the HumanEval-X dataset (Zheng et al., 2024). First, we identify "Python" and "JavaScript" lexical task heads with the criterion described in §2.2.2: at least 1 of the top 10 decoded tokens match a set of task-descriptive terms. These terms include single-token words (e.g., "python", "javascript") and their close variants, such as those with modified capitalization or special symbols (e.g., "_python"). Table 3 shows examples of decoded lexical task head outputs.

| Language | Head Index | Top Decoded Vocab |
|---|---|---|
| Python | L9, H13 | _Python, _PYTHON, _python |
| | L27, H28 | _Python, 编程(programming) |
| | L24, H26 | _pytest, .py, PYTHON |
| JavaScript | L09, H13 | _Javascript, _javascript, _code |
| | L14, H07 | _JavaScript, [js, _javascript |
| | L23, H10 | _JavaScript, 编程(programming) |

*Table 3.* Early-decoded outputs of lexical task heads.

To validate the causal role of the identified lexical task heads, we activate the identified lexical task heads to test if we can steer the model to generate code in a specific programming language. We find that the steering effect depends on the activation strength (scaling factor) applied. For instance, when "Python" heads are activated with a moderate scaling factor, the model switches from generating JavaScript to Python code, generating corresponding Python completions 90% of the time. When the scaling factor is small, the model continues to generate Javascript code matching the prompt. When the scaling factor is too strong, the model's generation deteriorates to repeating "Python Python". We include

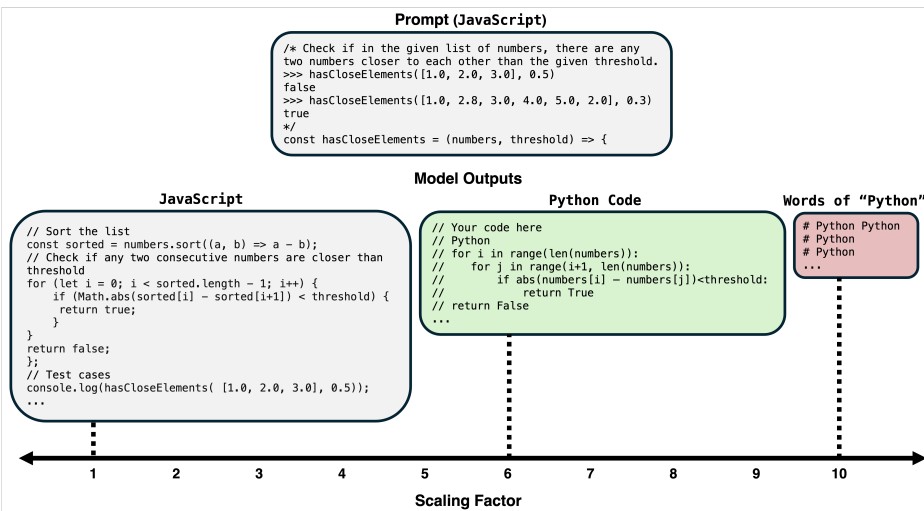

*Figure 11.* Steering the model to generate Python code by activating lexical task heads. Left: When the scaling factor is small, the model continues to generate Javascript code. Middle: With a moderate scaling factor, the model switches from generating JavaScript to Python code. Right: When the scaling factor is too strong, the model's generation deteriorates to repeating "Python".

detailed analysis of code generation tasks in Appendix L.

## 3. Discussion

Our investigation starts with a behavioral puzzle: why do Large Language Models (LLMs) exhibit sensitivity to prompt phrasing? Our primary contribution is the discovery of lexical task heads—attention heads that encode task information in the model's vocabulary space. We show that lexical task heads are shared across prompting styles and the degree to which these heads are activated can explain behavioral variance. We show that prompt ambiguity induces competing task representations that dilute the signal of the target task. Overall, we attempt to bridge several levels of analysis: from the behavioral phenomenon of prompt sensitivity, to the representational strategy of mapping tasks to words, and its implementation in transformer circuits.

At the implementation level, our findings offer a mechanistic explanation for the efficacy of many-shot prompting that aligns with the "Bayesian belief dynamics" model proposed by Bigelow et al. (2025). They model ICL as a Bayesian process where examples act as accumulating evidence, gradually shifting the model's posterior belief in a latent concept. Our findings on lexical task heads might be one of the implementation substrates of this process: We observe that as the number of examples increases, the quantity and the output norm of the lexical task heads rises. This result suggests that the "accumulation of evidence" described by Bigelow et al. (2025) can be implemented by the lexical task heads.

Our analysis that some failures can be caused by ambiguity in the prompts that trigger circuits of off-target tasks. This aligns with recent findings by Xiong et al. (2024), who

demonstrate that transformers exist in a state of "task superposition", capable of implementing multiple tasks in parallel. Cho et al. (2025) also proposed that in the zero-shot scenario, the model's hidden states contain information of many possible tasks, and in-context examples act to filter out non-selective representations and leave only the relevant signal. Our work shows that in-context examples can sometimes induce ambiguity, triggering a competing task and leading to the wrong answer.

Finally, the nature of lexical task representations has implications for AI interpretability. Several recent works study whether LM activations can be decoded into natural language explanations (Ghandeharioun et al., 2024; Chen et al., 2024; Pan et al., 2026; Li et al., 2025a; Karvonen et al., 2026; Fraser-Taliente et al., 2026), though Li et al. (2025b) argue that current evaluations are not sufficiently faithful. We find lexical task representations *can* be understood directly in vocabulary space, suggesting that vocabulary itself might act as a universal language for inter-model understanding.

## 4. Limitations

While our work provides a mechanistic account of prompt sensitivity through the lens of shared mechanisms, several limitations remain that offer avenues for future research.

In this work, we investigate mechanisms that are shared across two prompting styles of example-based and instruction-based prompting. We illustrate how the shared lexical task heads and retrieval heads can explain behavioral variability. However, there are differences in how these two prompting styles are processed by the model, including a different kind of task representations via function vec-

tors (Hendel et al., 2023; Todd et al., 2024; Davidson et al., 2025). Because causal effects of lexical task heads vary from 10% to 90% across tasks, this suggests that lexical task representations only account for a partial mechanism for task representation with varying significance across tasks.

In addition, the initial phase of "task detection" likely differs significantly: example-based prompts require the model to infer task information implicitly from demonstration pairs, whereas instruction-based prompts provide this explicitly.

Our methodology for identifying lexical task heads depends on projecting component outputs into vocabulary space using the Logit Lens (Nostalgebraist, 2020). It's possible that some task representations are encoded in a non-lexical, abstract format that does not project cleanly to human-readable tokens. Consequently, our study may overlook non-lexical components also contributing to task representation.

The criteria used to define lexical task heads—specifically the parameters $n$, $k$, and $p$—are heuristic in nature. Although we provide sensitivity analyses for these parameters (Appendix E), the choice of the predefined task-descriptive vocabulary lists remains subjective and may influence the specific set of heads identified for each task. As an attempt to address this, we leverage several frontier LLMs to generate task-descriptive terms. We find that the lexical task heads identified with those terms are comparable to the originally identified ones (Appendix O).

This study focuses on tasks that have simple lexical descriptions. It remains unclear if these shared lexical mechanisms hold for more abstract tasks or tasks that are less easily defined by simple vocabulary clusters. We acknowledge that prompt sensitivity is a complex phenomenon and cannot be fully explained by the shared mechanisms and the limited tasks studied here. This work provides a step toward a mechanistic understanding of prompt sensitivity and encourages further investigation into how models' internal mechanisms drive behavioral variability.

## 5. Related Work

**Prompt Format Sensitivity:** It is well-known that LLMs exhibit sensitivity to different prompt formulations of the same task (Errica et al., 2025; Shafiei et al., 2025; Cao et al., 2026). Reynolds & McDonell (2021) demonstrate that instruction prompts can sometimes outperform few-shot prompts, while Liu et al. (2024a) document several inconsistencies between instruction following and few-shot prompting for the same task. Kirsanov et al. (2025) show learned-prompt techniques such as soft prompting (Lester et al., 2021) for task adaptation operate distinctly from few-shot and instruction prompting, but can still achieve similar performance. Our work demonstrates the difference in activation of shared lexical task components can explain part of

the performance variance across prompting styles.

**Task Representations in LLMs:** Our work builds on previous findings identifying vector representations of tasks in LLMs called function or task vectors that emerge during in-context learning (Todd et al., 2024; Hendel et al., 2023; Liu et al., 2024b). Function vectors are extracted from attention heads found to be important for ICL tasks, though Davidson et al. (2025) find that for instruction prompting, LLMs leverage a different set of attention heads, aligning with Nam et al. (2025)'s observation that instruction following and in-context learning rely on separable mechanisms. Our work identifies a distinct mechanism for task representation, which differs from function vector heads in two ways: lexical task components are 1) shared across prompting styles; and 2) human-interpretable, as their outputs decode to descriptions of the task itself in the vocabulary space.

**Mechanisms of In-Context Learning:** There has been substantial prior work studying the mechanisms underlying different aspects of in-context learning (ICL) (Lampinen et al., 2025). Several works have studied in-context copying via induction heads (Olsson et al., 2022; Singh et al., 2024; Crosbie & Shutova, 2025; Feucht et al., 2025). However, Yin & Steinhardt (2025) shows that function vector heads (Todd et al., 2024), not induction heads, are primarily responsible for few-shot ICL performance. Other work has decomposes ICL into distinct computational stages, such as Kharlapenko et al. (2025) who study SAE feature-circuits of ICL prompts and find latents can be clustered as either "task-detection" or "task-execution" features. Similarly, Bakalova et al. (2025) identify few-shot ICL circuits and describe them as a two-step process of "contextualization" and "aggregation" by which task information is processed and then transported through label tokens (Wang et al., 2023b) to the final token. Our findings identify a similar two-step process where lexical task representations modulate retrieval mechanisms to produce correct outputs.

**Computational Reuse in Transformers:** A few works have studied how transformer components are reused across different tasks. Merullo et al. (2024) show many of the heads in the Indirect Object Identification (IOI) circuit (Wang et al., 2023a) perform the same functional roles for a related but distinct colored object task. Similary, Lan et al. (2024) show a key sub-circuit is reused for similar sequence continuation tasks such as numerals, months and number words, and Bhaskar et al. (2024) identify a "heuristic core" of attention heads that are important across several NLP tasks. Our work shows that computational reuse occurs not only across tasks, but also across different prompting styles for the same task.

Code is available at https://github.com/jojozyang/Lexical_Task_Representations.git

## Acknowledgement

We thank Eric A. Miller for helpful discussions throughout the project. We are also grateful to Eric A. Miller, Michael A. Lepori and Tianze Hua for their feedback on the manuscript.

This project was in part supported by Schmidt Sciences Grant #GR5300958 and a Young Faculty Award from the Defense Advanced Research Projects Agency, United States, Grant #D24AP00261. Eric Todd and David Bau are supported by a grant from Coefficient Giving.

Disclosure: Stephen Bach is an advisor to Snorkel AI, a company that provides software and services for data-centric artificial intelligence. Ellie Pavlick is a paid consultant for Google DeepMind. The content of this article does not necessarily reflect the views of the US Government or of Google, and no official endorsement of this work should be inferred.

## Impact Statement

This paper presents work whose goal is to advance the field of Machine Learning. There are many potential societal consequences of our work, none of which we feel must be specifically highlighted here.

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

# A. Experimental Setup

## A.1. Models

We selected models of different sizes and families to corroborate that our findings are robust and transfer across an appropriate range of models. The results shown in the main text are from Llama-3.1-8B-Instruct model.

- **Llama**

    - Llama-3.1-8B (Grattafiori et al., 2024)
    - Llama-3.1-8B-Instruct (Grattafiori et al., 2024)
    - Llama-3.1-70B-Instruct (Grattafiori et al., 2024)

- **Gemma**

    - gemma-2-9b (Gemma Team et al., 2024)
    - gemma-2-9b-it (Gemma Team et al., 2024)
    - gemma-2-27b-it (Gemma Team et al., 2024)

- **Qwen**

    - Qwen2.5-7B (Qwen et al., 2025)
    - Qwen2.5-7B-Instruct (Qwen et al., 2025)
    - Qwen2.5-32B-Instruct  (Qwen et al., 2025)
    - Qwen/Qwen3-4B-Instruct-2507 (Yang et al., 2025)
    - Qwen/Qwen3-4B-Thinking-2507 (Yang et al., 2025)
    - Qwen/Qwen3-30B-A3B-Instruct-2507 (Yang et al., 2025)
    - Qwen/Qwen3-30B-A3B-Thinking-2507 (Yang et al., 2025)

## A.2. Tasks

We use a set of relatively simple tasks that have been utilized in prior interpretability research (Hernandez et al., 2024; Todd et al., 2024; Davidson et al., 2025). Below, we detail each of the 17 tasks:

1. **product-producer:** Identify the producer or company associated with a given product.

2. **antonym:** Provide a word with the opposite meaning of a given word.

3. **synonym:** Provide a word with a similar meaning to a given word.

4. **singular-plural:** Convert a singular noun to its plural form, or identify multiplicity.

5. **country-capital:** Provide the capital city of a given country.

6. **park-country:** Identify the country in which a given park is located.

7. **next_item:** Provide the next item in a given sequence.

8. **prev_item:** Provide the previous item in a given sequence.

9. **english-french:** Translate an English word to French.

10. **english-german:** Translate an English word to German.

11. **english-spanish:** Translate an English word to Spanish.

12. **present-past:** Convert a verb from its present tense to past tense.

13. **person-occupation:** Provide the occupation or job of a given person.

14. **person-sport:** Identify the sport associated with a given athlete or person.

15. **person-instrument:** Identify the musical instrument played by a given musician.

16. **country-currency:** Provide the currency used in a given country.

17. **landmark-country:** Identify the country in which a given landmark is located.

### A.2.1. TASK DESCRIPTIVE TERMS

For each of the 17 tasks, we predefine a set of words that serve as the task-descriptive terms for a given task. These terms are used to identify the lexical task heads (See methods in §2.2.2).

| Task | Task-descriptive Terms |
|---|---|
| product-producer | manufacturers, company, organisation, Companies, _company, companies, corporations, .company, companies, organisation, institutions, firm, organisations, Company, corporation, company, institution, firm, -company, firms, institution, producer |
| antonym | reverse, reverse, .reverse, opposing, Reverse, _reverse, opposite, 反 |
| synonym | interchangeable, synonym, Similar, similar, analogous, equivalent, similar, alike, Equivalent |
| singular-plural | Multiplicity, Multiple, ones, 多, _multiple, _plural, ones, .ones, ONES, multiple, plural, multiple, plural |
| country-capital | city, _city, cities, -city, -capital, Cities, city, cities, capital, capital, City, capitals, .city, Capital |
| park-country | country, nation, countries, 国家, Countries, countries, .country, -country, _country, _countries, country, Country |
| next_item | .next, NEXT, following, _next, After, next, after, next, Next, _after, following, Following, follow, 接着, -next, after, Following, -after |
| prev_item | before, .last, before, _last, previous, Previously, Earlier, _before, _previous, .before, Previous, Before, last, LAST, previous, -last, Last, earlier, -before, previously, last, .previous |
| english-french | French, french, français, Français |
| english-german | german, German, Deutsch |
| english-spanish | Spanish, spanish, español, Español |
| present-past | past, Earlier, previously, Previously, past, Past, Earlier, _past |
| person-occupation | occupation, jobs, JOB, career, -job, jobs, Jobs, career, Career, .job, job, _job, job, .jobs, _jobs, occupation, Job |
| person-sport | .game, .games, Game, sport, games, games, _games, -game, sport, game, Games, Sports, GAME, game, SPORT, _game, Sport, sports, sports |
| person-instrument | instrument, devices, instrument, .device, device, DEVICE, Devices, apparatus, device, Instrument, .devices, _devices, instruments, devices, .instrument, _device, -device, Device |
| country-currency | -money, money, .money, _currency, money, currency, currency, Currency, Money, _money, currencies, .currency |
| landmark-country | Country, country, nation, countries, 国家, Countries, countries, .country, -country, nation, _country, Nation, _countries, country, nation |

### A.3. Prompts

**Prompting Styles**

We focus on two main prompting styles and provide examples below. In each prompt, we mark the source of two pieces of critical information, task or subject, in different colors.

- *Example-based Prompting*: Prompts that contain demonstration pairs, from which task information can be inferred. This is also known as in-context-learning (ICL) and few-shot prompting.
  *e.g." Germany: Berlin, Greece: Athens, ..., Japan: _____"*

- *Instruction-based Prompting*: Prompts that contain explicit description or instruction of a task.
  *e.g." Given a country, output its capital city. Q: Japan. A: _____"*

**Prompt Templates**

There are two ways to construct different *example-based prompting* templates.
1) Vary the number of demonstration pairs / shots:

- *" Germany: Berlin, Japan: _____"* (1-shot)
- *" Germany: Berlin, Greece: Athens, Japan: _____"* (2-shot)

2) Use different demonstration pairs:

- *" Germany: Berlin, Greece: Athens, Japan: _____"*
- *" Peru: Lima, France: Paris, Japan: _____"*

Many different prompt templates can be constructed for *instruction-based prompting*. e.g.

- Template A: *" What is the capital city of the country? Q: Japan. A: _____"*
- Template B: *" Tell me the capital city of Japan. A: _____"*
- Template C: *" The capital city of Japan is _____"*

A.3.1. TEMPLATES FOR INSTRUCTION-BASED PROMPTS

The templates for instruction-based prompts of each task is listed below.

| Task Name | Instruction |
|---|---|
| antonym | <ul><li>The antonym of {instance_input} is</li><li>Output the antonym of the word in the question. Q: {instance_input}\nA:</li><li>Give me the antonym of the following word. Q: {instance_input}\nA:</li><li>What word is the opposite of the word in the question? Q: {instance_input}\nA:</li><li>Give an antonym for the following word: Q: {instance_input}\nA:</li><li>Task Definition: Given a word, provide its antithesis. Q: {instance_input}\nA:</li></ul> |
| country-capital | <ul><li>The capital city of {instance_input} is</li><li>What is the capital of the country? Q: {instance_input}\nA:</li><li>Country -> Capital Q: {instance_input}\nA:</li><li>Task: given the country, answer the capital of the country. Q: {instance_input}\nA:</li><li>What capital city does this country have? Q: {instance_input}\nA:</li><li>Instructions: you are given a country and you need to output the capital city of that country. Q: {instance_input}\nA:</li></ul> |

*Continued from previous page*

| Task Name | Instruction |
|---|---|
| country-currency | • The currency of {instance_input} is the
• What is the currency used by the country? Q: {instance_input}\nA:
• Country -> Currency Q: {instance_input}\nA:
• Task: given the country, answer with the currency of the country. Q: {instance_input}\nA:
• What currency does this country use? Q: {instance_input}\nA:
• Instructions: you are given a country and you need to output the currency of that country. Q: {instance_input}\nA: |
| english-french | • The French translation of {instance_input} is
• Translate English to French: Q: {instance_input}\nA:
• What is the French translation of the following English sentence? Q: {instance_input}\nA:
• Given an English word, can you translate it into French? Q: {instance_input}\nA:
• Write the translation of the following sentence in French. Q: {instance_input}\nA:
• Tell me what the following word means in French. Q: {instance_input}\nA: |
| english-german | • The German translation of {instance_input} is
• Translate the following English word to German. Q: {instance_input}\nA:
• What is the German translation of the following English sentence? Q: {instance_input}\nA:
• Given an English word, can you translate it into German? Q: {instance_input}\nA:
• Write the translation of the following sentence in German. Q: {instance_input}\nA:
• Tell me what the following word means in German. Q: {instance_input}\nA: |
| english-spanish | • The Spanish translation of {instance_input} is
• Translate the following English sentences to Spanish. Q: {instance_input}\nA:
• Translate the following English word to Spanish. Q: {instance_input}\nA:
• What is the translation of the word in Spanish? Q: {instance_input}\nA:
• English to Spanish translation: Q: {instance_input}\nA:
• Task: English -> Spanish. Q: {instance_input}\nA: |
| landmark-country | • The country where {instance_input} is located is
• Which country does this landmark belong to? Q: {instance_input}\nA:
• Return the country that this landmark is in. Q: {instance_input}\nA:
• Help me find the country that the mentioned landmark is located in. Q: {instance_input}\nA:
• Landmark -> Country. Q: {instance_input}\nA:
• Tell me which country has the following landmark. Q: {instance_input}\nA: |
| next_item | • The next item after {instance_input} is
• What is the next item in the sequence? Q: {instance_input}\nA:
• Given the current item, what is the next item in the sequence? Q: {instance_input}\nA:
• Find the next item. Q: {instance_input}\nA:
• Return the item that is sequentially next of the given input Q: {instance_input}\nA:
• With the given item, discover the next item in the sequence. Q: {instance_input}\nA: |
| park-country | • The country where {instance_input} is located is
• What is the country of the park? Q: {instance_input}\nA:
• Which nation does the park belong to? Q: {instance_input}\nA:
• Park -> Country Q: {instance_input}\nA:
• Which country is this place located in? Q: {instance_input}\nA:
• Country? Q: {instance_input}\nA: |

*Continued from previous page*

| Task Name | Instruction |
|---|---|
| person-instrument | • The instrument {instance_input} plays is the
• Classify the musician into the instrument they play. Categories are: guitar, piano, violin, and trumpet. Q: {instance_input}\nA:
• Classify the musician into one of the following categories: "guitar", "piano", "violin", or "trumpet". Q: {instance_input}\nA:
• Output one of the "guitar", "piano", "violin", or "trumpet" to indicate the instrument the musician plays. Q: {instance_input}\nA:
• You are given the name of a musician. Your task is to choose the instrument that the musician plays from "guitar", "piano", "violin", or "trumpet". Q: {instance_input}\nA:
• Tell me which instrument is this musician known for? Hint: it's one of "guitar", "piano", "violin", and "trumpet". Q: {instance_input}\nA: |
| person-occupation | • The occupation of {instance_input} is a/an
• What is the occupation of the celebrity? Please provide a one-word answer in lowercase. Q: {instance_input}\nA:
• The occupation (in lower case) of the celebrity is? Q: {instance_input}\nA:
• Return the occupation of the celebrity in lower case. Q: {instance_input}\nA:
• Task: you are given a celebrity, your task is to guess their occupation and return in lower case Q: {instance_input}\nA:
• What occupation (in lowercase) is the celebrity given in the input? Q: {instance_input}\nA: |
| person-sport | • The sport {instance_input} plays is
• Return the sport that this athlete is known for. Q: {instance_input}\nA:
• What sports does this athlete play? Q: {instance_input}\nA:
• What sport does this athlete play? Output from "baseketball", "soccer", "football", "baseball", or "hockey". Q: {instance_input}\nA:
• Person -> sport. Labels: "basketball", "soccer", "football", "baseball", "hockey" Q: {instance_input}\nA:
• Classify the athlete into one the following categories: ["basketball", "soccer", "football", "baseball", "hockey"] Q: {instance_input}\nA: |
| present-past | • The past tense of {instance_input} is
• Write the word in its past tense form. Q: {instance_input}\nA:
• What is the past tense of the following verb? Q: {instance_input}\nA:
• Give me the past tense of 'Q' Q: {instance_input}\nA:
• I wonder what the past tense of this English word is. Can you tell me? Q: {instance_input}\nA:
• Instruction: Return the past tense. Q: {instance_input}\nA: |
| prev_item | • The previous item before {instance_input} is
• Given a number, letter, or month, return the previous item in the sequence. Q: {instance_input}\nA:
• Return the previous item in the sequence. Q: {instance_input}\nA:
• Instruction: given the current item, what is the previous one? Q: {instance_input}\nA:
• You are given an item in the squence, what is the previous item? Q: {instance_input}\nA:
• What is the previous item in the sequence? Q: {instance_input}\nA: |

*Continued on next page*

*Continued from previous page*

| Task Name | Instruction |
|-----------|-------------|
| product-producer | <ul><li>The company that produces {instance_input} is</li><li>Which company is the product from? Q: {instance_input}\nA:</li><li>Which company makes the given product? Q: {instance_input}\nA:</li><li>Task definition: given a product, which company makes it? Q: {instance_input}\nA:</li><li>Product -> Company owning it Q: {instance_input}\nA:</li><li>Tell me which company makes the following product. Q: {instance_input}\nA:</li></ul> |
| singular-plural | <ul><li>The plural form of {instance_input} is</li><li>What is the plural of the following word? Q: {instance_input}\nA:</li><li>Task: Singlular to Plural Q: {instance_input}\nA:</li><li>Could you please tell me what is the plural word? Q: {instance_input}\nA:</li><li>Output the plural word from the singular Q: {instance_input}\nA:</li><li>You are an English expert. You are asked to answer the question by outputing the plural form of the word. Q: {instance_input}\nA:</li></ul> |
| synonym | <ul><li>A synonym for {instance_input} is</li><li>Output the synonym of the question. Q: {instance_input}\nA:</li><li>Task: Synonym Q: {instance_input}\nA:</li><li>Word to synonym Q: {instance_input}\nA:</li><li>What is the synonym? Q: {instance_input}\nA:</li></ul> |

# B. Behavior Variance

In this section, we analyze behavioral variance across ten models from three distinct model families. We compare the performance of example-based versus instruction-based prompting across 17 tasks. In the box plots below, each data point represents the accuracy of a specific prompt template for a given task and prompting style.

### B.1. Llama-3.1 Models

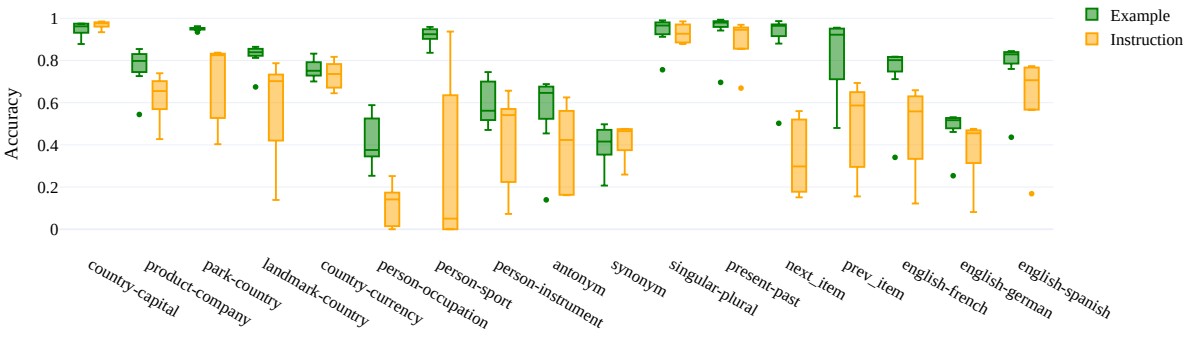

*Figure 12.* Behavior variance across all tasks for Llama-3.1-8B-Instruct model.

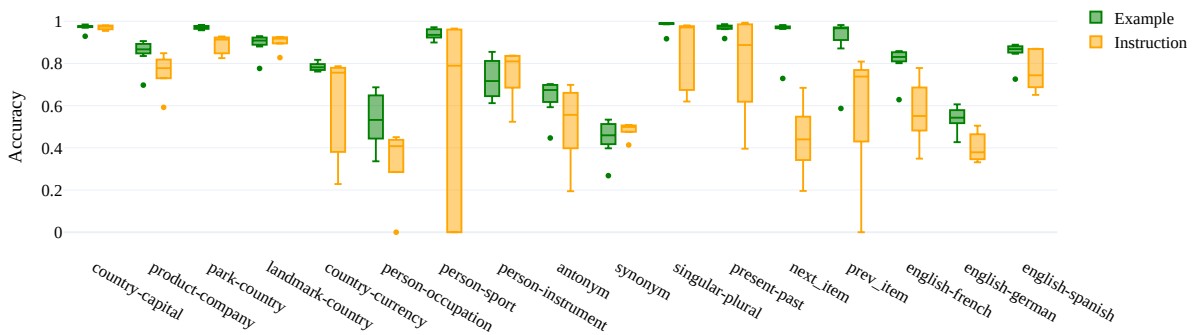

*Figure 13.* Behavior variance across all tasks for Llama-3.1-70B-Instruct model.

## B.2. Gemma-2 Models

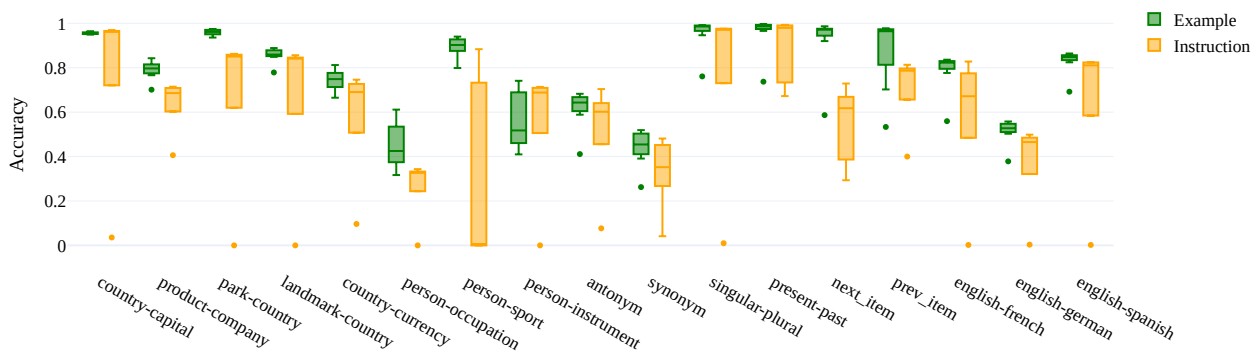

*Figure 14.* Behavior variance across all tasks for gemma-2-9b-it model.

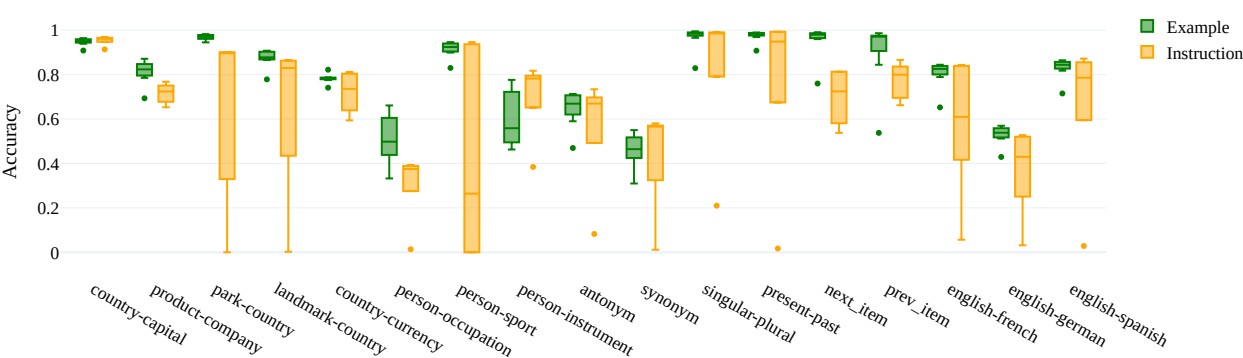

*Figure 15.* Behavior variance across all tasks for gemma-2-27b-it model.

## B.3. Qwen-2.5 Models

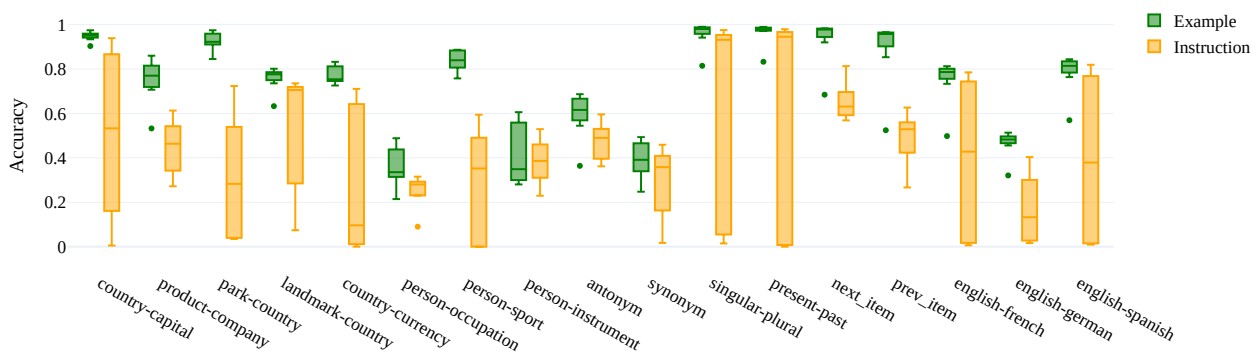

*Figure 16.* Behavior variance across all tasks for Qwen2.5-7B-Instruct model.

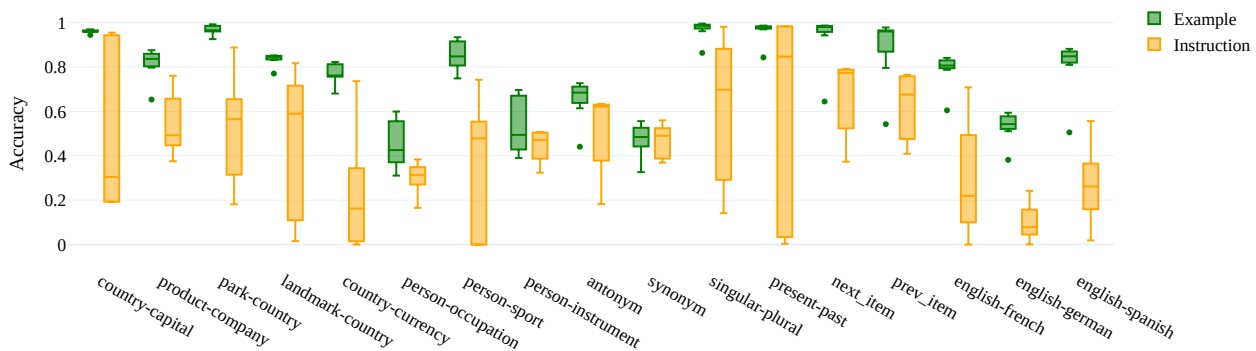

*Figure 17.* Behavior variance across all tasks for Qwen2.5-32B-Instruct model.

## B.4. Qwen-3 Models

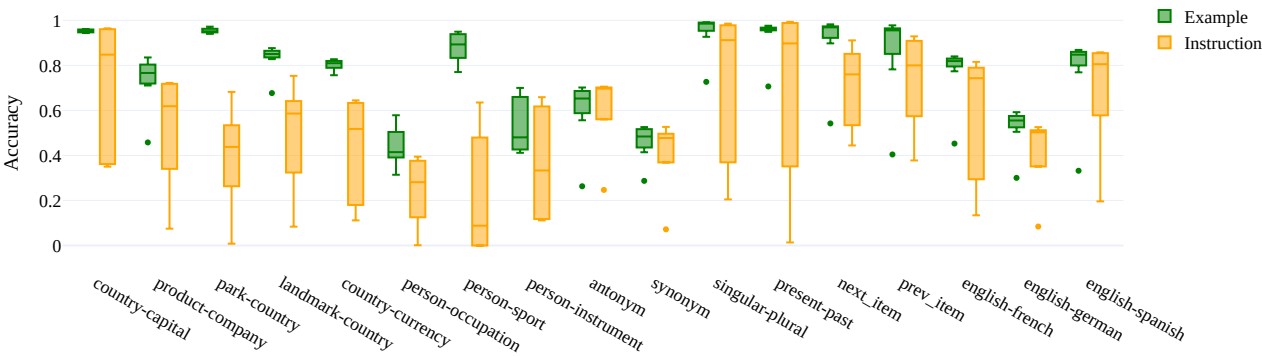

*Figure 18.* Behavior variance across all tasks for Qwen3-30B-A3B-Instruct-2507 model. Each data point represents the accuracy of a specific prompt template for a given task and prompting style.

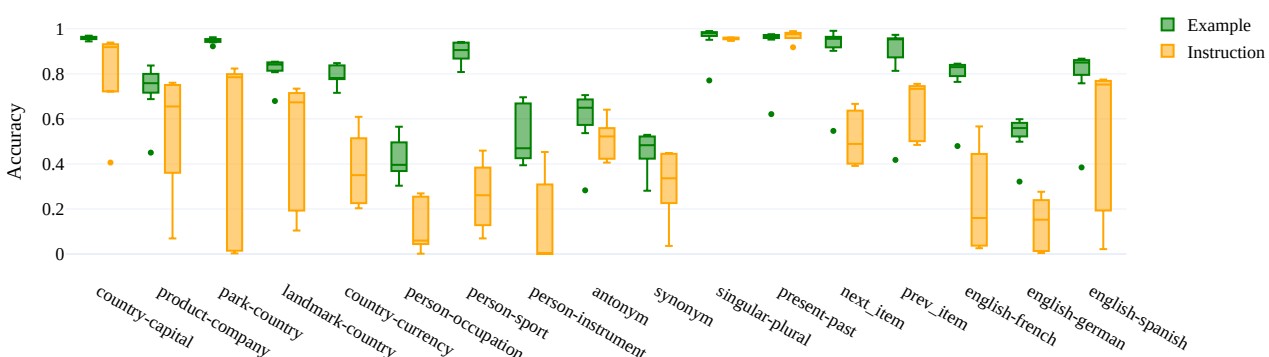

*Figure 19.* Behavior variance across all tasks for Qwen3-30B-A3B-Thinking-2507 model. Each data point represents the accuracy of a specific prompt template for a given task and prompting style.

# C. Characterizing Components in Vocabulary Space

## C.1. Lexical Task Heads

In this section, we provide examples of the early decoded outputs of the lexical task heads. The attention heads are early decoded to the vocabulary space using Logit Lens (Nostalgebraist, 2020).

| Prompt | Head Index | Top Decoded Vocab |
|--------|-----------|-------------------|
| What company makes the Game Boy Advance? It's ___ | (L14, H13) | _Quiz, _company, itors, _ce, _URI |
| | (L17, H4) | _companies, _organisation, _organization, _publishers, |
| | (L18, H24) | _company, _compan, _firm, _Company, 公司(company) |
| | (L22, H28) | _companies, companies, Companies, _Companies, _firms |
| | (L22, H29) | _game, game, _company, _Game, Game |
| Acura TL: Honda; Lancia 037: Fiat; Windows Server 2003: Microsoft; DualShock: | (L22, H10) | _design, _weld, _manufacturing, _weld, _manufacture |
| | (L22, H29) | _company, company, _company, _molecule, _company |
| | (L21, H00) | _company, Company, company, _Company, _brand |
| | (L21, H30) | Firm, _firm, _Firm, _Firm, _usual |
| | (L17, H04) | _organisation, morgan, _organization, _publisher, _publishers |

*Table 6.* Decoded attention head outputs on product-producer prompts

| Prompt | Head Index | Top Decoded Vocab |
|--------|-----------|-------------------|
| The antonym of push is ___ | (L13, H22) | _opposite, op, _reverse, ful, _противоп (opposite) |
| | (L17, H20) | _opposite, _contrario, _ngược (reverse), _contrary, |
| | (L21, H29) | _terms, _term, _opposite, '_terms', terms |
| | (L25, H12) | _Ant, _ant, Ant, ant, _ANT |
| | (L26, H29) | _ant, _Ant, Ant, _ANT, '_ant' |
| narrow: wide; avoid: confront; revoke: grant; push: | (L13, H21) | _opposite, _trade, kehr, xies, _oppos |
| | (L13, H22) | _reverse, _opposite, _ngược (reverse), 逆(reverse), ipa |
| | (L31, H17) | _wrong, _such, _that, _same, _Wrong |
| | (L25, H14) | _Rev, _rev, Rev, rev, .rev, revoke, |
| | (L20, H26) | _ukkit, _kre, _opp, _omain, _nze |

*Table 7.* Decoding attention head outputs on antonym prompts

| Prompt | Head Index | Top Decoded Vocab |
|--------|-----------|-------------------|
| A synonym for artistic is ___ | (L28, H09) | _itself, _similar, thing, _closely, _different |
| | (L15, H19) | _max, _equival, _correct, lik, _fatt |
| | (L17, H20) | adi, _equivalent, _equival, agram, essentially |
| | (L28, H09) | _itself, _similar, thing, _closely, _different |
| | (L00, H30) | _, _Stuart, _equiv, _Oaks |
| mediocre: average; artistic: creative; exhaustive: comprehensive; loose: unfastened; counsel: advise; humble: | (L06, H02) | _é, _synonym, kop, atis, cep |
| | (L13, H06) | engo, CCC, zsche, _equivalent, suz |
| | (L20, H21) | _equally, _big, _similarly, _BIG, -big, high, near |
| | (L24, H15) | _alike, _elong, 569, 812, 846, ooke, anz |
| | (L24, H26) | .Reporting, —</, .same, . \n\n\n\n\n\n\n, .partial |

*Table 8.* Decoded attention head outputs on synonym prompts

| Prompt | Head Index | Top Decoded Vocab |
|---|---|---|
| The plural of mouse is ___ | (L18, H8) | _multiple, _Multiple, multiple, _, _single |
| | (L19, H21) | Pl, _Pl, _Pla, pl, PL |
| | (L20, H23) | _several, _is, _of, _numerous, Several |
| | (L22, H24) | _fewer, _počtu (number), літ (years), _number, _plural |
| | (L28, H13) | _pl, _Pl, Pl, -pl, pl |
| dog: dogs; shoe: shoes; mouse: | (L13, H22) | '_multiple', _Multiple, Multiple, extended, multiple |
| | (L18, H8) | _trio, _dozen, _multiple, tec, _multiples |
| | (L27, H28) | ..., _..., _ironically, .. |
| | (L21, H16) | _abra, _oval, _iliz, , _Multiplicity |
| | (L22, H13) | _multiple, _multiple, _Multiple, _Multiple, _Monday |

*Table 9.* Decoded attention head outputs on singular-plural prompts

| Prompt | Head Index | Top Decoded Vocab |
|---|---|---|
| The capital city of Japan is ___ | (L16, H25) | _cities, _Cities, 城市(city), _towns, cities |
| | (L17, H3) | _city, -city, _urban, city, _City |
| | (L23, H6) | _capital, _Capital, capital, _CAPITAL, Capital |
| | (L24, H20) | _city, _City, city, City, -city |
| | (L29, H31) | _city, _budget, _culture, _government, _travel |
| Ecuador: Quito; England: London; Korea: Seoul; Japan: | (L15, H21) | _location, _locations, location, _located, _Location |
| | (L16, H25) | _cities, _Cities, _towns, cities, 城市(city) |
| | (L17, H4) | _cities, _town, _towns, cities, _Cities |
| | (L22, H28) | _cities, _Locations, _Cities, Cities, _lands |
| | (L29, H31) | _travel, _tour, _tourist, _tours, _tourism |

*Table 10.* Decoding attention head outputs on country-capital prompts

| Prompt | Head Index | Top Decoded Vocab |
|---|---|---|
| The country where Band-e Amir National Park is located is ___ | (L08, H19) | _country, _foreign, alink, IFn, _concentr |
| | (L16, H25) | countries, Countries, 国家(country), nations |
| | (L17, H04) | _countries, countries, _country, _Countries, _países |
| | (L20, H01) | _countries, ř, _.Dictionary, _.country, Country |
| Hirkan National Park: Azerbaijan; Altyaghach National Park: Azerbaijan; Absheron National Park: Azerbaijan; Gizilagaj National Park: | (L16, H25) | _countries, avel, ritt, oot, 国の(country) |
| | (L23, H25) | _national, _National, national, _NATIONAL, _nacional |
| | (L24, H23) | _National, National, _NATIONAL, _Aren, _Avatar |
| | (L30, H29) | _national, _National, national, National, _NATIONAL |
| | (L23, H25) | federal, national, nation, Ethnic, ethnic, Federal, Republic |

*Table 11.* Decoded attention head outputs on park-country prompts

| Prompt | Head Index | Top Decoded Vocab |
|---|---|---|
| The next item after seventeen is | (L14, H12) | _next, ieve, クルゼイロ (Cruzeiro), StandardItem, _bio |
| | (L16, H04) | _upcoming, esterday, _next, afx, _previously |
| | (L28, H25) | following, _following, Following, _Following, antor |
| | (L28, H24) | _after, after, _After, After, again |
| | (L28, H27) | _after, _AFTER, after, -after, After |
| December: January; zero: one; one: two; two: " | (L15, H22) | 第 (No. / -th), _nth, cee, tar, _next |
| | (L17, H21) | _soon, _Soon, _next, soon, oro |
| | (L31, H17) | ses, _wrong, _same, _who, _following |
| | (L20, H29) | _next, _tomorrow, next, _NEXT, (next |
| | (L28, H24) | once, Once, once, _once, .once, 一次 (once) |

*Table 12.* Decoded attention head outputs on next-item prompts

| Prompt | Head Index | Top Decoded Vocab |
|---|---|---|
| Tell me what comes before 'S'. It's | (L17, H21) | madan, preced, дат, rok, _before |
| | (L20, H29) | emarks, BeforeEach, _after, _preced, HeaderCode |
| | (L21, H28) | _before, before, _Before, Before, .before |
| | (L28, H09) | _next, _latest, _first, _which, _recent |
| | (L28, H10) | _something, _nothing, something, nothing, _what |
| xx: xix; tuesday: monday; wednesday: tuesday; thursday: wednesday; friday: thursday; saturday: | (L20, H29) | _yesterday, _Yesterday, 昨 (previous), _previous, _Yesterday |
| | (L15, H07) | dit, akhir, 締 (tighten), _odd, last |
| | (L24, H07) | ista, _t, aoke, _last, [... |
| | (L24, H10) | _inclusive, inclusive, _inclus, clusive, early |
| | (L26, H21) | before, _before, _antes, _Before, Before |

*Table 13.* Decoded attention head outputs on previous-item prompts

| Prompt | Head Index | Top Decoded Vocab |
|---|---|---|
| The French translation of dependent is | (L14, H24) | France, French, 法国 (France), французз (French), France |
| | (L17, H23) | French, French, français (French), french, французский (French) |
| | (L24, H03) | _French, _France, French, _french, France |
| | (L27, H20) | _French, French, _french, _французз (French), _francouz |
| | (L07, H01) | onas, _Frances, aina, umat, ionales |
| guess: deviner; elaborate: élaborer; poverty: pauvreté; opens: ouvre; dependent: dépendant; those: | (L14, H24) | _French, France, French, `, французский (French) |
| | (L24, H03) | _French, French, _France, french, ö |
| | (L31, H07) | _French, _France, _french, French, 法国 (France) |
| | (L17, H15) | _France, . . . \n, _Paris, _&, _Bureau |
| | (L04, H30) | _cru, ierz, grass, etler, ※ |

*Table 14.* Decoded attention head outputs on english-french prompts

| Prompt | Head Index | Top Decoded Vocab |
|---|---|---|
| The German translation of penalties is ___ | (L14, H24) | _Germany, _German, German, _кеамс, _german |
| | (L19, H25) | _GmbH, hausen, バイ (by), _Germany, itz |
| | (L27, H20) | _German, German, _Germans, _german, _Germany |
| | (L24, H03) | German, _Germany, _German, _Germans, _Germany |
| | (L27, H20) | German, _German, _Germans, _german, _Germany |
| court: Gericht; fishing: fischen; pay: zahlen; raid: üfcberfall; bird: vogel; much: | (L14, H24) | ō, German, Germans, German, options |
| | (L24, H03) | _Germany, _German, Germany, German, _Germans |
| | (L27, H20) | _German, German, _Germans, _German, _Mexican |
| | (L31, H07) | _Hamburg, _Germans, _German, German, _Germany |
| | (L17, H23) | acher, Swiss, European, sw, urally, CTR |

*Table 15.* Decoded attention head outputs on english-german prompts

| Prompt | Head Index | Top Decoded Vocab |
|---|---|---|
| The Spanish translation of not is ___ | (L17, H23) | Spanish, _untranslated, _translated, _translator, Spanish |
| | (L21, H01) | _Spanish, _French, _German, _Portuguese, _Greek |
| | (L24, H03) | _Spanish, _Spain, _Span, Spanish, Spain |
| | (L27, H20) | _Spanish, Spanish, _spanish, _Japanese, _German |
| | (L31, H07) | _Spanish, Spanish, _English, _Latin, _Hispanic |
| journal: diario; rangers: guardabosques; girl: chica; opponent: oponente; marks: marcas; not: | (L17, H23) | _Spanish, ugu, _equ, orida, Spanish |
| | (L24, H03) | _Spanish, Spanish, _Spain, _spanish, Spain |
| | (L31, H07) | _Spanish, _spanish, Spanish, _Buenos, _Hispan |
| | (L07, H09) | latin, _, ppo, dek, POCH |
| | (L16, H19) | uben, _blue, umat, zik, veau |

*Table 16.* Decoded attention head outputs on english-spanish prompts

| Prompt | Head Index | Top Decoded Vocab |
|---|---|---|
| The past tense of adapt is ___ | (L21, H28) | past, _past, Past, _Past, _present |
| | (L25, H07) | _Past, _past, Past, past, .past |
| | (L31, H21) | _past, _Past, Past, _Pass, _pass |
| | (L17, H26) | _starting, _beginning, _initial, _early, Starting |
| | (L04, H26) | awa, ondo, imeter, _early, im |
| work: worked; worry: worried; write: wrote; yield: yielded; zoom: zoomed; accelerate: | (L14, H05) | 旧(old), _Past, _past, _old, old |
| | (L17, H21) | 曾(past), ennon, _inevitable, _past, _previously |
| | (L20, H29) | _during, _throughout, _earlier, _prior, during |
| | (L28, H24) | be, actually, exactly, around, generally |
| | (L02, H14) | eken, bsite, -web, ži, pert, |

*Table 17.* Decoded attention head outputs on present-past prompts

| Prompt | Head Index | Top Decoded Vocab |
|---|---|---|
| The occupation of David Yost is a/an ___ | (L22, H11) | _roles, _positions, _position, _careers, _jobs |
| | (L22, H28) | _jobs, _occupations, _lives, istributions, ocations |
| | (L22, H29) | _person, _occupation, _operation, .opts, _activity |
| | (L23, H05) | _occupation, _Occup, _occup, _Occupation, _Occup |
| | (L17, H04) | _profession, _roles, bart, 348, _lic |
| Joan Leslie: actor; Ludvig Holberg: philosopher; Subair: actor; Billy Roche: actor; Arun Nehru: politician; Gregg Edelman: | (L22, H11) | talen, 员(member), role, roles, iras, talents |
| | (L16, H27) | _career, _career, _himself, vere, career |
| | (L24, H18) | _activity, _behavior, _Activity, _affect, _Behavior |
| | (L28, H23) | _action, _actions, _activity, _Action, _ACTION |
| | (L15 H23) | imity, Knot, adesh, 道(path), kke |

*Table 18.* Decoded attention head outputs on person-occupation prompts

| Prompt | Head Index | Top Decoded Vocab |
|---|---|---|
| The sport Roberto Clemente plays is ___ | (L22, H11) | _games, _game, _role, _roles, games |
| | (L22, H28) | _plays, _games, _Plays, _Games, Games |
| | (L29, H31) | _sports, _sport, _play, _playing, _games |
| | (L22, H29) | _sport, _Sport, sport, Sport, alars |
| | (L23, H08) | ě, _Sports, ESP, _Matt, _Greg |
| Joe Garagiola Sr.: baseball; Pavel Bure: hockey; Ernie Barnes: football; Peyton Manning: football; Megan Rapinoe: soccer; Andreas Ivanschitz: | (L17, H03) | _game, _play, _performance, _sports, _playing |
| | (L22, H11) | _games, _game, _sport, game, games |
| | (L17, H16) | ball, Extr, sports, playing, (passage) |
| | (L18, H17) | _sports, _covering, dration, _bubble, _sports |
| | (L21, H15) | _sports, _Sports, _football, _Sport, Sports |

*Table 19.* Decoded attention head outputs on person-sport prompts

| Prompt | Head Index | Top Decoded Vocab |
|---|---|---|
| The instrument Tom Fletcher plays is the ___ | (L16, H08) | _instrument, _instruments, _dro, _dow, _Robin |
| | (L16, H25) | azzi, _Seeder, 陵(tomb), _instruments, _pillar |
| | (L17, H04) | _instruments, _Instruments, _tools, _instrument, $\alpha\rho\acute{\alpha}$ |
| | (L18, H18) | DMI, _instruments, _Instruments, _Instrument, _instruments |
| | (L21, H14) | _guitar, _guitars, _instrument, _instruments, _Instruments |
| Russell Gunn: trumpet; Brian May: guitar; Yfrah Neaman: violin; Vlatko Stefanovski: guitar; Yvonne Hubert: piano; Friedrich Gulda: | (L22, H11) | _instruments, _instrument, _keyboards, _roles, instrument |
| | (L29, H30) | _instrument, Instrument, _track, _instruments, _instrument |
| | (L29, H31) | _piano, _music, _instrument, _musical, _compositions |
| | (L31, H03) | _Piano, _Guitar, _Composer, _Keyboard, _Audio |
| | (L17, H23) | _Alps, _Charlottesville, _Austria, _Austrian, _Vienna |

*Table 20.* Decoded attention head outputs on person-instrument prompts

| Prompt | Head Index | Top Decoded Vocab |
|---|---|---|
| The currency of Algeria is the ___ | (L15, H26) | _Terminal, _ATM, _money, _decimal, _currency |
| | (L22, H11) | hccurrency, _currencies, currency, _money, _Currency |
| | (L23, H22) | money, _Money, 錢 (currency), Money, _money |
| | (L29, H31) | _currency, _currencies, _money, _exchange, _cash |
| | (L21, H14) | _coin, _coins, _Coin, Coin, _gold |
| Azerbaijan: Manat; Bahamas: Bahamian Dollar; Bahrain: Bahraini Dinar (BHD); Bangladesh: Taka; Barbados: Barbadian Dollar (BBD); Belarus: Belarusian Ruble (BYN); Belgium: Euro (EUR); Belize: Belize Dollar (BZD); Benin: CFA Franc (XOF); Bhutan: Ngultrum (BTN); Bolivia: | (L17, H03) | _currency, _foreign, currency, _money, foreign |
| | (L22, H11) | _currency, _currencies, _money, currency, _Currency |
| | (L23, H22) | 錢 (currency), money, _Money, Money, _money |
| | (L17, H04) | currencies, currency, ö, Languages, .dtype |
| | (L22, H27) | _currency, _currency, _coin, currency, _currencies |

*Table 21.* Decoded attention head outputs on country-currency prompts

| Prompt | Head Index | Top Decoded Vocab |
|---|---|---|
| The country where Autonomous University of Madrid is located is ___ | (L15, H14) | _land, _country, _hom, _LAND, eland |
| | (L17, H15) | /tos, _ABI, country, ´, erdem |
| | (L17, H26) | edback, country, ilian, rika, Пос |
| | (L21, H00) | 它(it), Country, country, its, Country |
| | (L22, H29) | _country, OUNTRY, country, _Country, Country |
| Valdemarsvik: Sweden; Ozumba: Mexico; Piper Verlag: Germany; Attingal: India; Georgians: Azerbaijan; Nizampatnam: | (L21, H00) | _country, _town, country, Country, _Country |
| | (L17, H15) | _country, loven, ibar, _subplot, cta, 984, Chow |
| | (L29, H30) | _country, _Country, Country, country, _book |
| | (L16, H24) | _statewide, _Nationwide, _nationwide, arella, -local |
| | (L16, H25) | _countries, avel, _nations, icles, _places |

*Table 22.* Decoded attention head outputs on landmark-country prompts

# D. Quantify Lexical Task Heads

## D.1. Number of Lexical Task Heads

For each task we collect a set of heads separately for each prompting style—i.e., separately for the example-based prompts and the instruction-based prompts.

We quantify the number of lexical task heads that consistently represent the task in vocabulary space for each prompting style.

### D.1.1. EXAMPLE-BASED PROMPTS

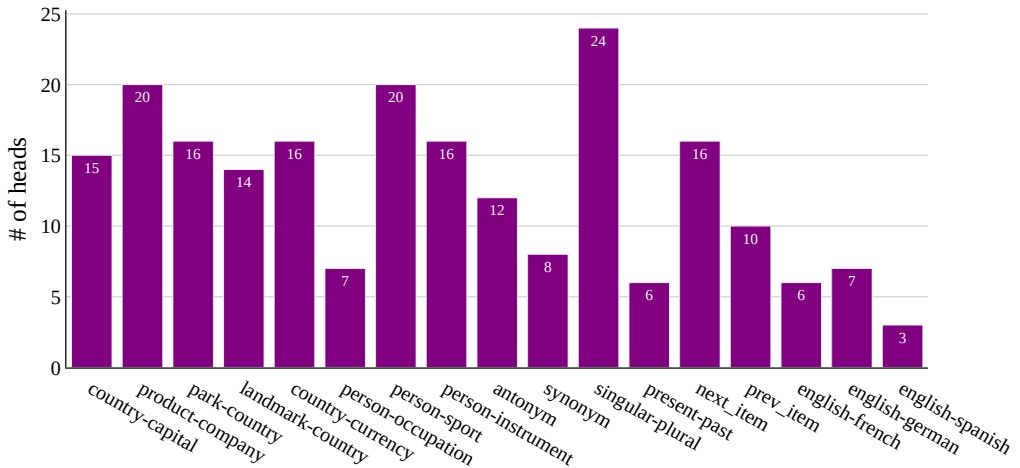

*Figure 20.* Quantification of the number of lexical task heads for example-based prompts in Llama-3.1-8B-Instruct (1,024 total heads)

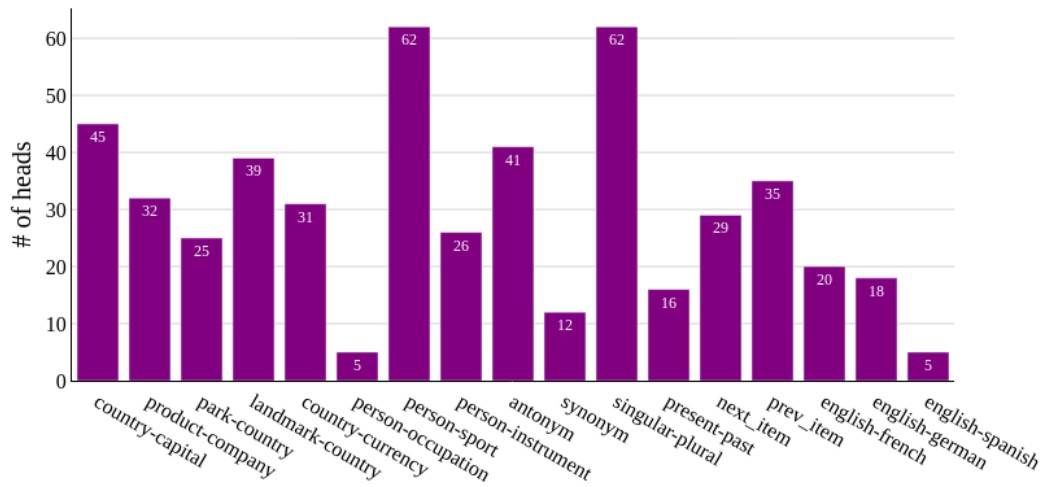

*Figure 21.* Quantification of the number of lexical task heads for example-based prompts in Llama-3.1-70B-Instruct (5,120 total heads)

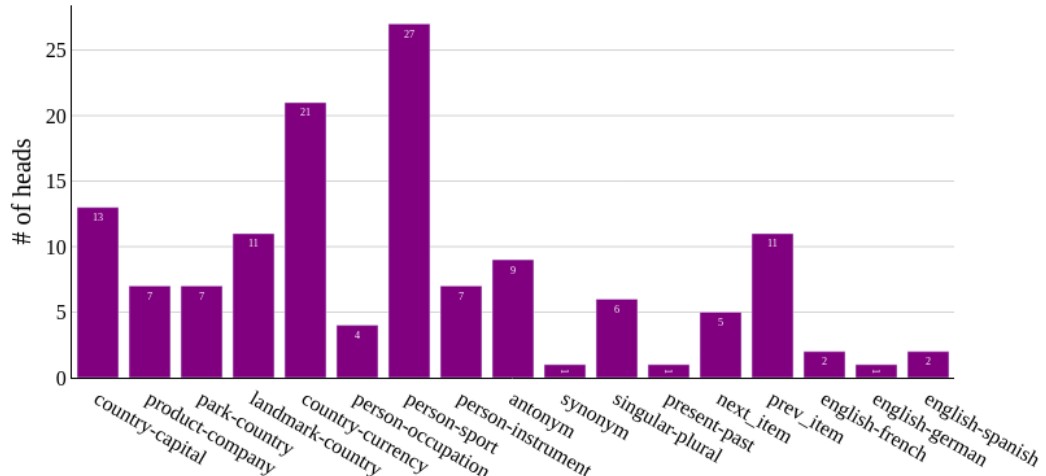

*Figure 22.* Quantification of the number of lexical task heads for example-based prompts
in gemma-2-9b-it (672 total heads)

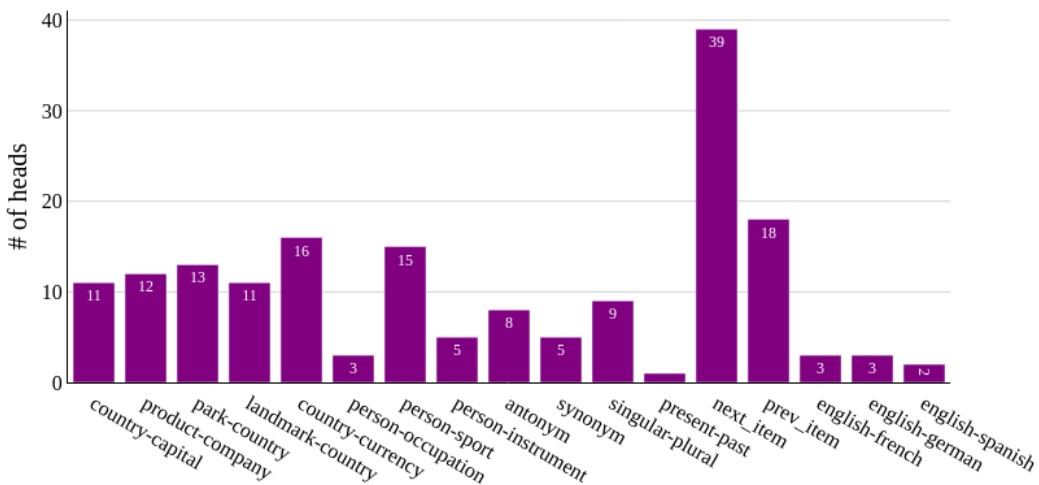

*Figure 23.* Quantification of the number of lexical task heads for example-based prompts
in gemma-2-27b-it (1,472 total heads)

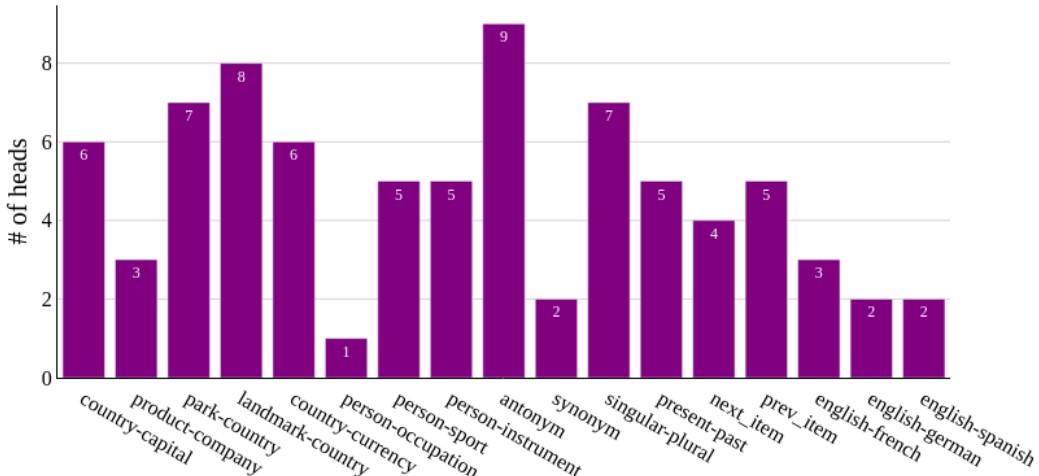

*Figure 24.* Quantification of the number of lexical task heads for example-based prompts
in Qwen2.5-7B-Instruct (784 total heads)

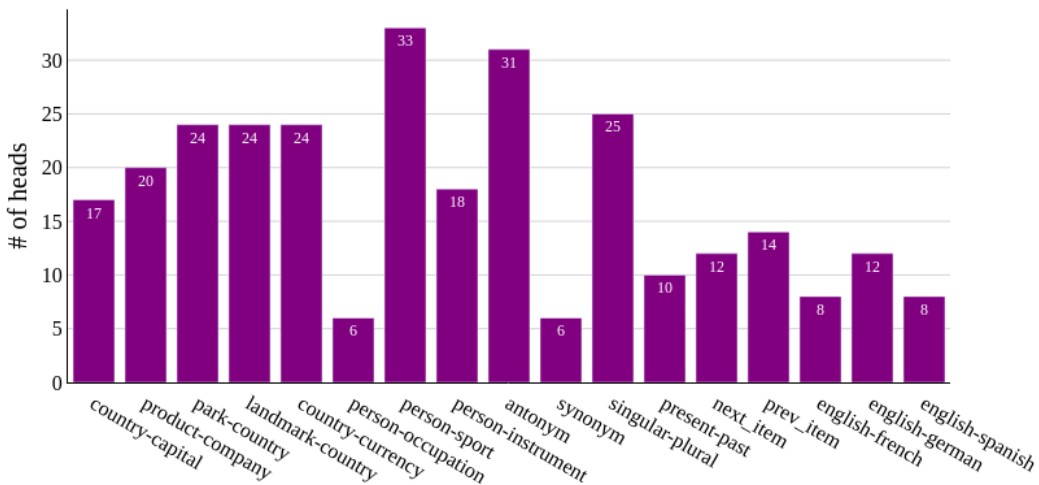

*Figure 25.* Quantification of the number of lexical task heads for example-based prompts
in Qwen2.5-32B-Instruct (2,560 total heads)

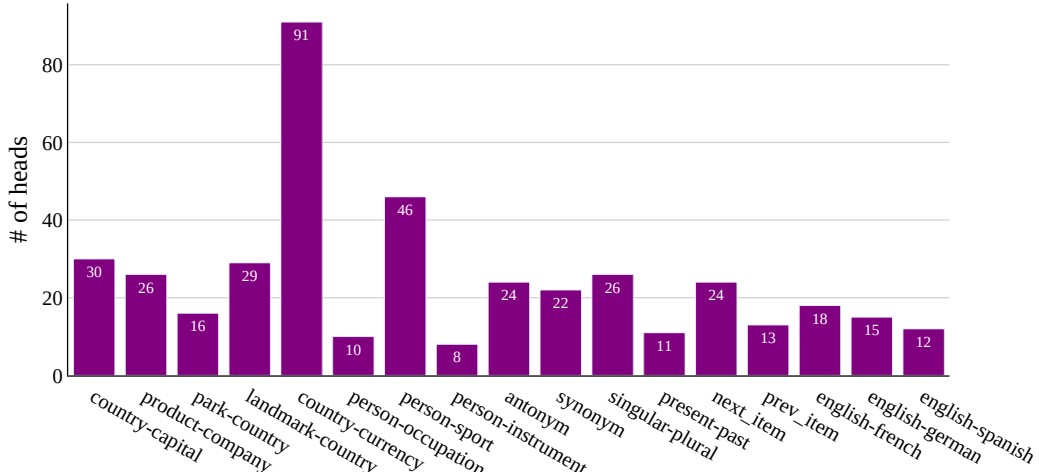

*Figure 26.* Quantification of the number of lexical task heads for example-based prompts in Qwen3-30B-A3B-Instruct model.

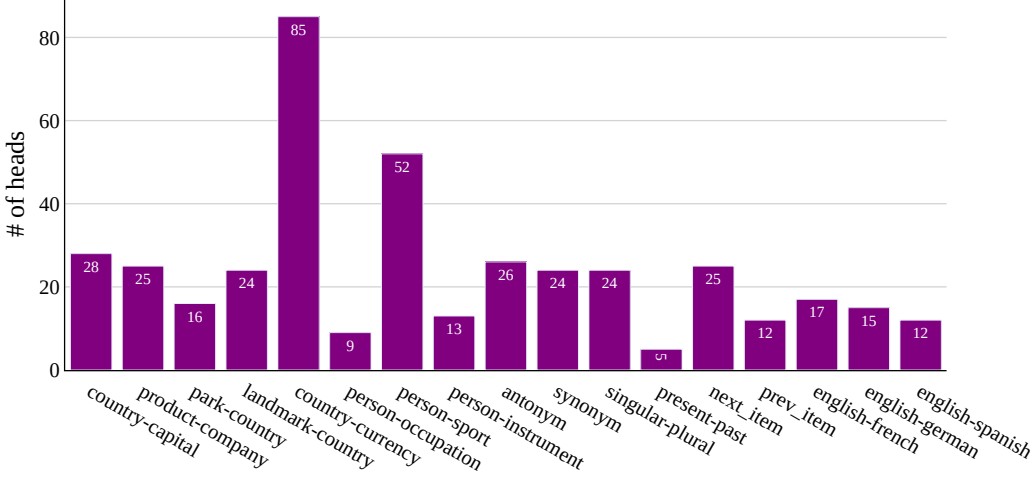

*Figure 27.* Quantification of the number of lexical task heads for example-based prompts in Qwen3-30B-A3B-Thinking model.

### D.1.2. INSTRUCTION-BASED PROMPTS

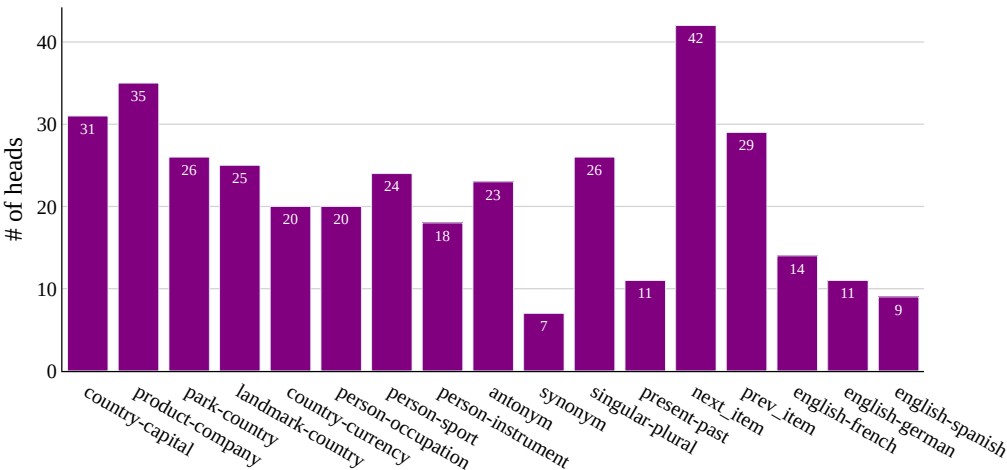

*Figure 28.* Quantification of the number of lexical task heads for instruction-based prompts in Llama-3.1-8B-Instruct (1,024 total heads)

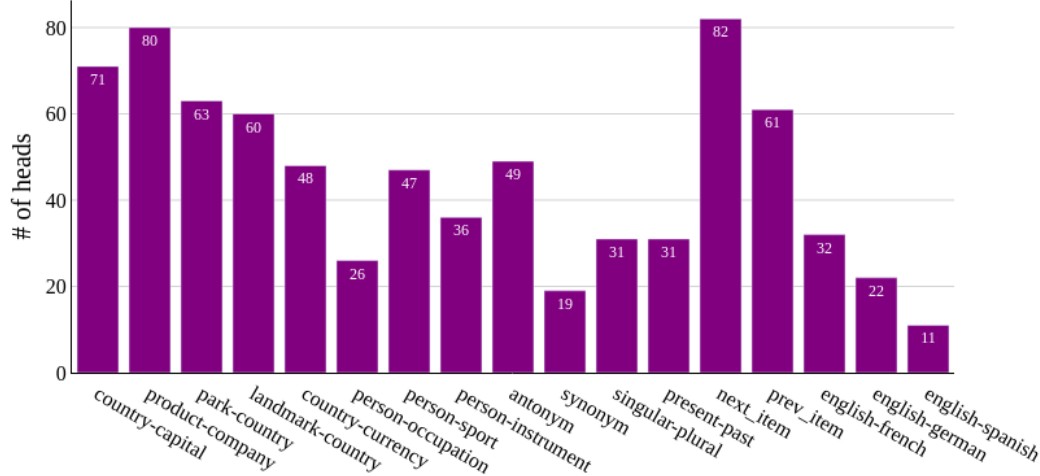

*Figure 29.* Quantification of the number of lexical task heads for instruction-based prompts in Llama-3.1-70B-Instruct (5,120 total heads)

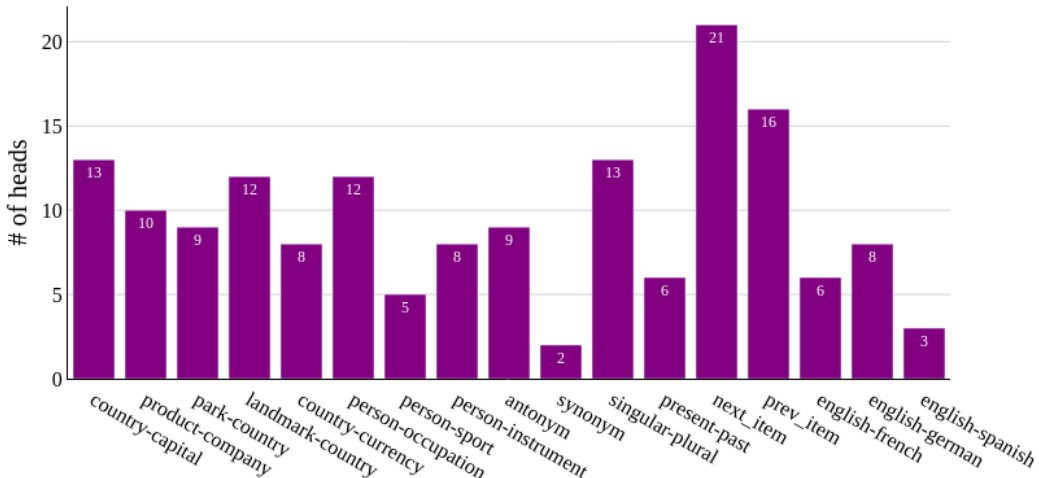

*Figure 30.* Quantification of the number of lexical task heads for instruction-based prompts
in Qwen2.5-7B-Instruct (784 total heads)

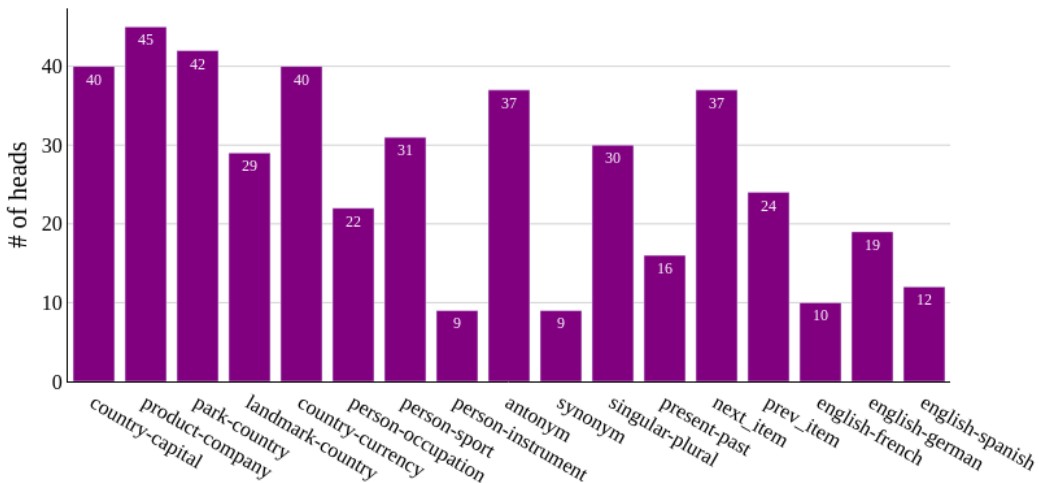

*Figure 31.* Quantification of the number of lexical task heads for instruction-based prompts
in Qwen2.5-32B-Instruct (2,560 total heads)

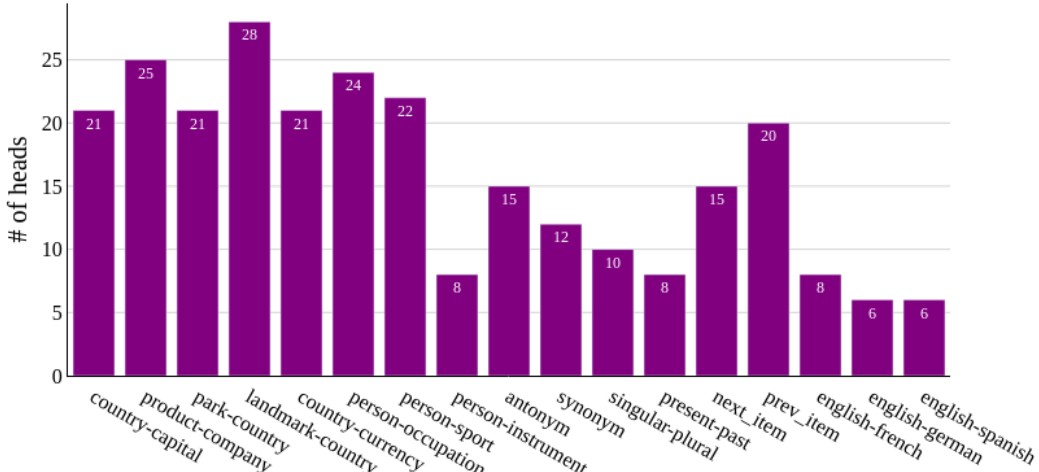

*Figure 32.* Quantification of the number of lexical task heads for instruction-based prompts in gemma-2-9b-it (672 total heads)

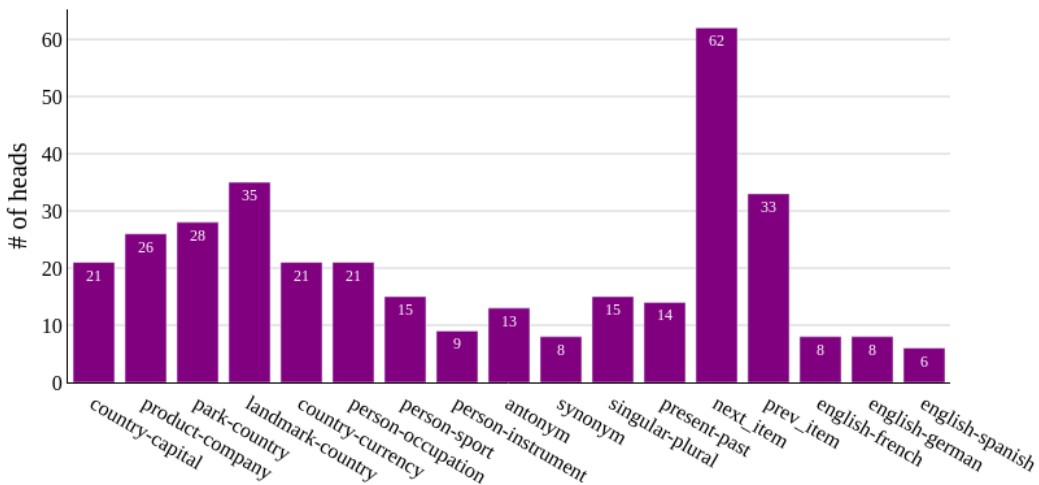

*Figure 33.* Quantification of the number of lexical task heads for instruction-based prompts in gemma-2-27b-it (1,472 total heads)

## D.2. Distribution of heads across layers

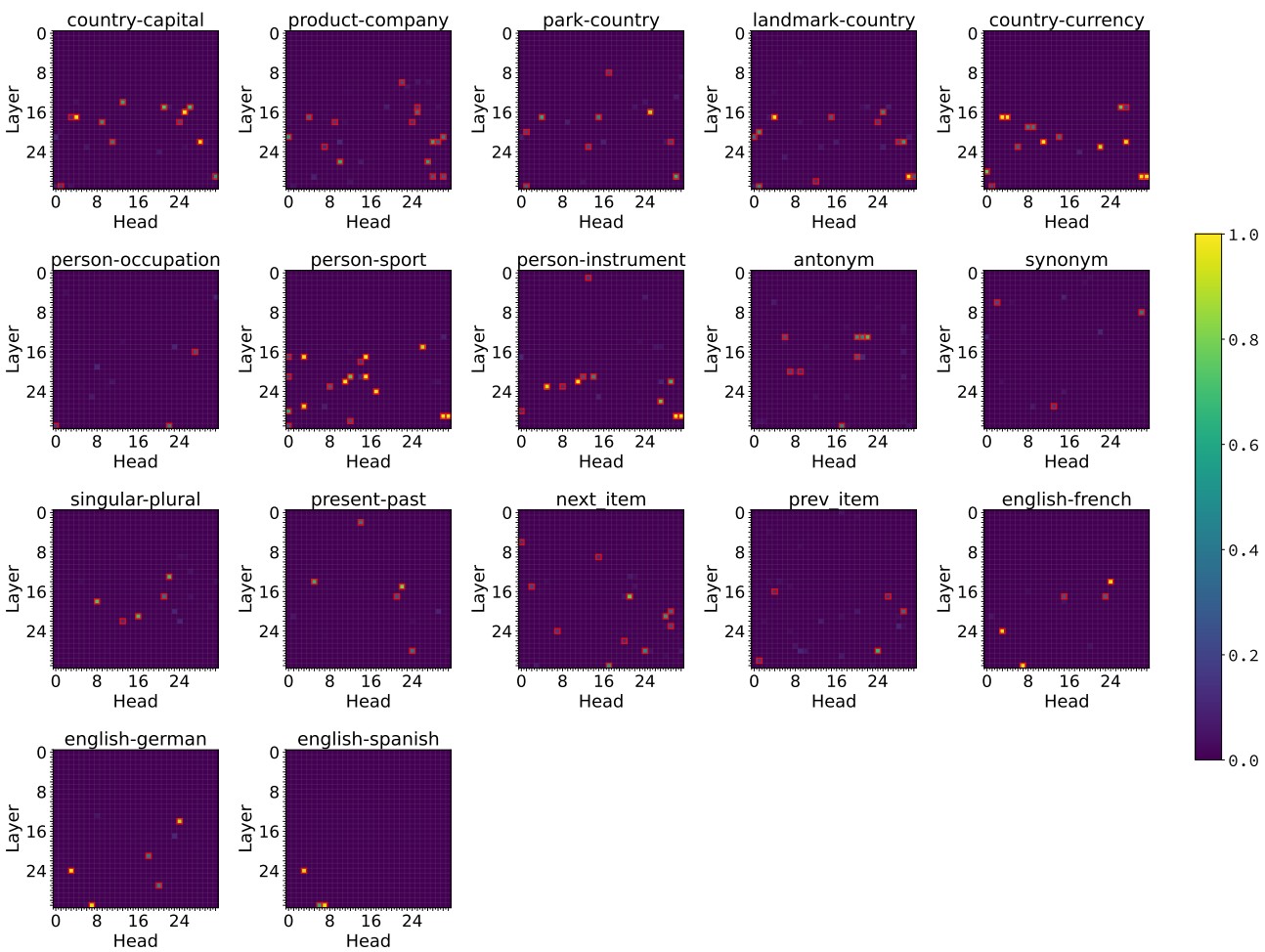

*Figure 34.* Distribution of the lexical task heads across model layers
in Llama-3.1-8B-Instruct

# E. Sensitivity analysis

In this section, we present the sensitivity analyses of parameters $n$ (number of tokens that match a predefined set of task-descriptive terms), $p$ (proportion of prompts), and $k$ (top $k$ decoded tokens) in Llama-3.1-8B model. We first show how the number of heads varies with different values of $n$ & $p$, then $k$.

In each task, as $n$ and $p$ increase, the criteria are more stringent, resulting in less lexical task heads detected for a given prompting style.

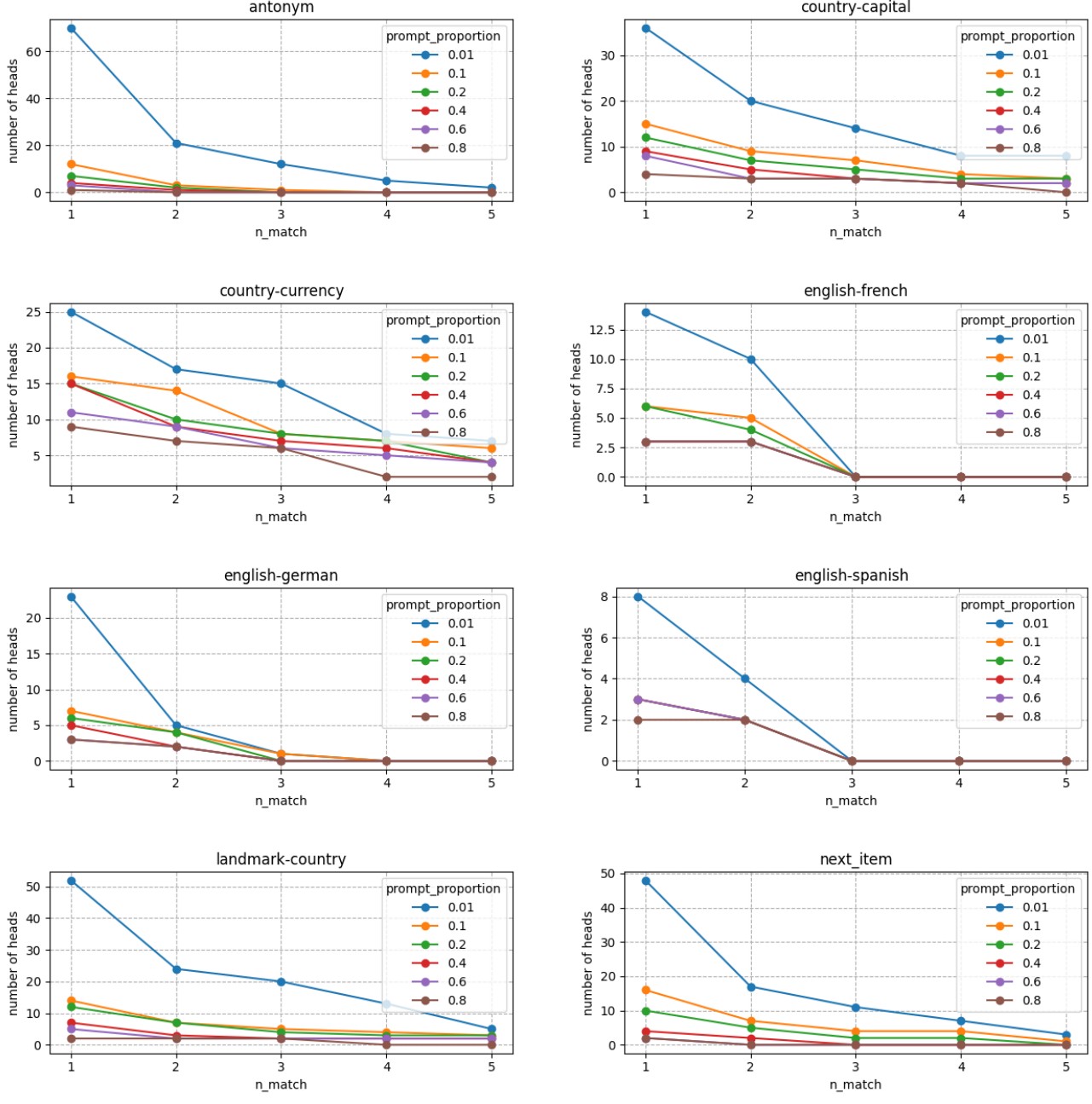

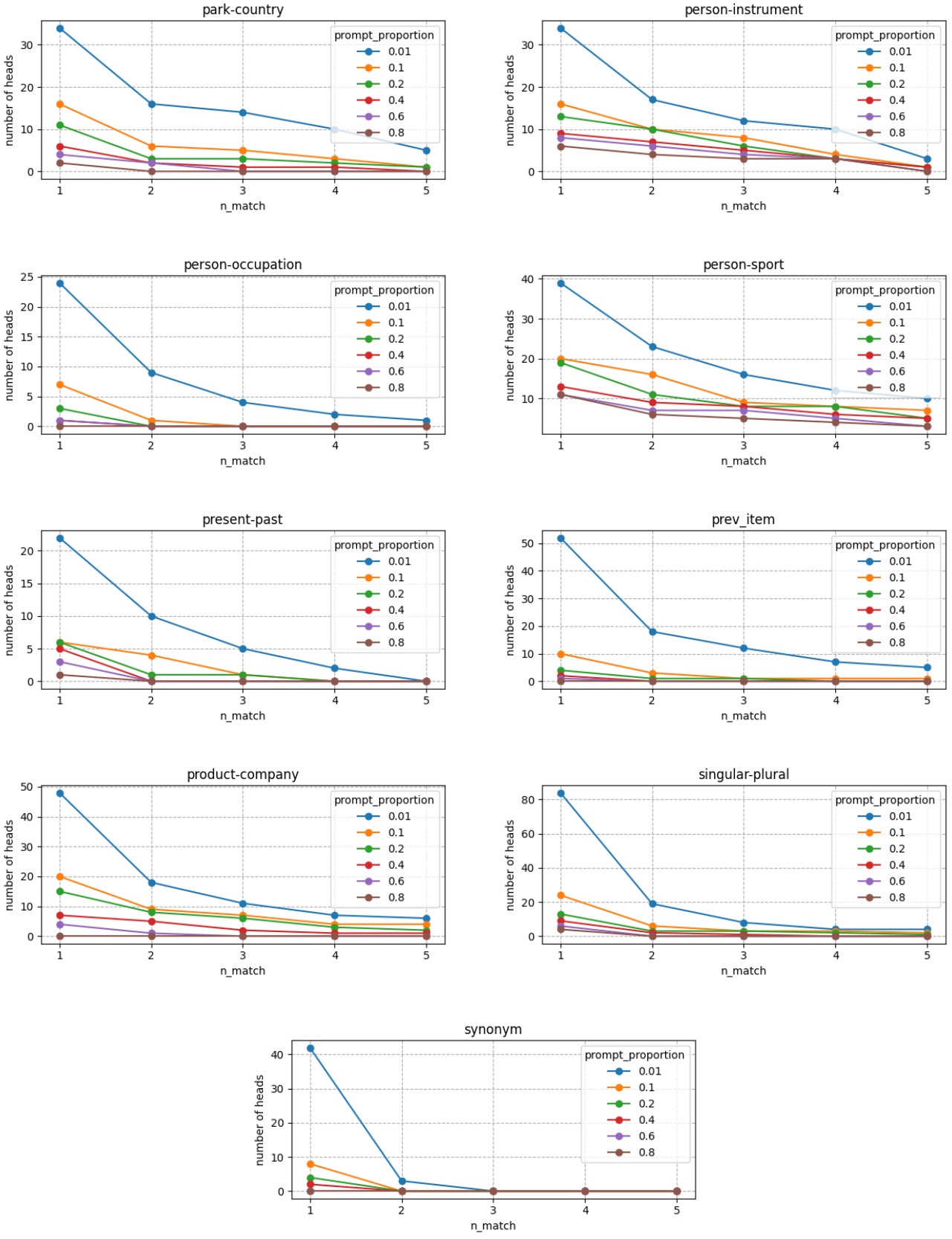

We then show the relationship between the number of heads and $k$. In each task, as $k$ increases, the criteria are less stringent, resulting in more lexical task heads detected for a given prompting style.

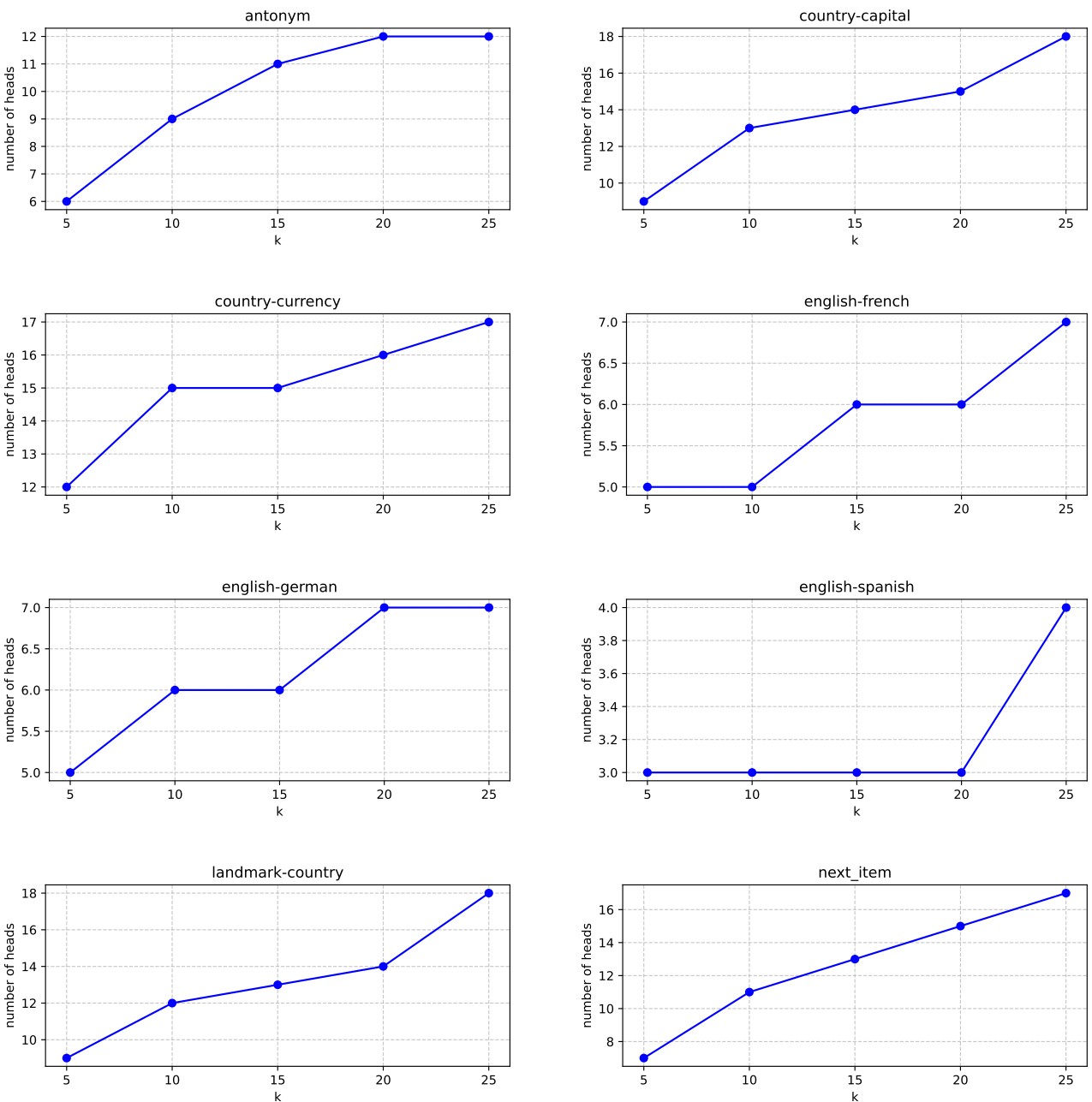

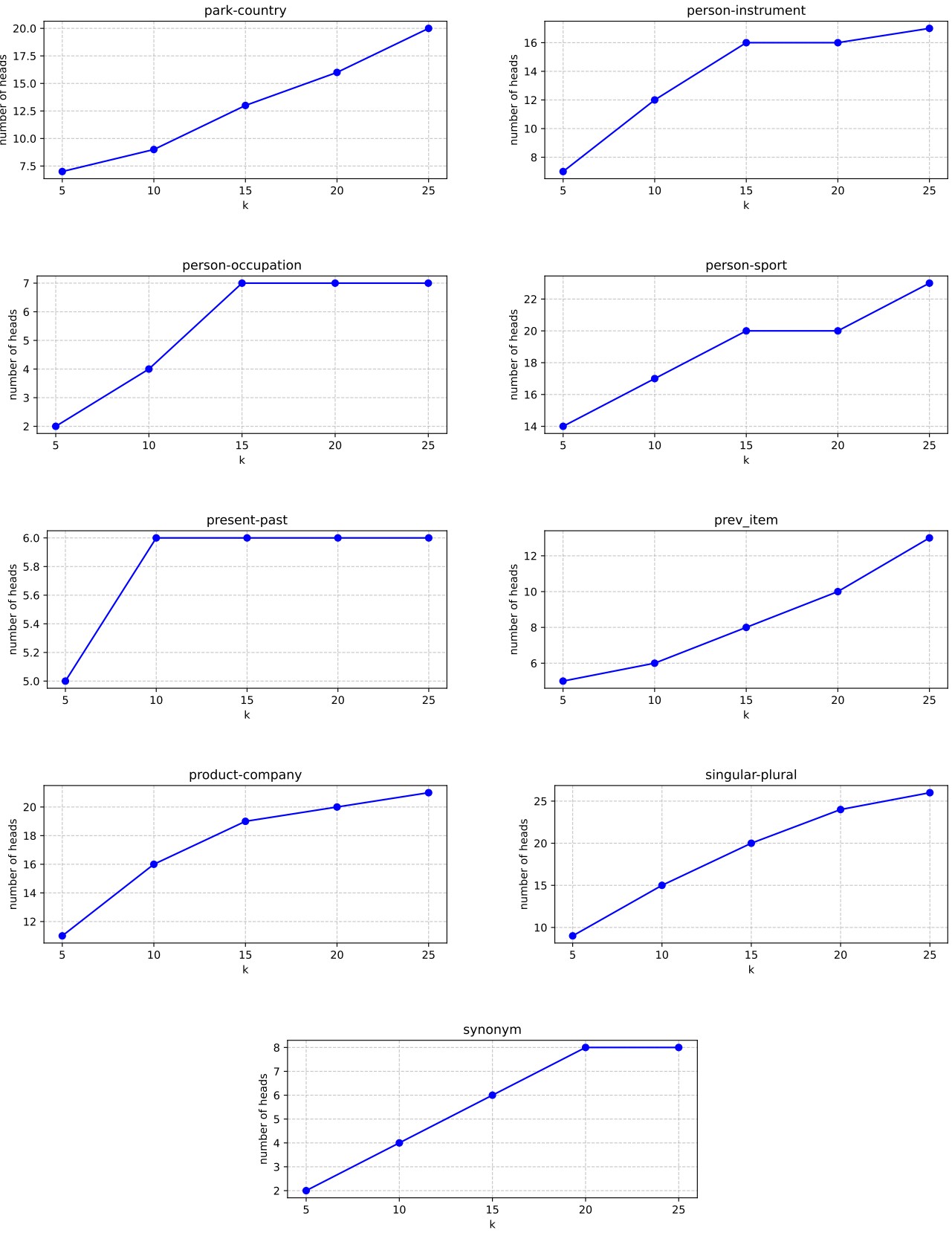

# F. Shared Lexical Task Heads Across Prompting Styles

In this section, we measure the circuit reuse of lexical task heads across prompting styles. For each task (e.g., country-capital, antonym, etc.), we collect a set of heads separately for each prompting style—i.e., separately for the example-based prompts and the instruction-based prompts. We focus only on attention heads that are consistently activated by prompts that produce *correct* answers for a prompting style.

For the heatmaps of each model, X-axis represents tasks prompted with instruction-based prompting and Y-axis represents example-based prompting. A higher color intensity (darker shading) indicates a greater percentage of overlapping heads for a specific task across the two prompting styles.

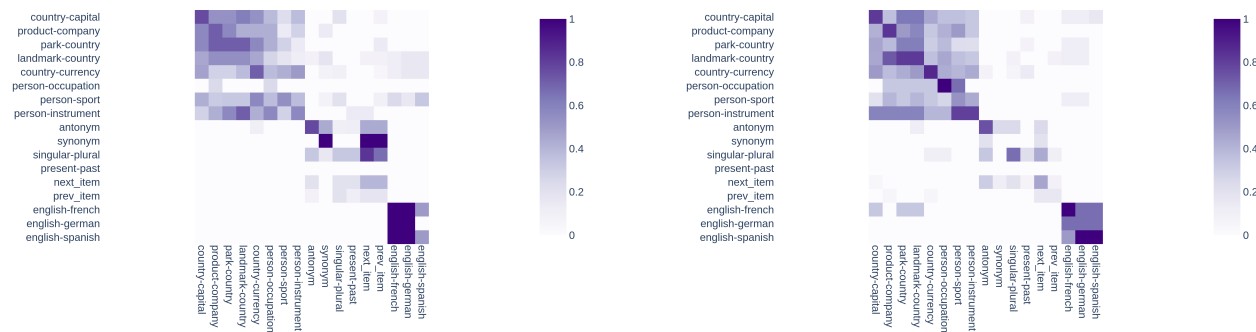

*Figure 51.* Gemma-2-9b-it

*Figure 52.* Gemma-2-27b-it

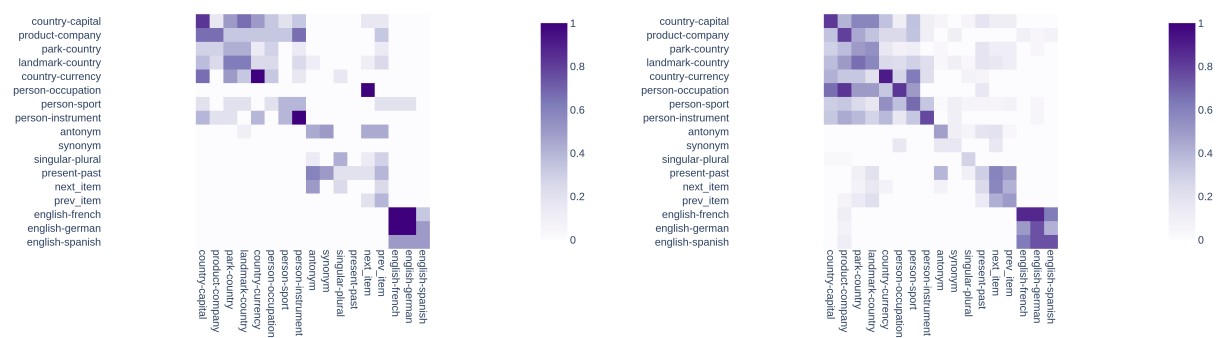

*Figure 53.* Qwen2.5-7B-Instruct

*Figure 54.* Qwen2.5-32B-Instruct

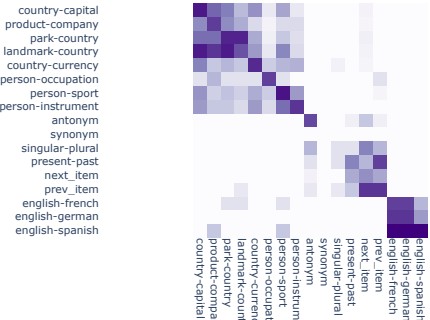

*Figure 55.* Llama-3.1-8B-Instruct

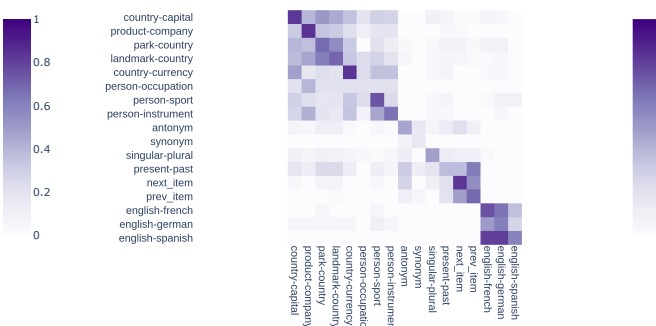

*Figure 56.* Llama-3.1-70B-Instruct

# G. Causal Intervention

An autoregressive transformer language model $f$ takes an input prompt $p_i$ and outputs a next-token prediction $y_i$. The model is composed of multiple components $c$ (MLP neurons or attention heads). In a forward pass, we can cache the output activation at the last token position of each attention head and denote it as $A_c \in \mathbb{R}^d$, where $d$ is the dimension of the residual stream.

Formally, for each task $t \in T$ in a set of tasks $T$, we can do reference runs where we sample a prompt $p_i^t$ from a prompt set of the task $P^t$. For each component $c$, we can cache the activation $A_{c,i}$ (from a reference prompt $p_i^t$) or average activation $\bar{A}_c^t$ (across many prompts of a task $t$). The average activation is calculated as

$$\bar{A}_c^t = \frac{1}{|P^t|} \sum_{p_i^t \in P^t} A_c(p_i^t) \tag{1}$$

We can also do an alternative run and get an output $\tilde{y}_i$ with an alternative prompt $\tilde{p}_i^t \in \tilde{P}^t$, $\tilde{y}_i = f(\tilde{p}_i^t)$. The alternative prompts $\tilde{p}_i^t$ can be corrupted prompts that lack critical information (e.g., task or subject, see Appendix A.3). To measure the causal effect of a set of components, we can patch $A_{c,i}$ or $\bar{A}_c^t$ to an alternative prompt $\tilde{p}_i^t$, and observe the impact of the intervention by comparing $\tilde{y}_i$ and $y_i^{\text{interv}}$.

**Activation Patching:** To measure indirect causal effect of a set of components, we conduct activation patching, where the average activation from reference prompts are patched to each alternative run.

$$y_i^{\text{interv}} = f(\tilde{p}_i^t \mid A_c := \bar{A}_c^t) \tag{2}$$

We use activation patching to determine whether these heads perform similar roles across different prompts. To do this, we activate lexical task heads by patching their activations from one prompt template into another one (see examples in the Table below) and measure the change in task accuracy induced by this patching intervention. We patch to zero-shot prompts so that we can test if lexical task heads can makeup the lack of task information in these prompts.

| Cache *average* activations from | Patch to |
|---|---|
| Example-based prompts: 
 The capital city of Japan is ___ | Japan: ___ |
| Instruction-based prompts: 
 Korea: Seoul; Japan: ___ | Japan: ___ |

*Table 23.* A table of example prompts used for activation patching.

| Cache activations from | Patch to |
|---|---|
| Example-based prompts: 
 The capital city of Japan is ___ | The capital city of [MASK] is ___ |
| Instruction-based prompts: 
 Korea: Seoul; Japan: ___ | Korea: Seoul; [MASK]: ___ |

*Table 24.* Note, we use special tokens like "<|reserved_special_token_1|>" to mask parts of a prompt for llama-3.1-8B model.

## G.1. Lexical Task Heads Create Shared Representations

We use activation patching to determine whether lexical task heads perform similar roles across different prompts. To do this, we activate lexical task heads by patching their activations from one prompt template into another one (see examples in Table 2) and measure the change in task accuracy induced by this patching intervention. We patch to zero-shot prompts so that we can test if lexical task heads can makeup the lack of task information in these prompts.

Below, we separately show the results for intervening on the lexical task heads extracted from instruction-based and example-based prompts.

### G.1.1. ACTIVATION PATCHING ON LEXICAL TASK HEADS EXTRACTED FROM INSTRUCTION-BASED PROMPTS

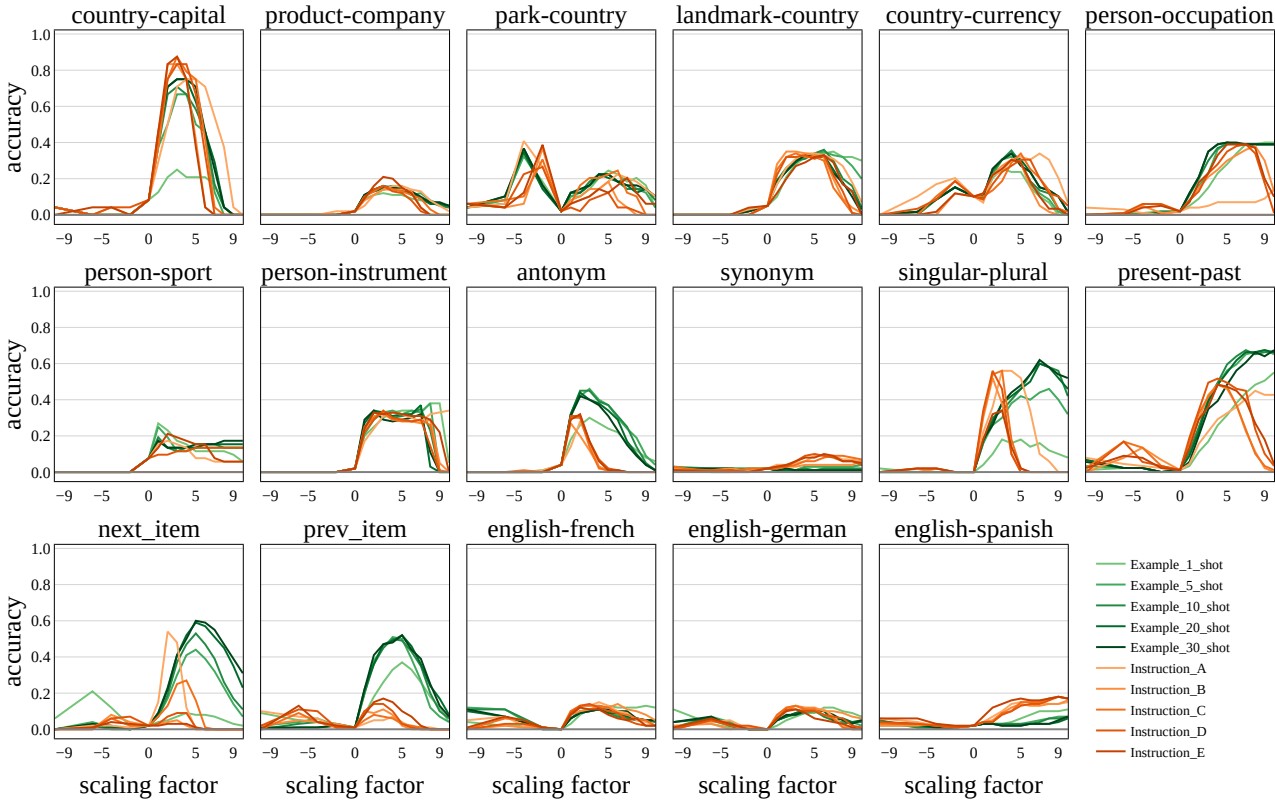

*Figure 57.* Quantification of the causal effect of lexical task heads in the Llama-3.1-8B-Instruct model

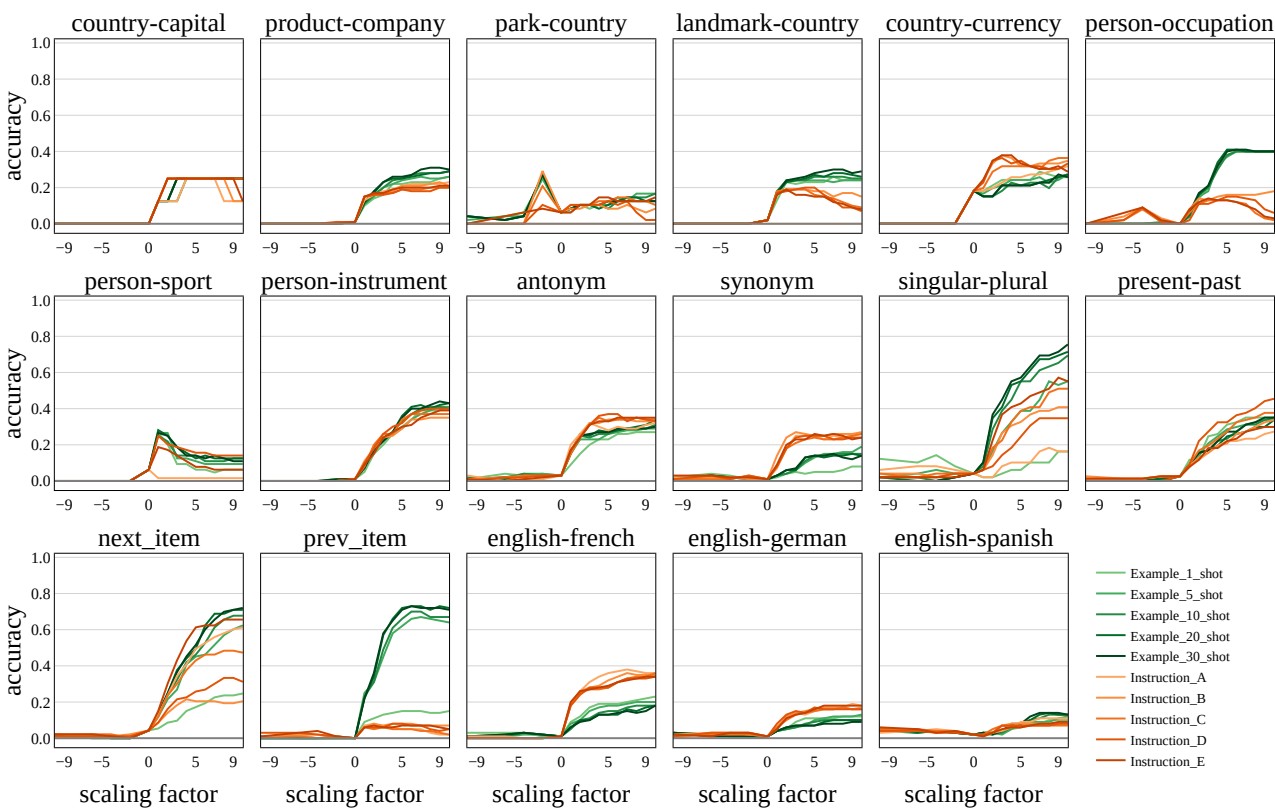

*Figure 58.* Quantification of the causal effect of lexical task heads in the gemma-2-9b-it model

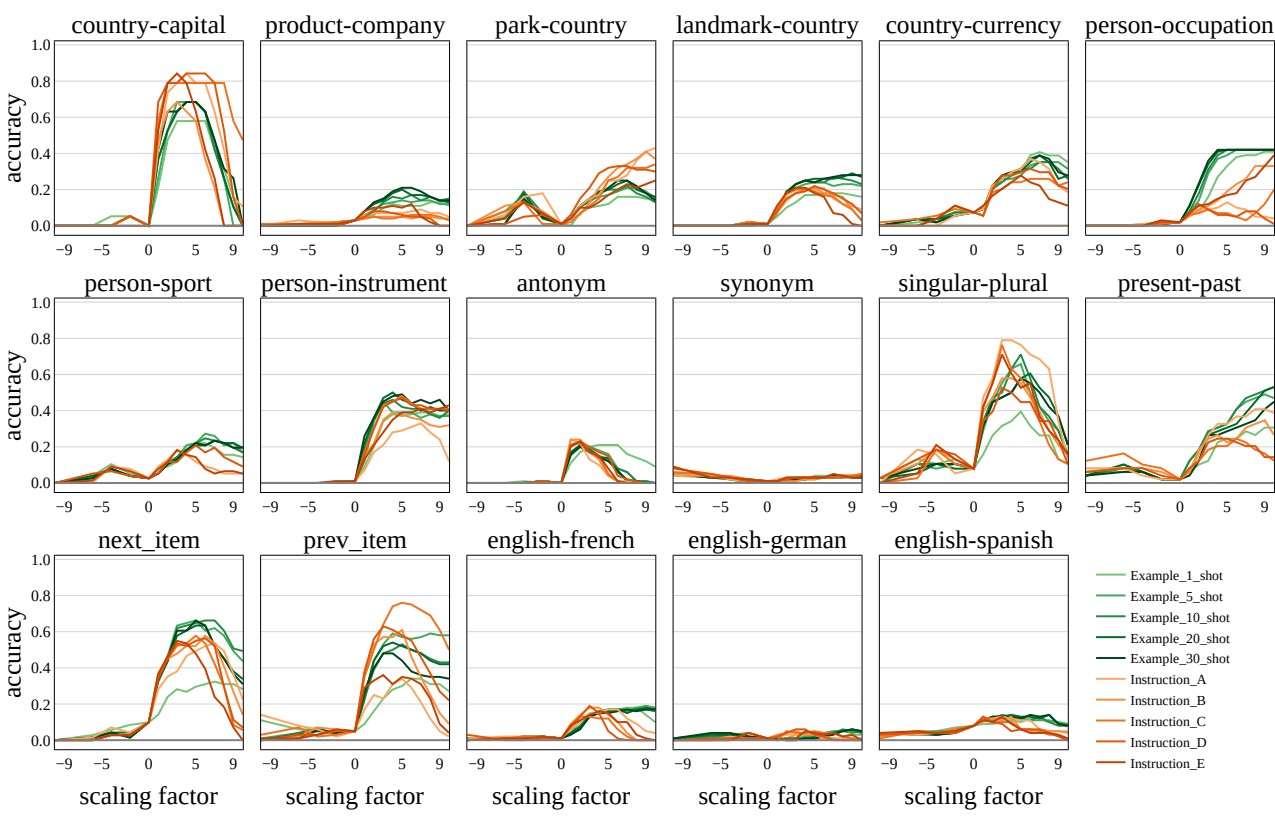

*Figure 59.* Quantification of the causal effect of lexical task heads in the Qwen2.5-7B-Instruct model

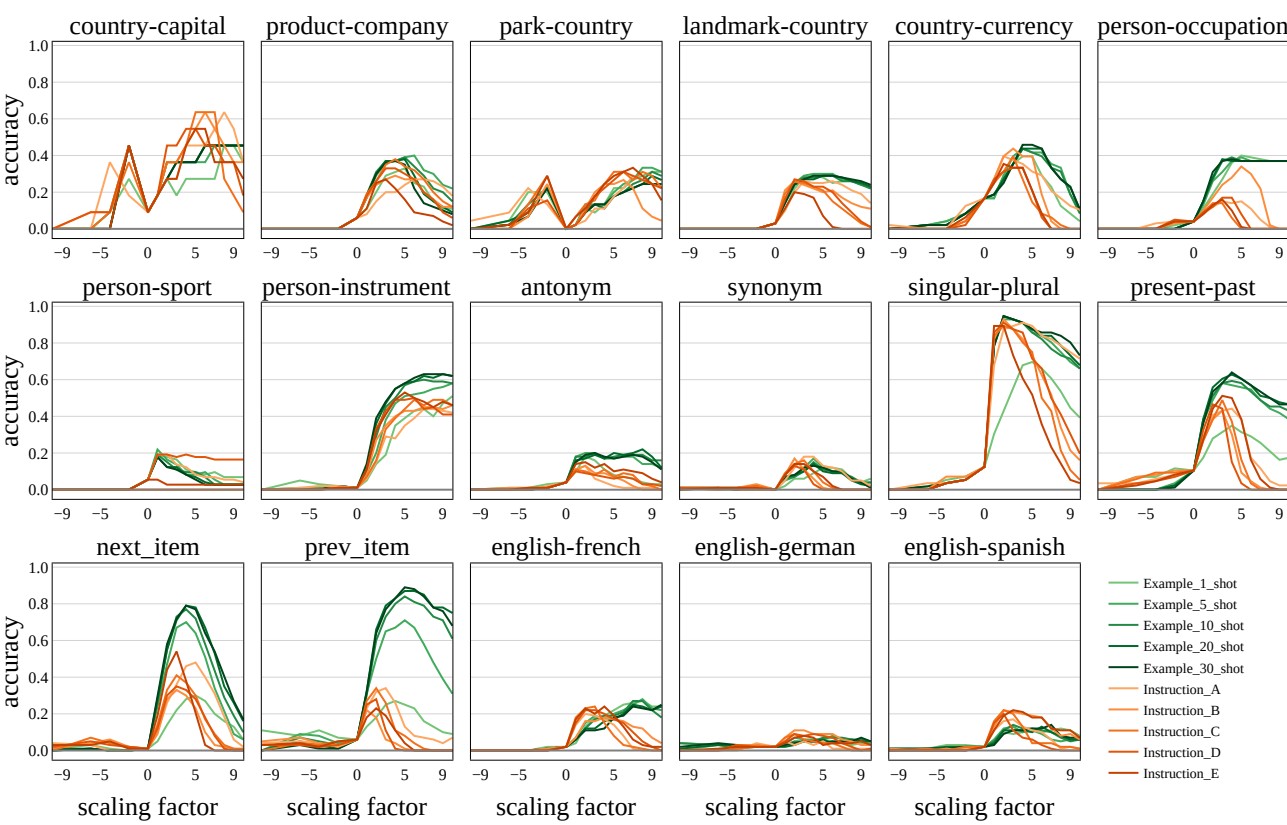

*Figure 60.* Quantification of the causal effect of lexical task heads in Qwen3-30B-A3B-Instruct-2507 model.

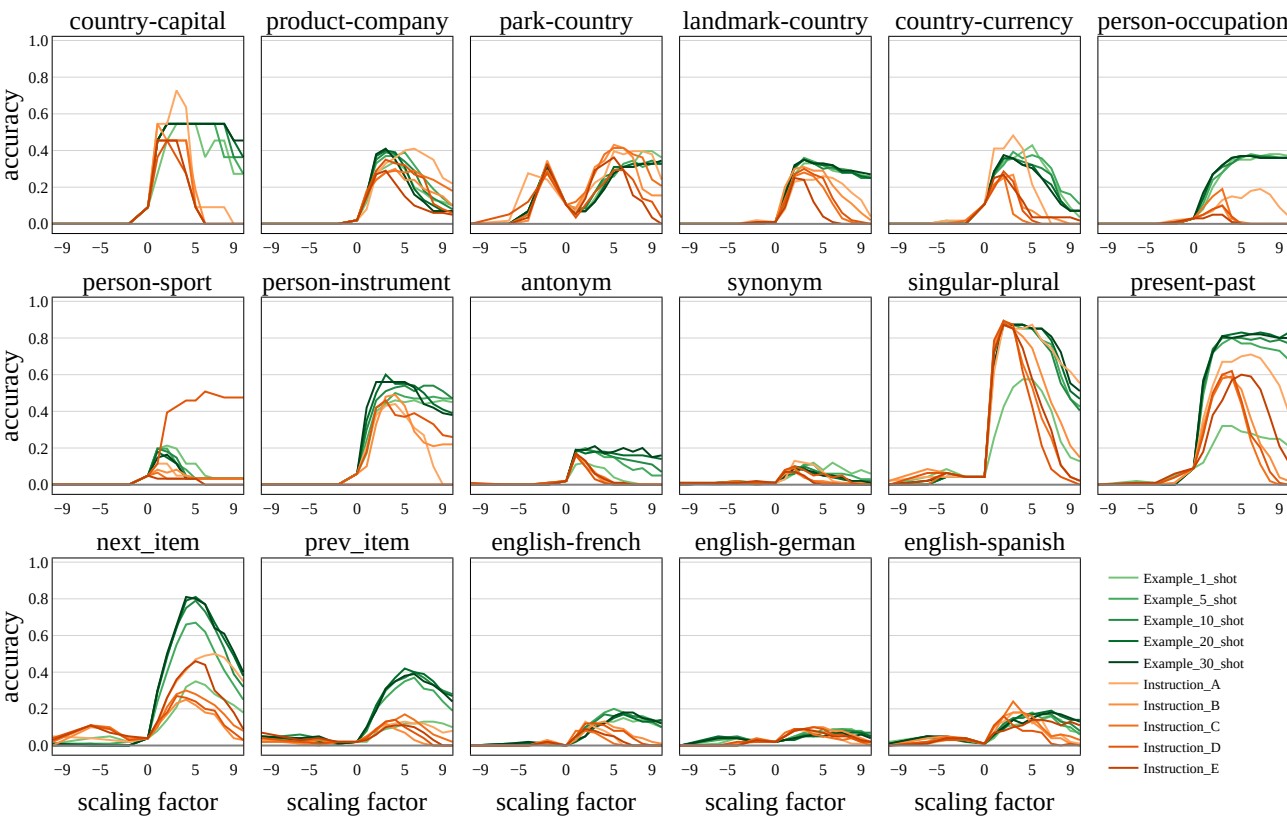

*Figure 61.* Quantification of the causal effect of lexical task heads in Qwen3-30B-A3B-Thinking-2507 model.

G.1.2. ACTIVATION PATCHING ON LEXICAL TASK HEADS EXTRACTED FROM EXAMPLE-BASED PROMPTS

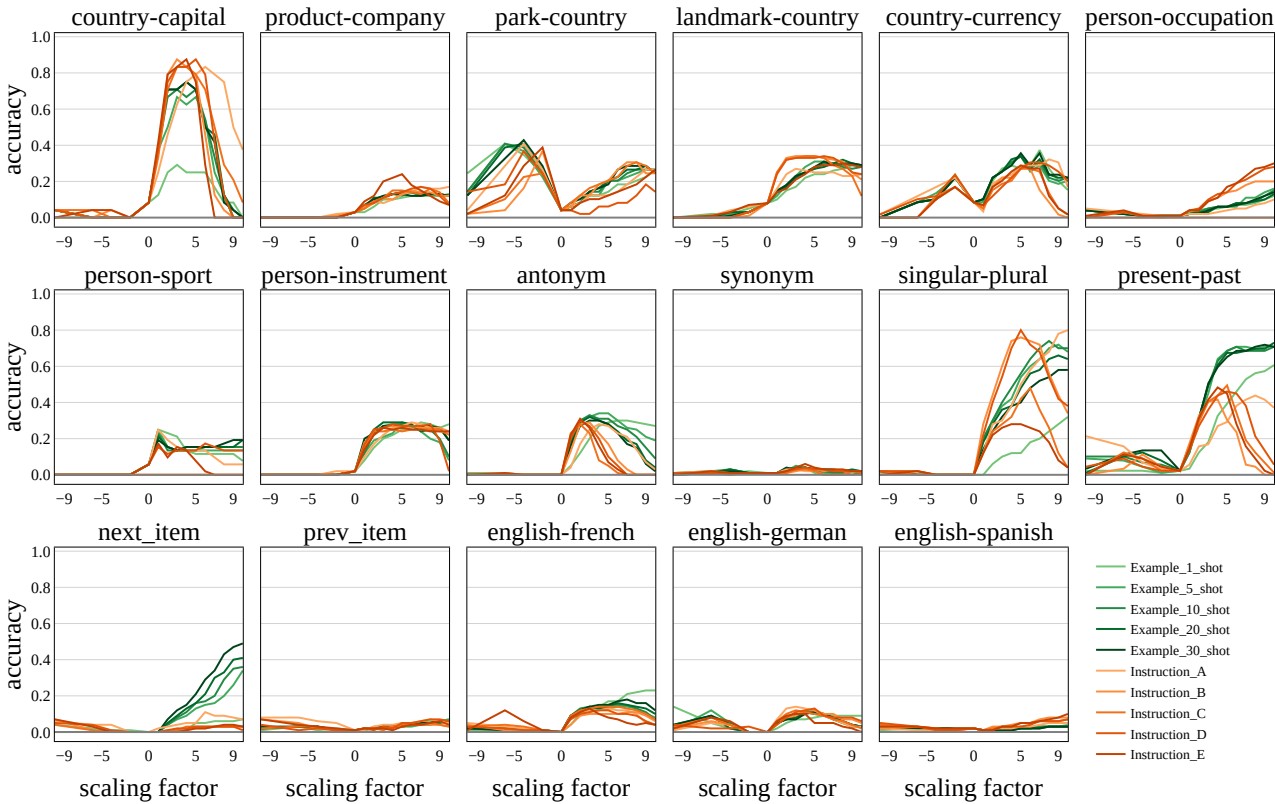

*Figure 62.* Quantification of the causal effect of lexical task heads in the Llama-3.1-8B-Instruct model

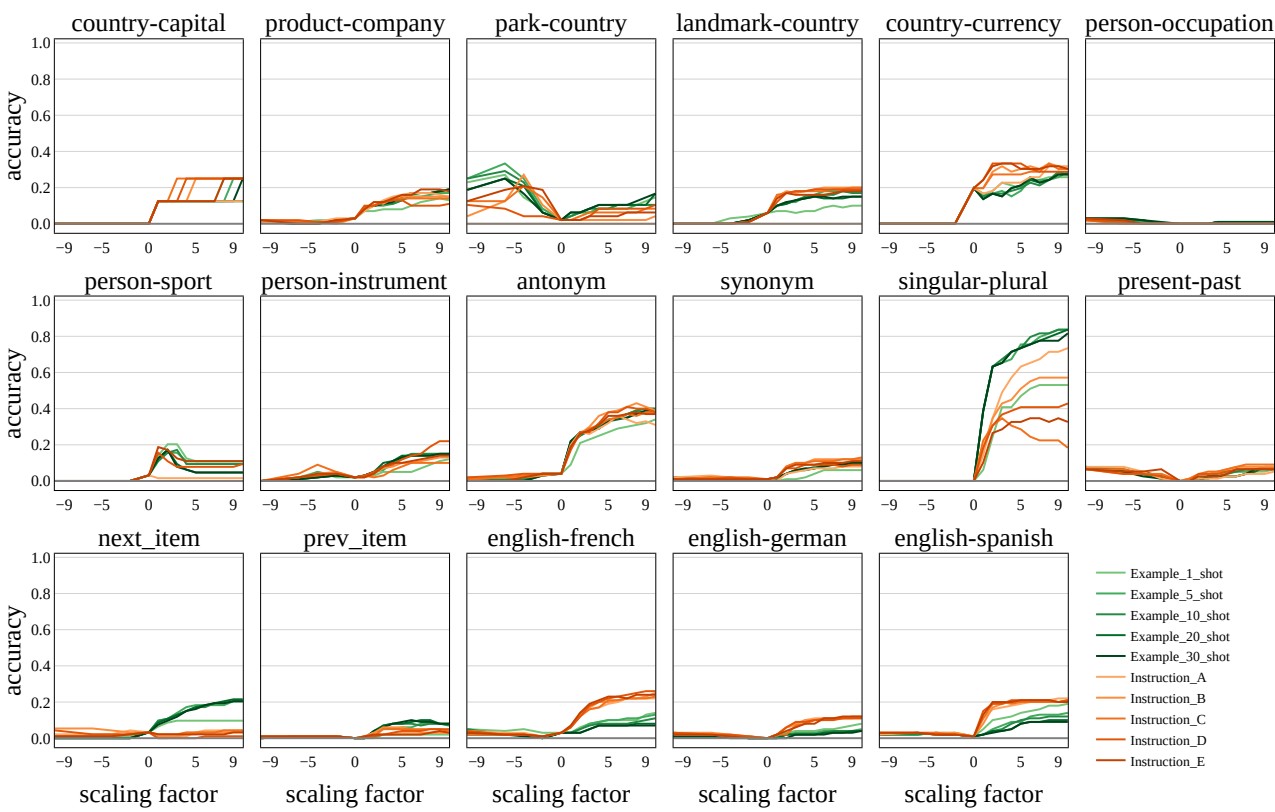

*Figure 63.* Quantification of the causal effect of lexical task heads in the gemma-2-9b-it model

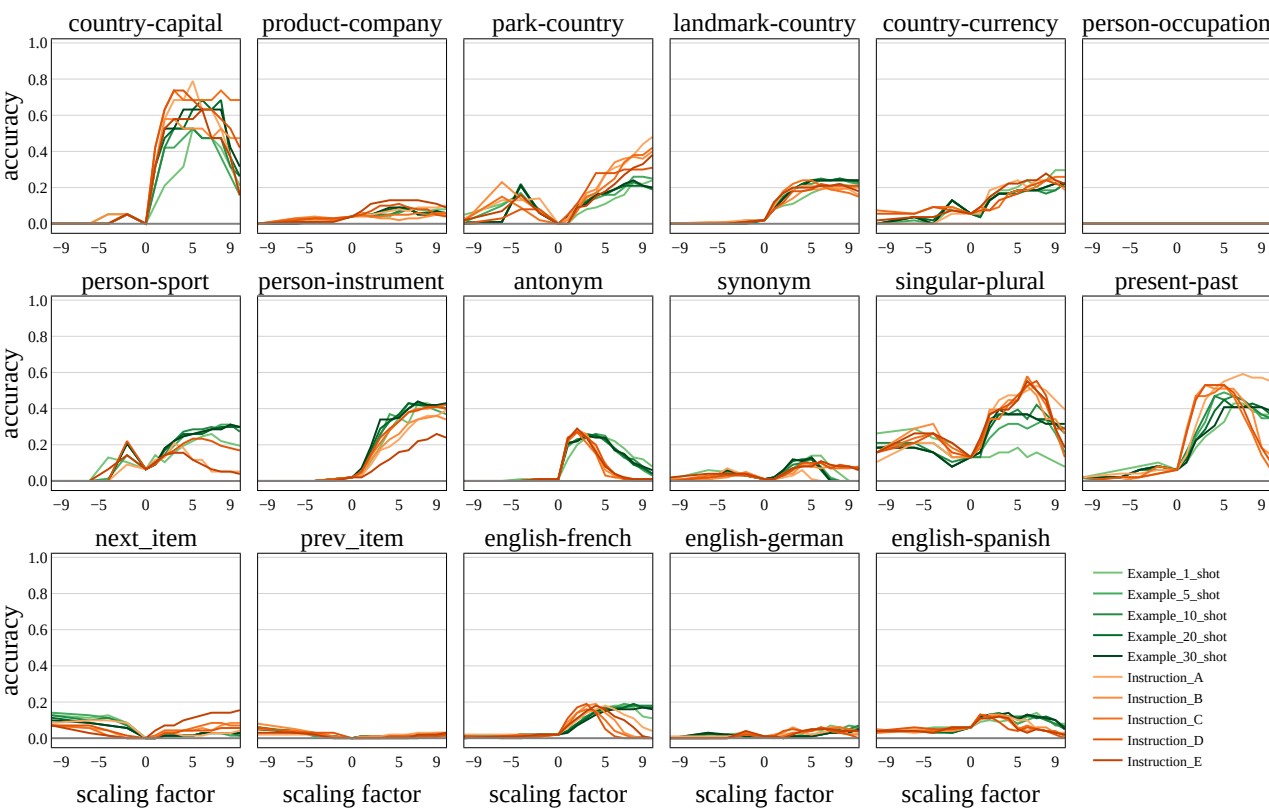

*Figure 64.* Results of the Qwen2.5-7B-Instruct model

# H. Lexical Task Heads are Insufficiently Activated

## H.1. Activation Strength Explains Behavior Variance

### H.1.1. COMPARE SUCCESS AND FAILURE CASES

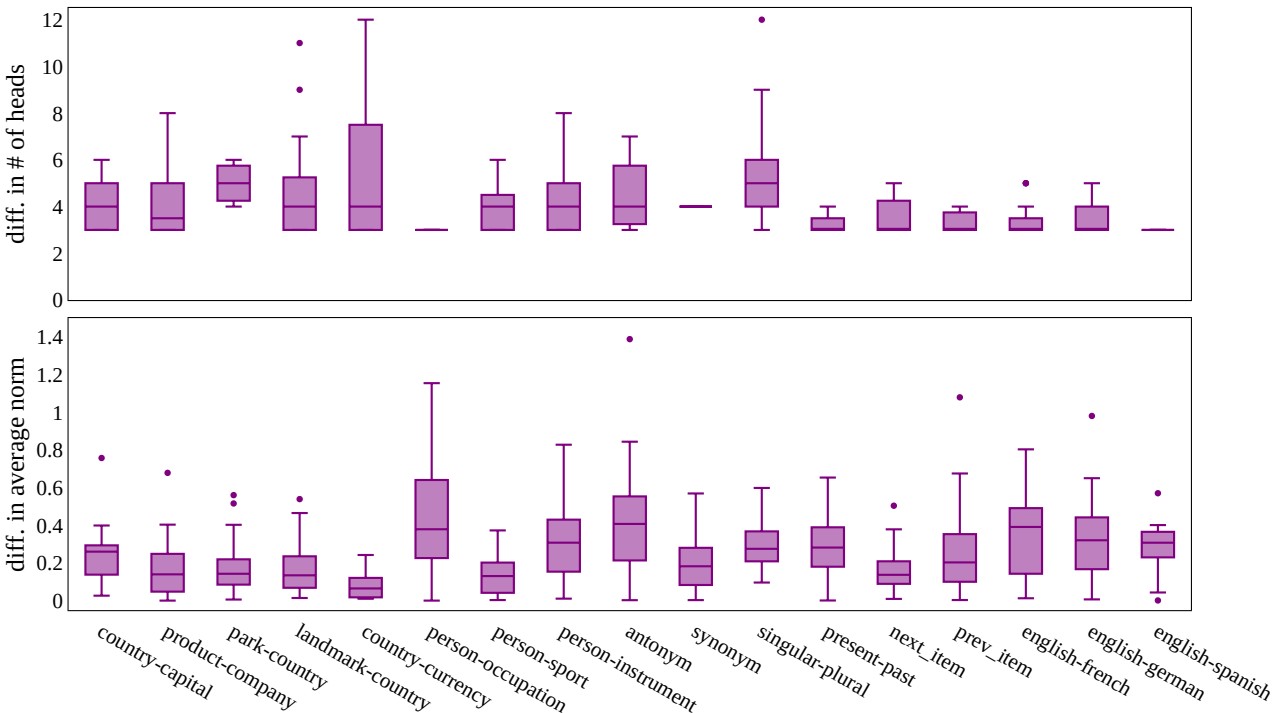

*Figure 65.* Some correct prompts activate more lexical task heads and they are activated more strongly than incorrect prompts. The y-axis is the difference of the number of heads or their norms between correct and incorrect prompts.

## H.1.2. MANY-SHOT EFFECT IN ICL

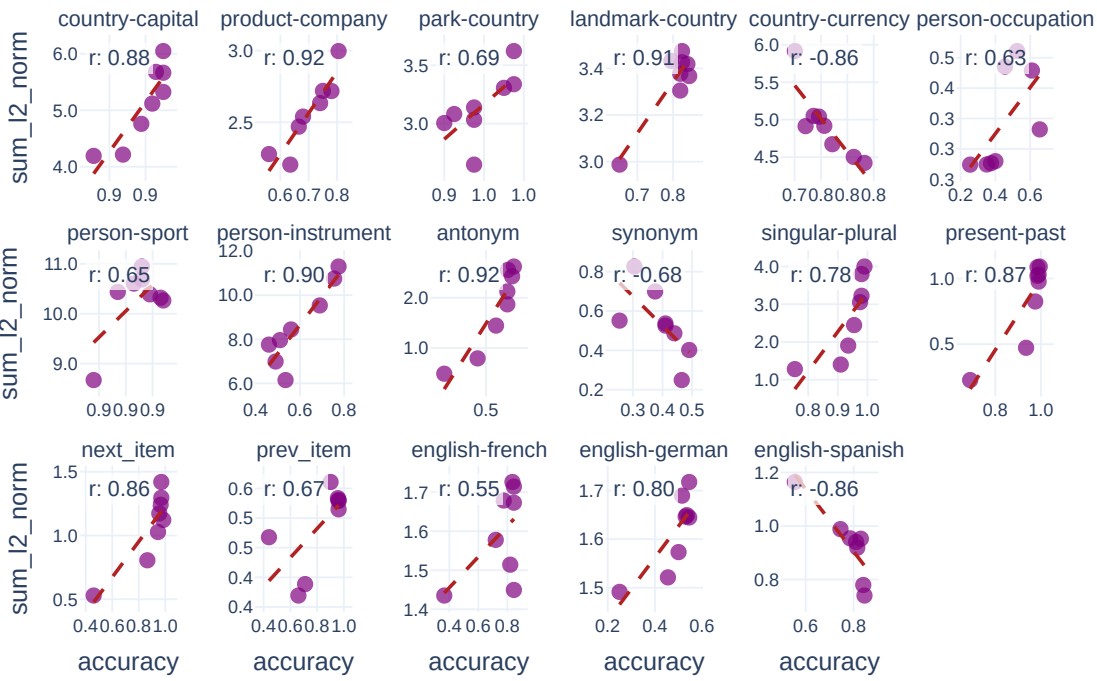

*Figure 66.* In 14 out of the 17 tasks, there is positive correlation between the accuracy and the magnitude of the outputs of lexical task heads. In each subplot for a task, each dot represents a given shot count.

H.1.3. INCREASING THE ACTIVATION OF THE LEXICAL TASK COMPONENTS FIX INCORRECT PROMPTS

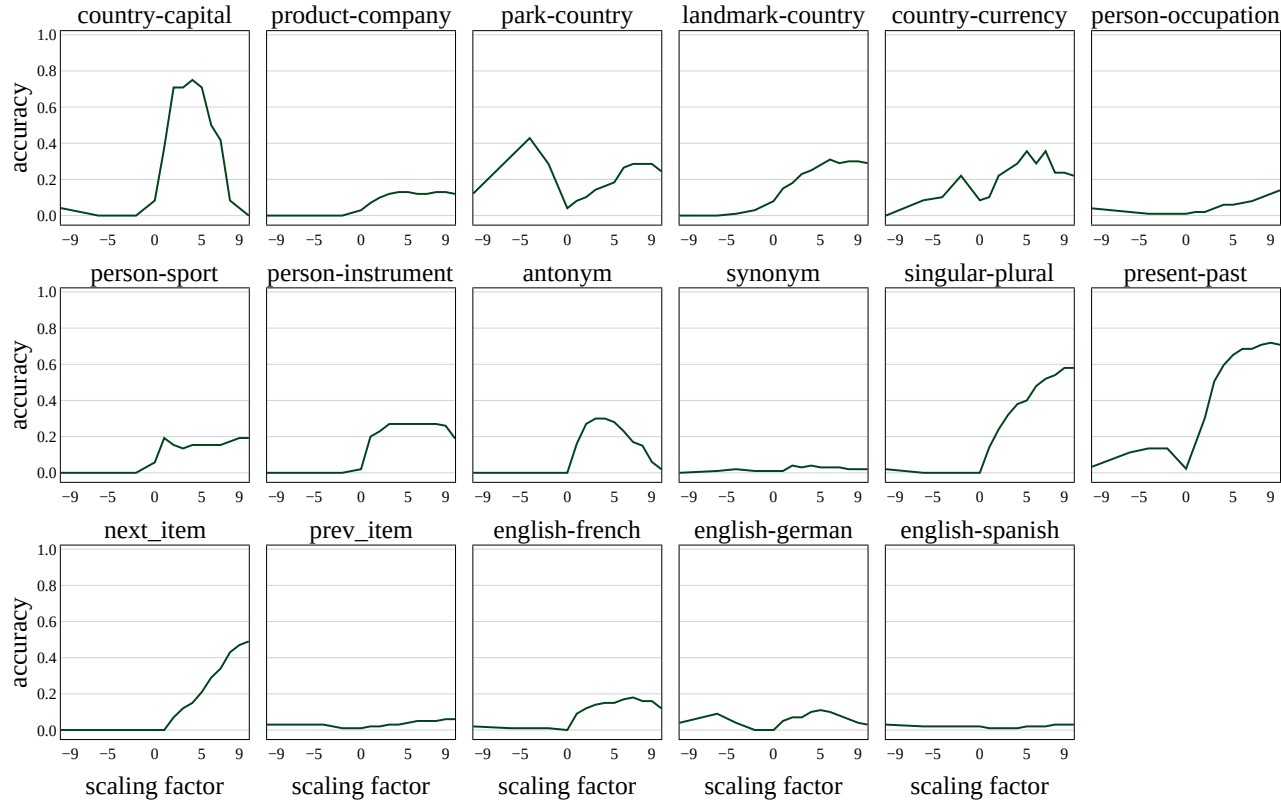

*Figure 67.* Scaling up the activation of lexical task heads can fix a portion of originally failed prompts. The average outputs of lexical task heads from correct prompts are patched to the incorrect prompts. The baseline accuracy is 0 for the incorrect prompts.

# I. Lexical Task Heads and Function Vector Overlap

Function vectors (Todd et al., 2024; Hendel et al., 2023) are latent representations of a task. In this work, we identify a different kind of task representation that we refer to as lexical task representations. A natural question emerges: Do lexical task representations and function vectors share underlying components (attention heads)?

To identify function vector heads, we follow the standard procedure from prior work (Todd et al., 2024; Davidson et al., 2025) to extract the *universal* function vector heads, which are a set of function vector heads shared across tasks. We then compare the universal function vector heads against the lexical task heads (identified using the procedure detailed in §2.2.2).

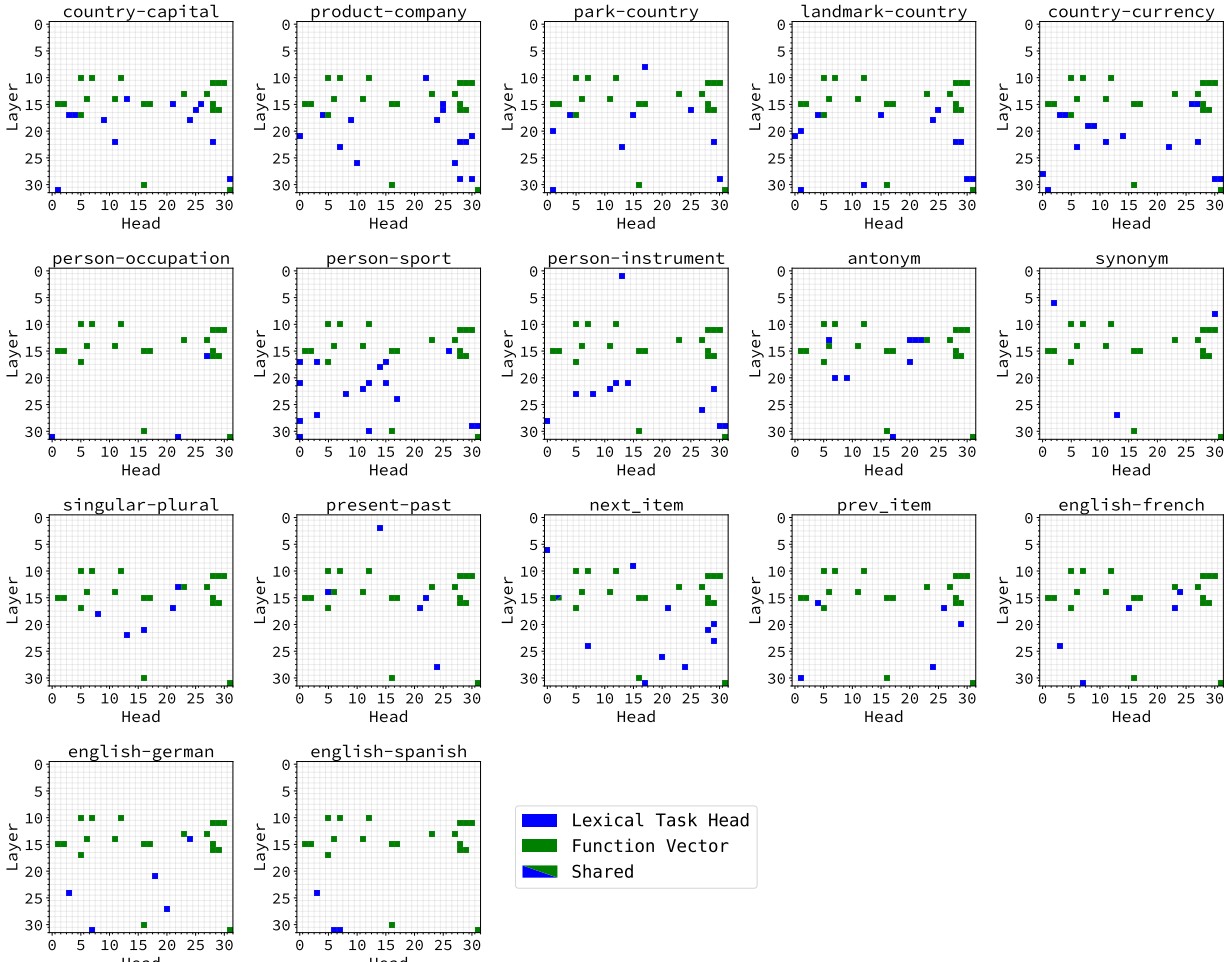

*Figure 68.* The distribution of lexical task heads and the top 20 universal function vector heads across layers of the Llama-3.1-8B-Instruct model. Despite both kinds of heads generate task representations, they are largely disjoint sets of heads.

## J. Retrieval Heads

### J.1. Criteria

Retrieval Head *Per-Prompt*: A head is classified as a retrieval head for a specific prompt if at least $n$ of its top $k$ decoded tokens match the correct answer (where $n = 1$, $k = 10$).

Retrieval Head *Per-Prompt-Style*: A head is considered a retrieval head for a given prompt style if it consistently retrieve the correct answer across many prompts of the prompting style. In this work, the criteria is that it satisfies the per-prompt criteria for at least $p\%$ of prompts in a prompt style for a given task (where $p = 10$).

### J.2. Number of Retrieval Heads

For each task we collect a set of heads separately for each prompting style—i.e., separately for the example-based prompts and the instruction-based prompts.

We quantify the number of retrieval heads that consistently retrieve the correct answer for each prompting style.

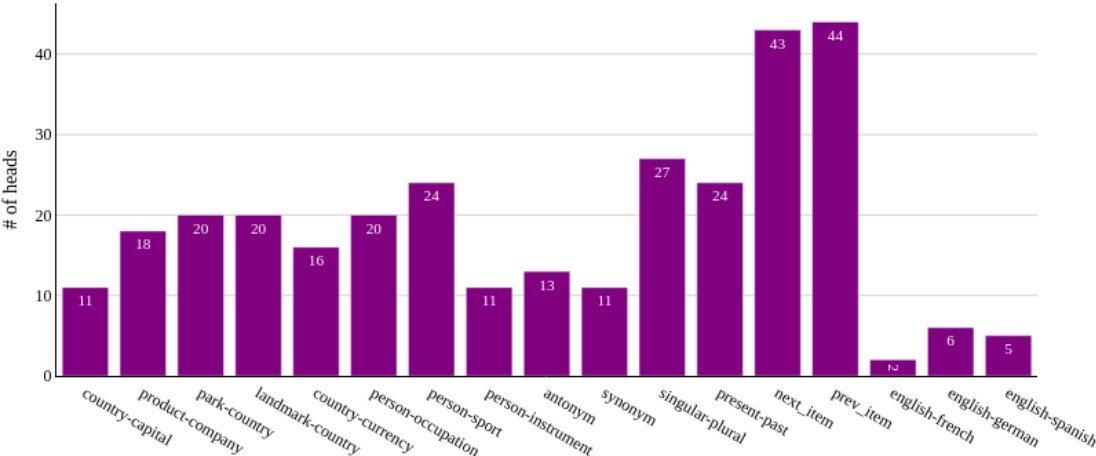

*Figure 69.* The number of retrieval heads across tasks for example-based prompts in Llama-3.1-8B model.

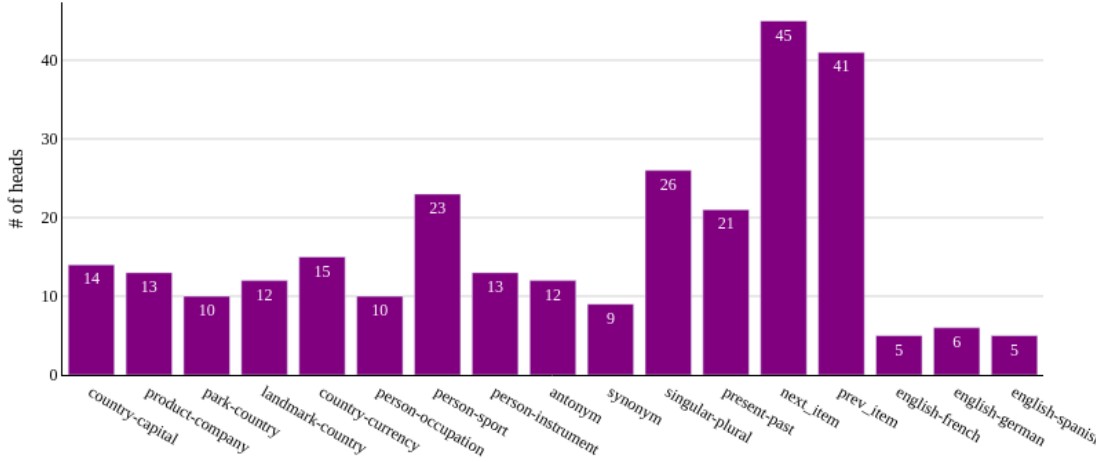

*Figure 70.* The number of retrieval heads across tasks for instruction-based prompts in Llama-3.1-8B model.

## J.3. Distribution Across Layers

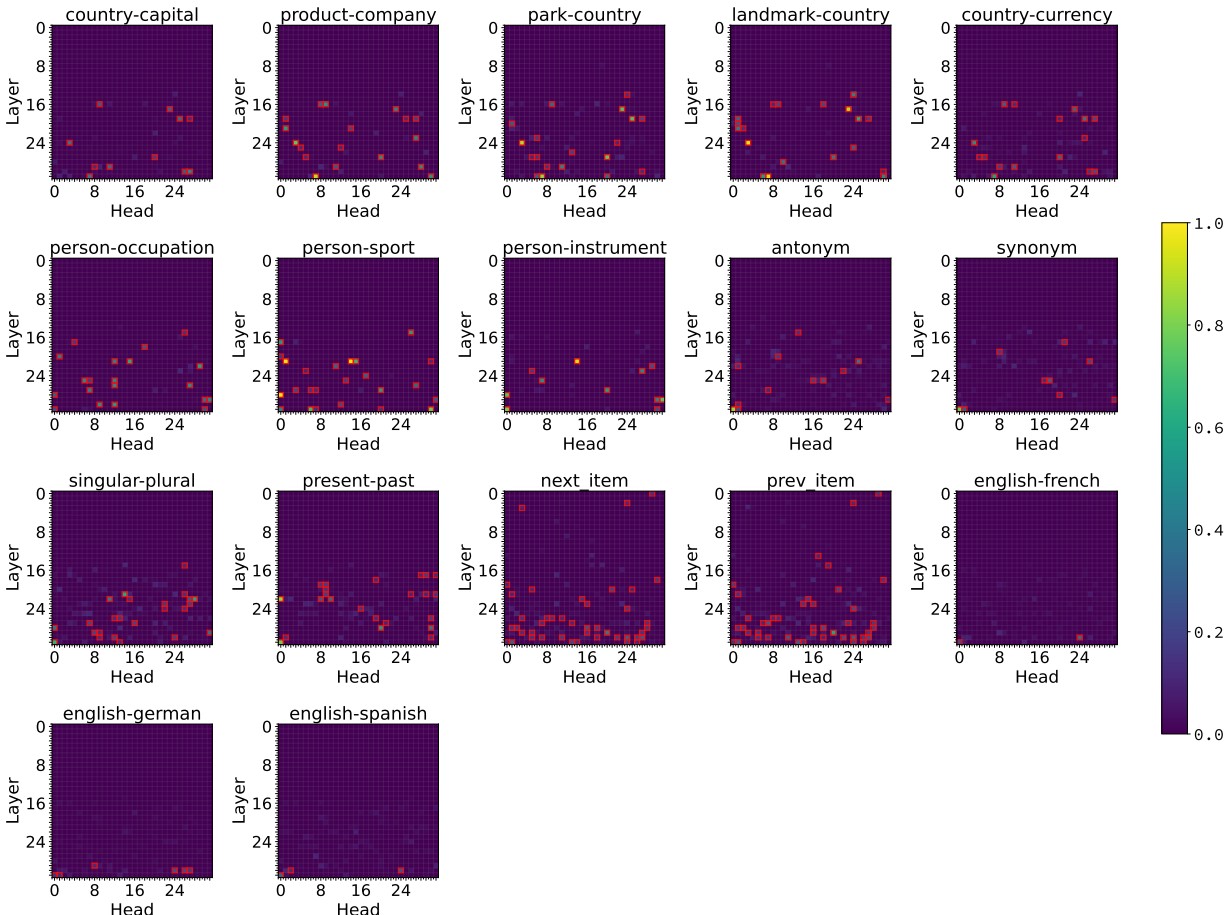

*Figure 71.* The heatmaps visualize the distribution of retrieval heads in the Llama-3.1-8B-Instruct model. The color scale represents the proportion of prompts that a given head retrieves the correct answer for. We highlight in red the heads that surpass the 10% threshold, indicating that they are the retrieval heads for a given prompting style.

## J.4. Component Overlap Across Prompting Styles

In this section, we measure the circuit reuse of retrieval heads across prompting styles. For each task (e.g., country-capital, antonym, etc.), we collect a set of heads separately for each prompting style—i.e., separately for the example-based prompts and the instruction-based prompts. We focus only on attention heads that are consistently activated by prompts (see criteria in J.1), producing the *correct* answers for a prompting style.

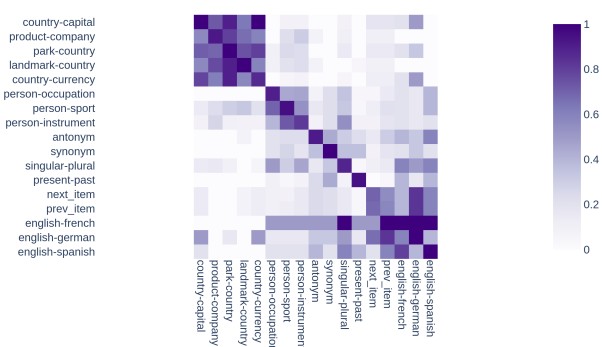

*Figure 72.* X-axis of the heatmap displays tasks for instruction-based prompting and Y-axis for example-based prompting. The darker the color, the more % of heads overlap for a given task across the two prompting styles.

# K. Analysis of Failure Cases

## K.1. Product-Producer Task (a knowledge retrieval task)

We compare prompts that contain ambiguity that implies a competing task against those that do not. The ambiguous prompts all contain the same misleading demonstration pairs (e.g., "Hwasong-6: North Korea"), and the query words are a set of different products. The control prompts contain the same number of demonstration pairs and the same set of query words, but the demonstration pairs are not ambiguous (at least to an average human, e.g., "Gmail: Google"). We measure the amount of lexical task heads (Fig. 73) and their activations (Fig. 9) of the intended target task (Product-Producer) as well as the competing off-target task (Product-Country).

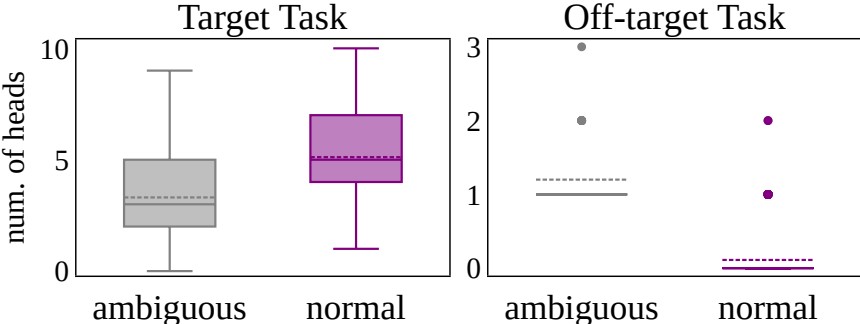

*Figure 73.* Ambiguous prompts dilute the signals of the intended target task (Product-Producer) and trigger the internal circuits of a competing task (Product-Country).

## K.2. "Number list" Task (not a knowledge retrieval task)

The "Number list" task requires the model to identify an underlying selection rule from a series of few-shot demonstrations. In this task, each example consists of a list of integers (the "input") followed by a single integer (the "label"). To succeed, the model needs to identify the relationship between the input set and the label in the few-shot examples, and select a number from the query list as the output.

### K.2.1. PROMPTS

An example prompt for the "Number list" task is shown below, where the selected number should be the even number in the list.

$[2, 5] \rightarrow 2,$
$[3, 4] \rightarrow 4,$
$[1, 6] \rightarrow \underline{\quad}$

An ambiguous prompt for the "Number list" task is shown below, where the selected number can either be the even number or the first number in the list.

$[2, 5] \rightarrow 2,$
$[4, 3] \rightarrow 4,$
$[6, 1] \rightarrow \underline{\quad}$

### K.2.2. LEXICAL TASK HEADS

Following the method described in 2.2.2, we identify the lexical task heads for the prompts of selecting the "even" or the "first" number in the list.

| Prompt | Head Index | Top Decoded Vocab |
|---|---|---|
| $[2, 5] \rightarrow 2,$ | (L13, H21) | pair, two, 两个，duo，Two，_pair |
| $[3, 4] \rightarrow 4,$ | (L18, H8) | pairs, pair, Pair, duo, couple, _pair |
| $[1, 6] \rightarrow$ ___ | (L21, H29) | similar, remaining, pairs, Remaining, pair, companion |

*Table 25.* Decoding attention head outputs on prompts of selecting the even number from a list

| Prompt | Head Index | Top Decoded Vocab |
|---|---|---|
| $[2, 5] \rightarrow 2,$ | (L17, H26) | first, starting, beginning, Starting, findFirst |
| $[5, 4] \rightarrow 5,$ | (L20, H29) | first, obil, first, 201, abaj, uhn |
| $[1, 6] \rightarrow$ ___ | (L28, H11) | .One, -BEGIN, initial, one, once |

*Table 26.* Decoding attention head outputs on prompts of selecting the first number from a list

### K.2.3. BEHAVIORAL RESULTS

We compare prompts that contain ambiguity that implies a competing task against those that do not (control prompts). The ambiguous prompts all contain the same demonstration pairs (e.g., $[2, 5] \rightarrow 2$, $[4, 3] \rightarrow 4$, where the task could be interpreted as either selecting the first number or the even number), and the query lists vary. The control prompts contain the same number of demonstration pairs and the same set of query lists as in the ambiguous prompts, but the demonstration pairs are not ambiguous (at least to an average human, e.g., $[2, 5] \rightarrow 2$, $[3, 4] \rightarrow 4$, where the even numbers are selected).

The accuracy (for correctly selecting the even number in the list) of the control prompts is 94%, suggesting that the model is able to identify and execute the target task. The accuracy of the ambiguous prompts is 0%, suggesting that the task ambiguity induce failures.

### K.2.4. PROMPT AMBIGUITY TRIGGERS COMPETING TASKS

For both the target (selecting the even number) and off-target task (selecting the first number), we measure and compare the amount of the lexical task heads of the ambiguous as well as the control prompts (defined in K.2.3).

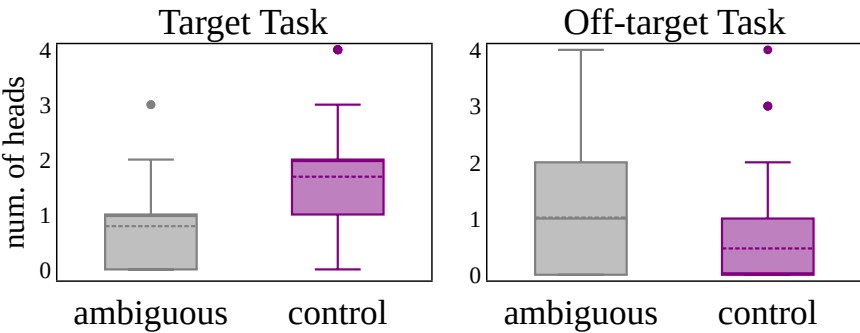

*Figure 74.* Ambiguous prompts dilute the signals of the target task (selecting the even number) and trigger the internal circuits of the off-target task (selecting the first number).

## L. Code Generation Tasks

As a proof of concept, we use Python and JavaScript code generation tasks. The task descriptive terms for identifying the heads are summarized in the Table below.

| Task | Task-descriptive Terms |
|------|------------------------|
| Python | _python, python, -python, Python, PYTHON, .python |
| Javascript | Javascript, _JavaScript, javascript, JavaScript |

We summarize the main results below. We detect and quantify Python and JavaScript lexical task heads (Fig. 75) and subsequently use them to steer the model to generate code in a specific programming language. For example, when "Python" heads are activated, the model switches from generating JavaScript to Python code. This steering effect depends on the activation strength (scaling factor) applied (Table 28 & 29). Our results across 164 coding problems from the HumanEval-X dataset (Zheng et al., 2024) show that lexical task heads remain functional even in sophisticated, long-form generation contexts.

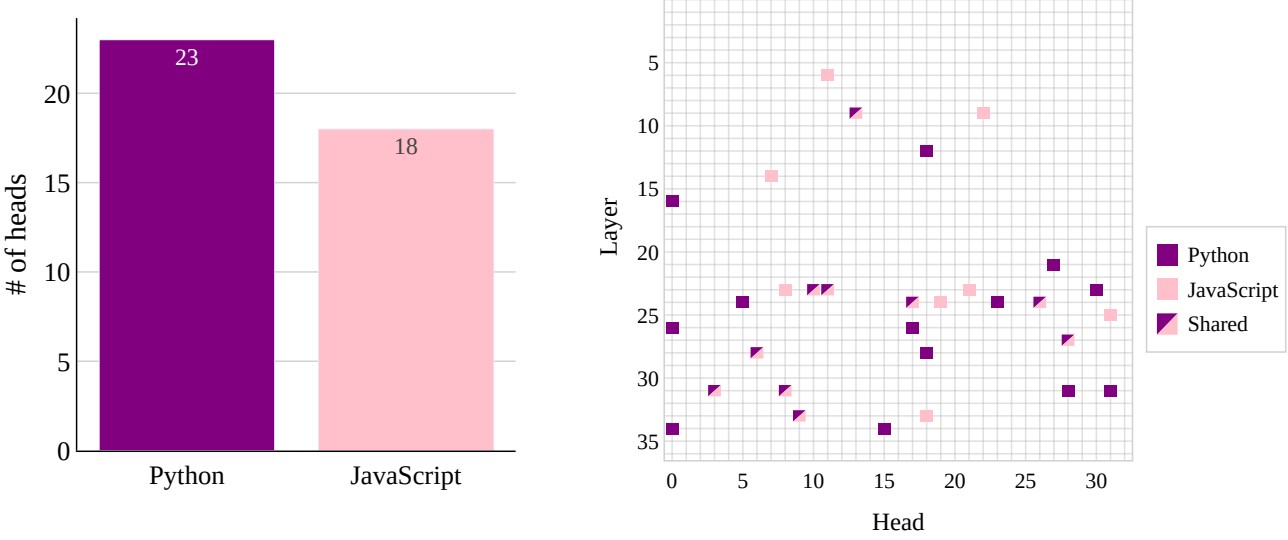

*(a)* The number of identified lexical task heads.   *(b)* The distribution of lexical task heads across layers.

*Figure 75.* Quantification and visualization of lexical task heads.

**Prompt**:
```
/* Check if in given list of numbers, are any two numbers closer to each other than
    given threshold.
    >>> hasCloseElements([1.0, 2.0, 3.0], 0.5)
    false
    >>> hasCloseElements([1.0, 2.8, 3.0, 4.0, 5.0, 2.0], 0.3)
    true
    */
const hasCloseElements = (numbers, threshold) => {
```

| Scaling factor | Model output |
| --- | --- |
| 1 | ```// Sort the list``` 
 ```const sorted = numbers.sort((a, b) => a - b);``` 
 ```// Check if any two consecutive numbers are closer than threshold``` 
 ```for (let i = 0; i < sorted.length - 1; i++) {``` 
 ```  if (Math.abs(sorted[i] - sorted[i + 1]) < threshold) {``` 
 ```    return true;``` 
 ```  }``` 
 ```}``` 
 ```return false;``` 
 ```};``` 
 ```// Test cases``` 
 ```console.log(hasCloseElements([1.0, 2.0, 3.0], 0.5)):``` 
 ```...``` |
| 6 | ```// Your code here``` 
 ```// Python``` 
 ```// for i in range(len(numbers)):``` 
 ```//    for j in range(i+1, len(numbers)):``` 
 ```//        if abs(numbers[i] - numbers[j]) < threshold:``` 
 ```//            return True``` 
 ```// return False``` 
 ```...``` |
| 10 | ```# Python Python``` 
 ```# Python``` 
 ```# Python``` 
 ```...``` |

*Table 28.* Activating "Python" lexical task heads steers the model from generating JavaScript to Python code. The steering effect depends on the activation strength (scaling factor) applied.

**Prompt**:
from typing import List
def has_close_elements(numbers: List[float], threshold: float) -> bool:
    """ Check if in given list of numbers, are any two numbers closer to each other than
    given threshold.
    >>> has_close_elements([1.0, 2.0, 3.0], 0.5)
    False
    >>> has_close_elements([1.0, 2.8, 3.0, 4.0, 5.0, 2.0], 0.3)
    True
    """

| Scaling factor | Model output |
|---|---|
| 1 | `for i in range(len(numbers)):`
   `for j in range(i+1, len(numbers)):`
     `if abs(numbers[i] - numbers[j]) < threshold:`
       `return True`
`return False`

... |
| 6 | `# JavaScript`
`# function hasCloseElements(numbers, threshold) {`
`#     for (let i = 0; i < numbers.length; i++) {`
`#         for (let j = i + 1; j < numbers.length; j++) {`
`#             if (Math.abs(numbers[i] - numbers[j]) < threshold) {`
`#                 return true;`
`#             }`
`#         }`
`#     }`
`#     return false`

... |
| 10 | `# JavaScript`
`# let's see if we can do this in JavaScript`
`# JavaScript`
`# let's see if we can do this in JavaScript`

... |

*Table 29.* Activating "JavaScript" lexical task heads steers the model from generating Python to JavaScript code. The steering effect depends on the activation strength (scaling factor) applied.

# M. Compositional Tasks

We test the lexical task representation mechanisms on three tasks: Landmark-Country-Capital, Park-Country-Capital, and Product-Company-Ceo, which are two-hop compositional tasks studied in prior interpretability work (Khandelwal & Pavlick, 2025). We replicate key findings in our paper summarized below: 1) Behavioral variability (Fig. 76), 2) Identification of lexical task heads for each of the two composing tasks. For example, for the Park-Country-Capital task, we separately detect country and capital city heads (Fig. 77), 3) Causal effect: activating these heads recovers 12%–50% of the performance (Fig. 78).

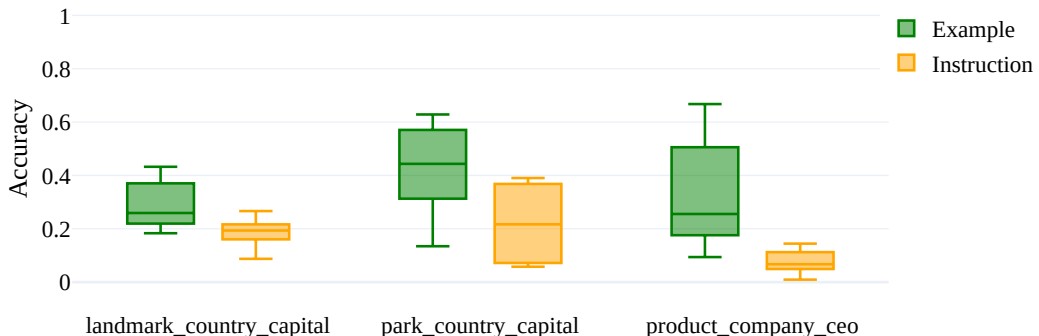

*Figure 76.* Behavioral results of two-hop compositional tasks.

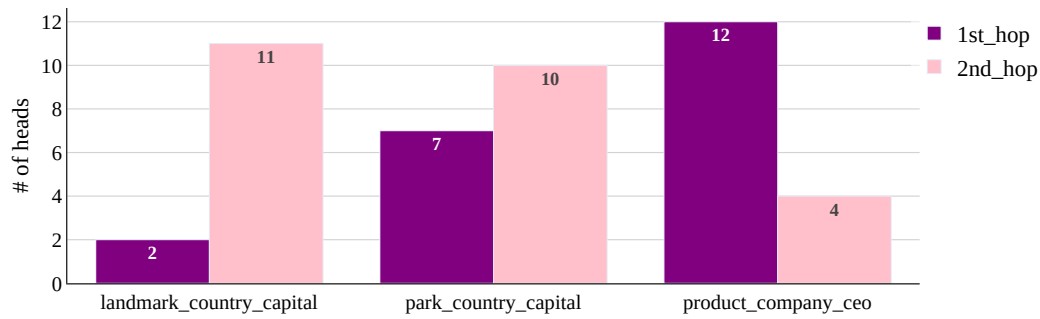

*Figure 77.* Quantification of the number of lexical task heads in two-hop compositional tasks.

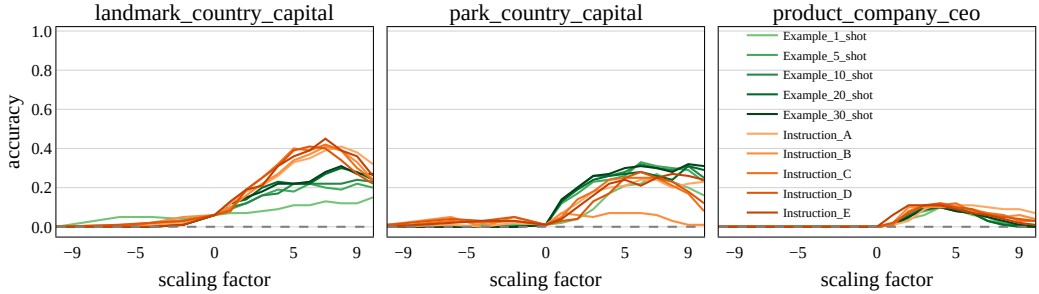

*Figure 78.* Quantification of the causal effect of lexical task heads. Scaling up the activation of lexical task heads can fix a portion of originally failed prompts. The average outputs of lexical task heads from correct prompts are patched to the incorrect prompts. The baseline accuracy is 0 for the incorrect prompts. Each solid line represents an activation patching experiment. For all lines, the activations of a same set of heads are patched to same prompts. The only difference is the source of the patched activation, which is cached from different prompting templates and styles, represented by different colors. Instruction templates A-E are wording variations.

# N. Compare Pretrained and Instruction-finetuned Models

We investigate the emergence of lexical task heads across training stages. We summarize key findings below: 1) Lexical task heads exist in base models. However, for most of the tasks, there are less lexical task heads than the instruction fine-tuned models (Fig. 79-81). Lexical task heads in base models and instruction models largely overlap (Fig 82-84). The causal effect of the lexical task heads is comparable to instruction models (Fig 85-87). We will include these analyses in the final version of the paper and we believe readers will find them informative.

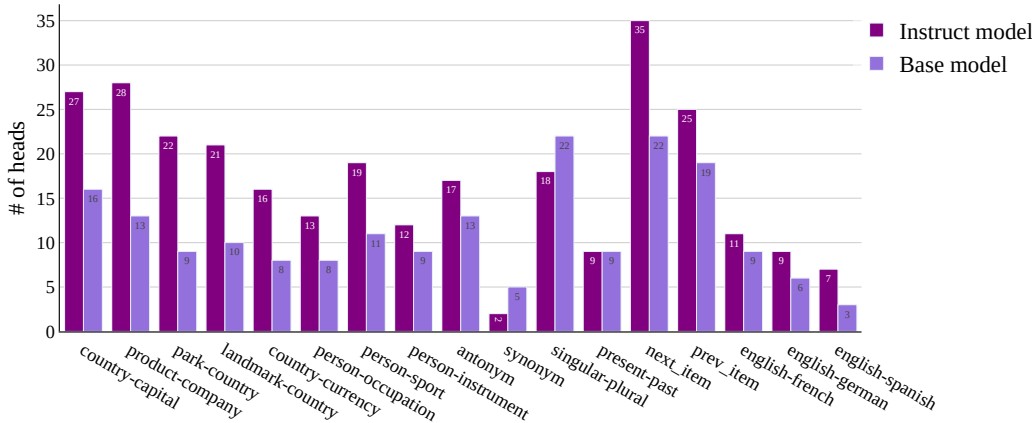

*Figure 79.* Compare the number of lexical task heads in the Instruct model (Llama-3.1-8B-Instruct) and the Base model (Llama-3.1-8B).

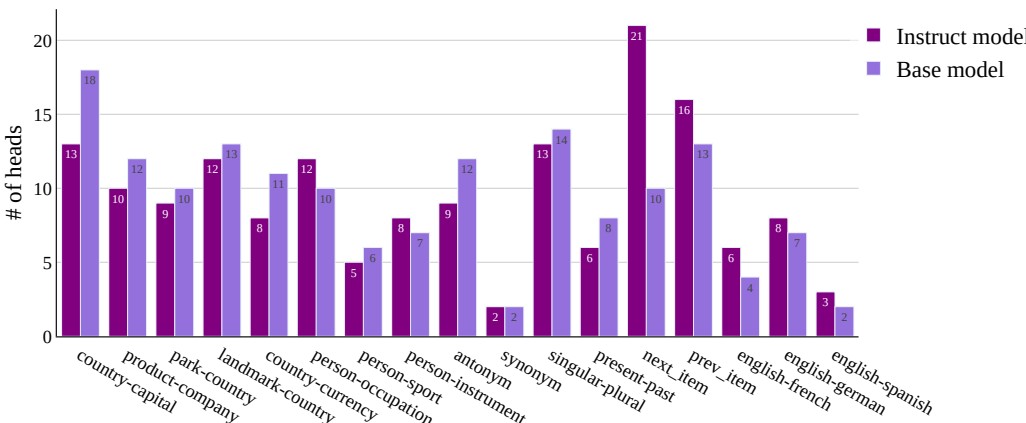

*Figure 80.* Compare the number of lexical task heads in the Instruct model (Qwen2.5-7B-Instruct) and the Base model (Qwen2.5-7B).

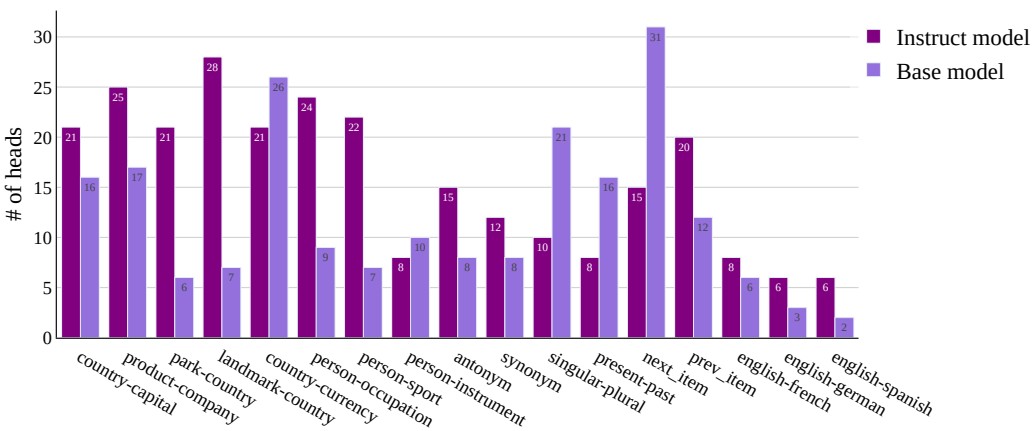

*Figure 81.* Compare the number of lexical task heads in the Instruct model (gemma-2-9b-it) and the Base model (gemma-2-9b).

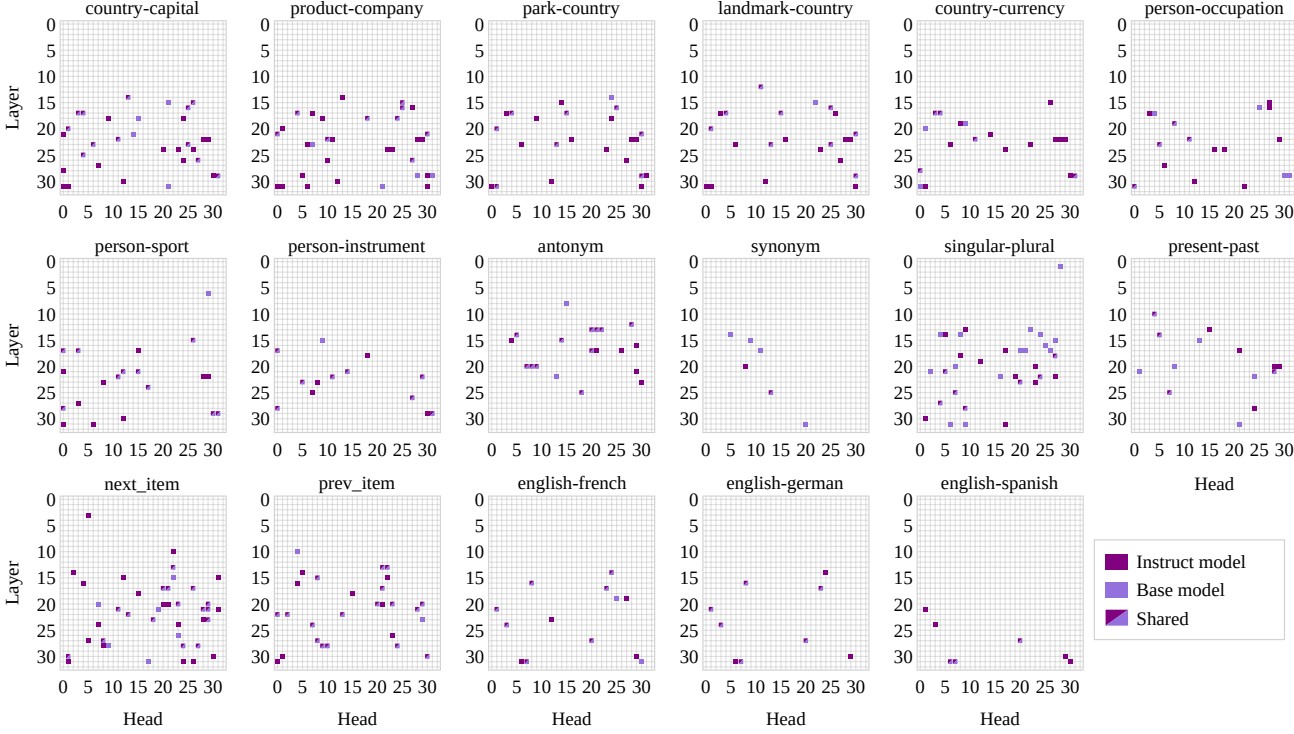

*Figure 82.* Compare the locations of lexical task heads in the Instruct model (Llama-3.1-8B-Instruct) and the Base model (Llama-3.1-8B).

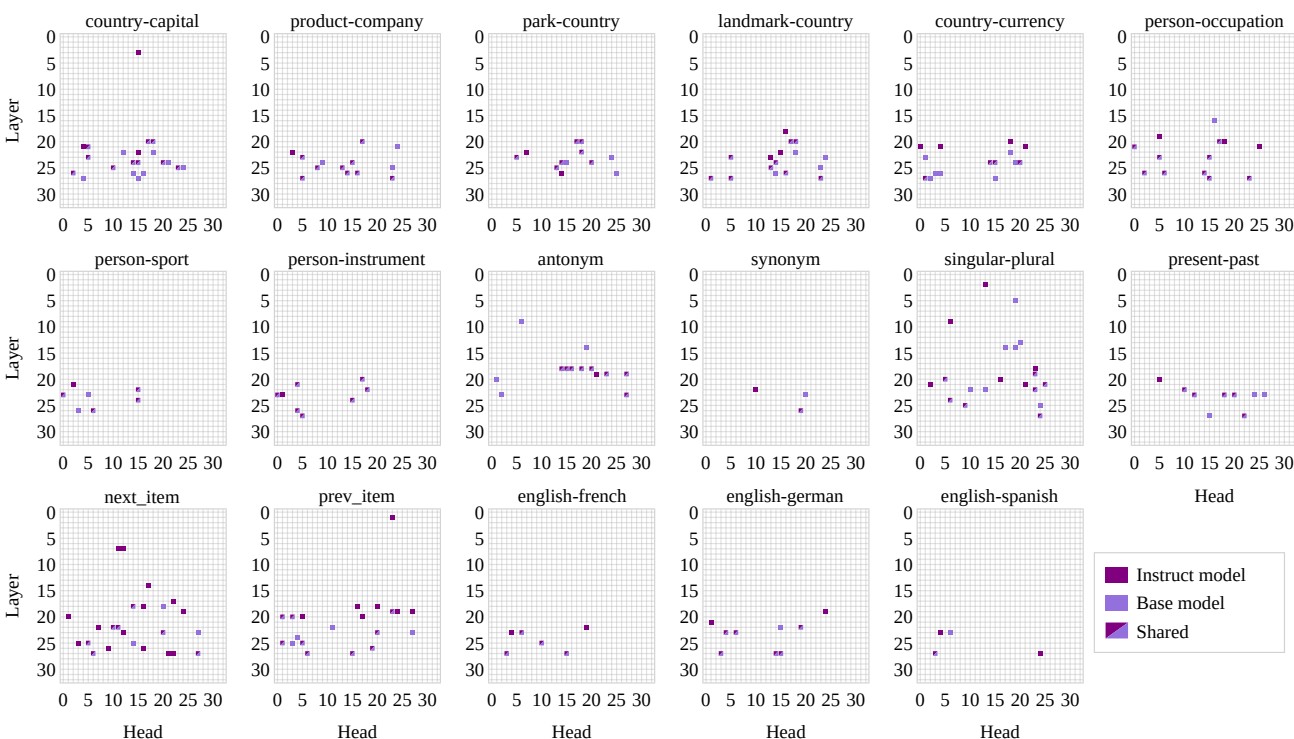

*Figure 83.* Compare the locations of lexical task heads in the Instruct model (Qwen2.5-7B-Instruct) and the Base model (Qwen2.5-7B).

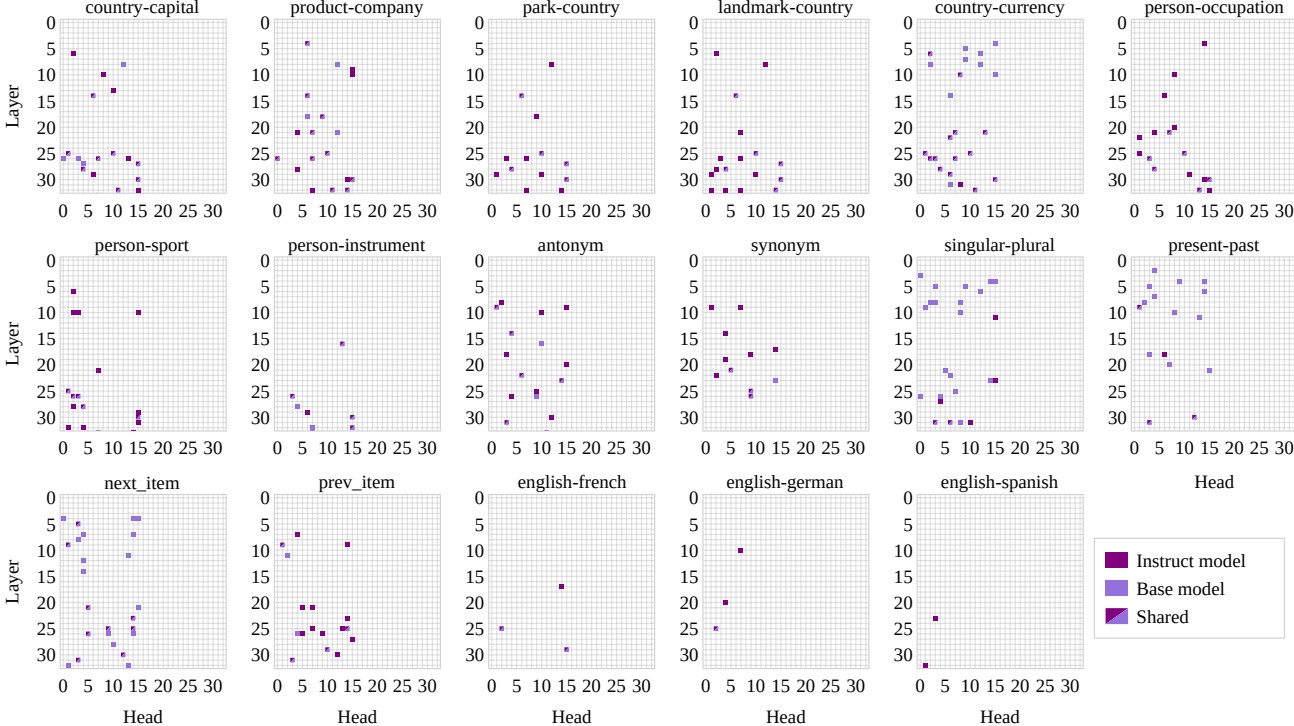

*Figure 84.* Compare the locations of lexical task heads in the Instruct model (gemma-2-9b-it) and the Base model (gemma-2-9b).

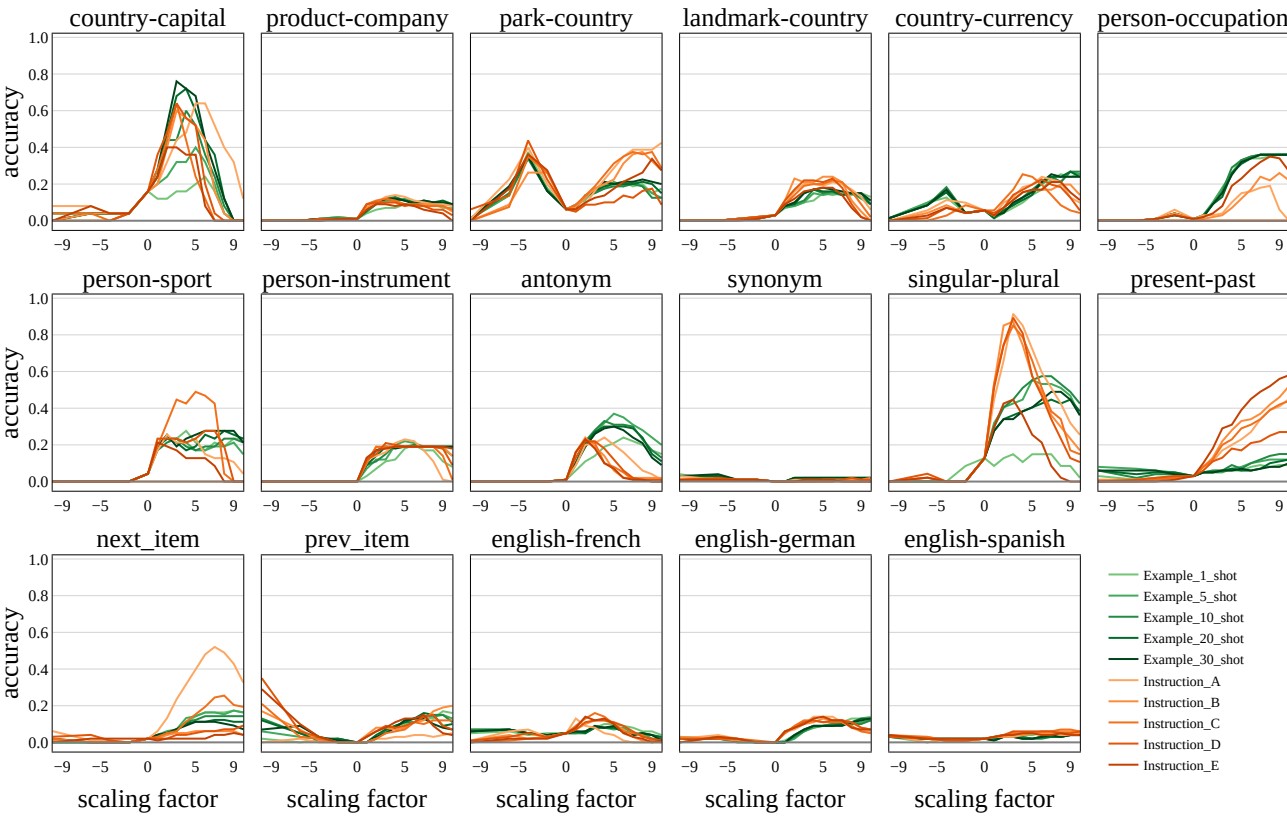

*Figure 85.* Quantification of the causal effect of lexical task heads in Llama-3.1-8B model. Scaling up the activation of lexical task heads can fix a portion of originally failed prompts. The average outputs of lexical task heads from correct prompts are patched to the incorrect prompts. The baseline accuracy is 0 for the incorrect prompts. Each solid line represents an activation patching experiment. For all lines, the activations of a same set of heads are patched to same prompts. The only difference is the source of the patched activation, which is cached from different prompting templates and styles, represented by different colors. Instruction templates A-E are wording variations.

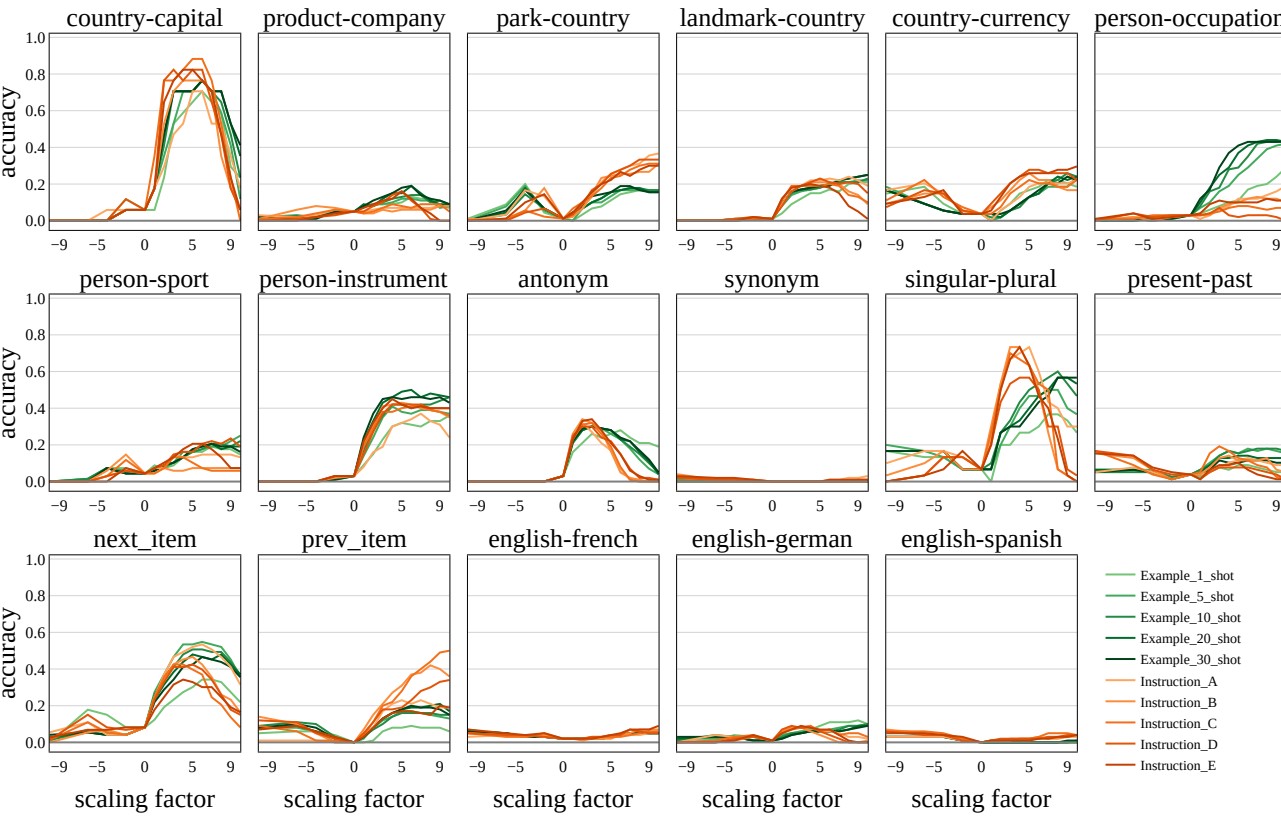

*Figure 86.* Quantification of the causal effect of lexical task heads in Qwen2.5-7B model. Scaling up the activation of lexical task heads can fix a portion of originally failed prompts. The average outputs of lexical task heads from correct prompts are patched to the incorrect prompts. The baseline accuracy is 0 for the incorrect prompts. Each solid line represents an activation patching experiment. For all lines, the activations of a same set of heads are patched to same prompts. The only difference is the source of the patched activation, which is cached from different prompting templates and styles, represented by different colors. Instruction templates A-E are wording variations.

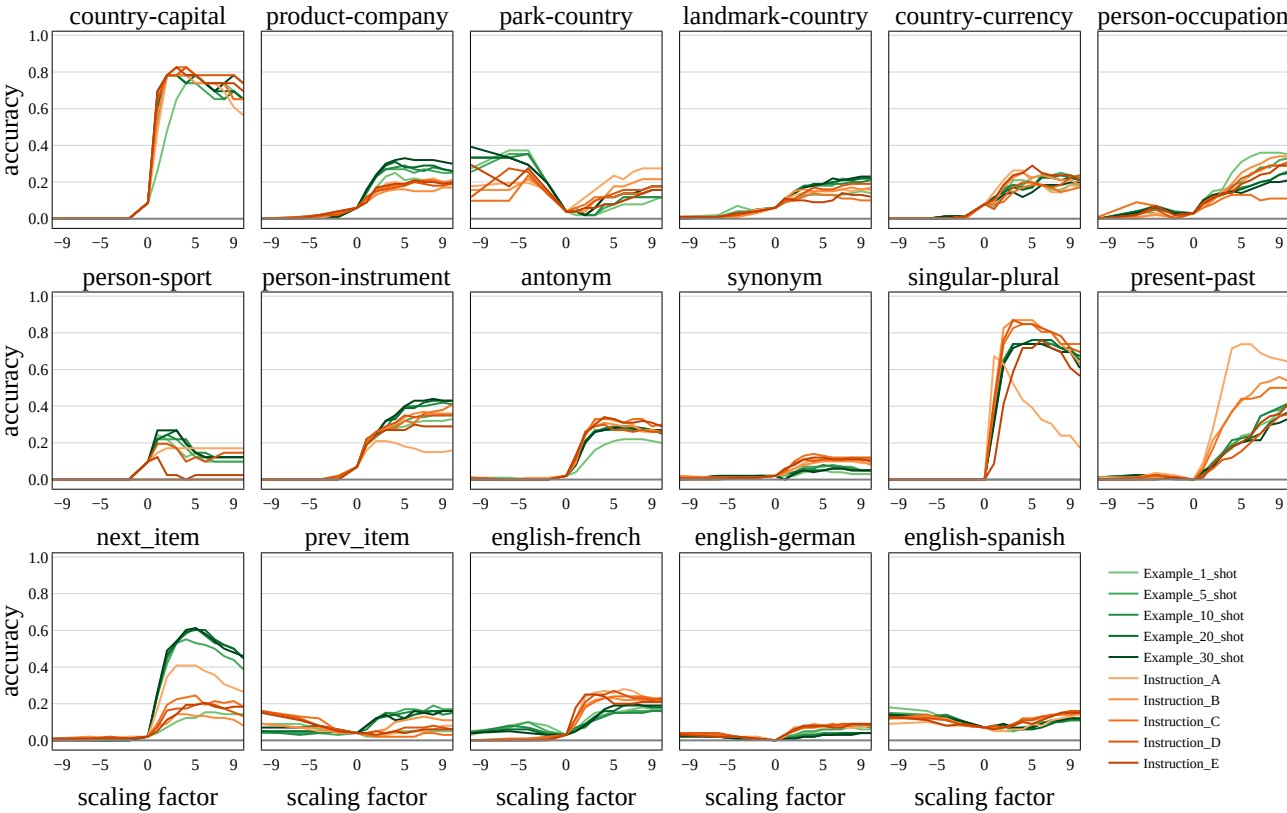

*Figure 87.* Quantification of the causal effect of lexical task heads in gemma-2-9b model. Scaling up the activation of lexical task heads can fix a portion of originally failed prompts. The average outputs of lexical task heads from correct prompts are patched to the incorrect prompts. The baseline accuracy is 0 for the incorrect prompts. Each solid line represents an activation patching experiment. For all lines, the activations of a same set of heads are patched to same prompts. The only difference is the source of the patched activation, which is cached from different prompting templates and styles, represented by different colors. Instruction templates A-E are wording variations.

## O. Use of LLMs to Generate Task Descriptive Terms

The definition of LTH relies on a predefined list of task-descriptive terms, which can potentially introduce bias into the results. As an attempt to address this, we leverage several frontier LLMs (GPT-5.4 Mini, Claude 4.6 Sonnet, and Grok 4.2 Beta) to generate task-descriptive terms. We find that the lexical task heads identified with those terms are comparable to the originally identified ones (Fig. 88 & 89). Thus, we conclude that our predefined task-descriptive terms encompass relatively comprehensive task descriptions and are sufficient to locate lexical task heads that have causal effect on model performance.

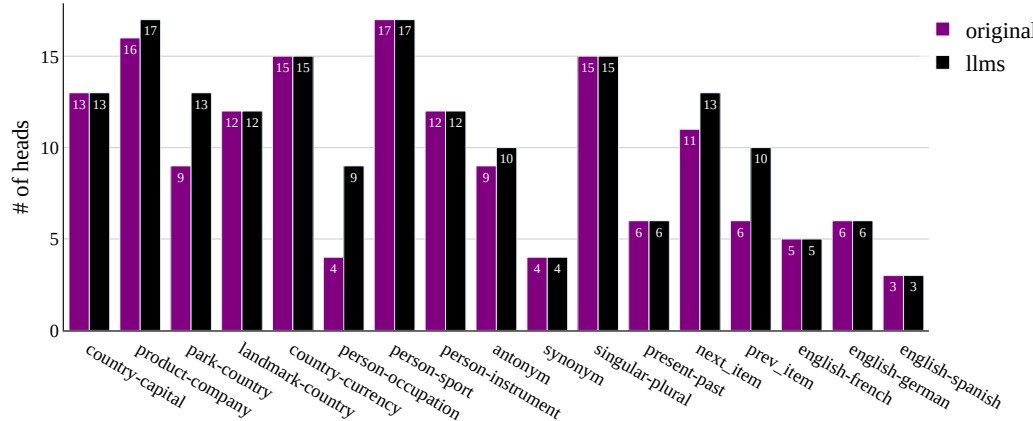

*Figure 88.* Compare the number of lexical task heads identified with the original predefined task descriptive terms versus the terms generated by three language models: GPT-5.4 Mini, Claude 4.6 Sonnet, and Grok 4.2 Beta.

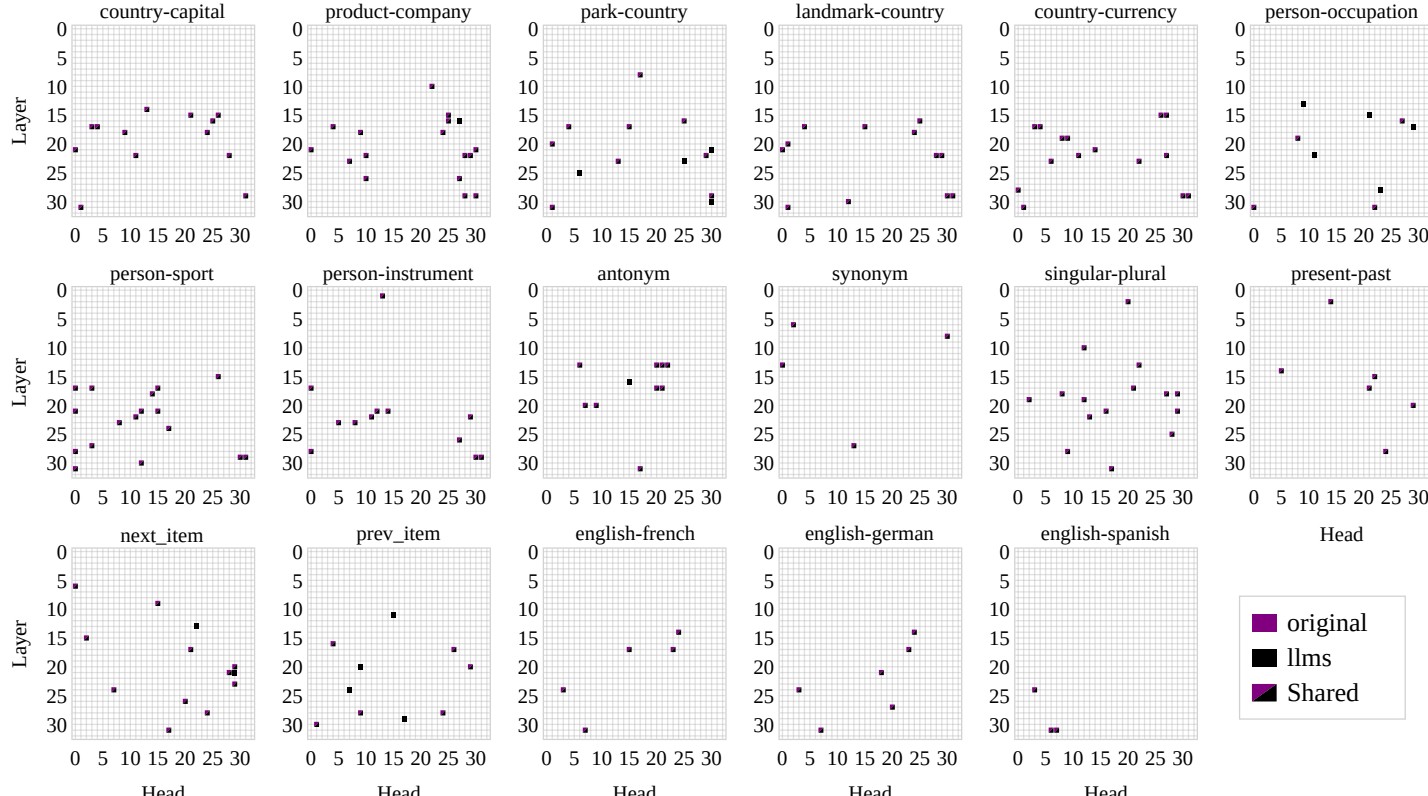

*Figure 89.* Compare the distribution of lexical task heads identified with the original predefined task descriptive terms versus the combined set of terms generated by three language models: GPT-5.4 Mini, Claude 4.6 Sonnet, and Grok 4.2 Beta.

