# OpenReview forum: "Shared Lexical Task Representations Explain Behavioral Variability In LLMs"
_ICML.cc/2026/Conference — ICML 2026 regular_

### Official Review · Reviewer_2g3w · 2026-03-12

**Soundness:** 3
**Presentation:** 3
**Significance:** 3
**Originality:** 3
**Overall Recommendation:** 5
**Confidence:** 3

**Summary:**

This paper explores why LLMs react differently to instructions compared to few-shot examples. The paper suggests that both methods actually rely on the same internal mechanism, and name it lexical task heads, which are specific attention heads that translate prompts into understandable task descriptions within the model's own vocabulary. The authors found that differences in how the model behaves are directly linked to how strongly these shared task heads are activated, and when a prompt is unclear, it can cause competition between different task representations, leading to inconsistent outputs.

**Compliance With Llm Reviewing Policy:**

Affirmed.

**Final Justification:**

The authors have done additional experiments that address my concerns.

**Key Questions For Authors:**

do lexical task heads emerge in pretraining or after instruct finetuning?

**Limitations:**

See weaknesses

**Strengths And Weaknesses:**

Strengths:
- This paper study an interesting question on why different prompting style leads to different performance on the same task.
- the proposed lexical task heads is novel and is a meaningful contribution to mechanistic interpretability.

Weaknesses:
- the tasks scope is limited to simple tasks like country-captial, product-company, etc. It is unclear if the findings generalize to other complex tasks. In real world, prompt sensitivity matters most to tasks like reasoning or coding.
- lexical task heads rely heavily on projecting component outputs into the vocabulary space using the logit lens technique, which is known to be unreliable and heuristic. Meanwhile, by explicitly searching for attention heads that outputs some specific words, it is guaranteed to find some heads.
- Given the findings, it still seems fails to imply which prompts will fail or giving direction for a better prompt design.

---

> ### Author Rebuttal · Authors · 2026-03-29
>
> Thank you for providing constructive feedback that we address below to strengthen the paper.
>
> > Limited task scope — "the tasks scope is limited to simple tasks like country-capital, product-company, etc. It is unclear if the findings generalize to other complex tasks. In real world, prompt sensitivity matters most to tasks like reasoning or coding"
> - Our choice of tasks is consistent with recent interpretability work (Hendel et al., Todd et al., Davidson et al.) utilizing the same kinds of tasks. We leverage these simple tasks to isolate specific mechanisms in a complex system, which is a necessary first step before evaluating generalization to more complex scenarios.
> -  As a proof of concept, we use Python and JavaScript code generation tasks. We describe the results below and link the **figures / tables [here](https://github.com/anonymous-conf-acount/icml_submission/blob/e0c328e6b3329bb47364d7bb1d2e7aaaedc28e6e/results/code_generation/Code_generation.pdf)**. We detect and quantify _Python_ and _JavaScript_ lexical task heads (linked Fig. 1 and Table 1) and subsequently use them to steer the model to generate code in a specific programming language. For example, when ``Python’’ heads are activated, the model switches from generating JavaScript to Python code. This steering effect depends on the activation strength (scaling factor) applied (see linked pp. 2–3, Table 2–3). Our results across 164 coding problems from the HumanEval-X dataset (Zheng et al., 2024) show that lexical task heads remain functional even in sophisticated, long-form generation contexts.
> - We find that lexical task heads are involved in reasoning models. We highlight three key results on the Qwen3-30B-A3B-Thinking-2507 model and link the **figures [here](https://github.com/anonymous-conf-acount/icml_submission/blob/6e7663a7b8372bb5c14d91389eab7da2fd90e741/results/Qwen3_models.pdf)**: 1) Behavioral variability (linked Fig. 2),  2) Lexical task heads identification (linked Fig. 4), 3) Causal effect: activating these heads recover 10%--90% of the performance (linked Fig. 6).
> - While our extension to long-form code generation tasks and reasoning models offers preliminary evidence of the generalizability of the lexical task representations, an exhaustive characterization of this generality remains a promising avenue for future research.
>
> > Reliability of logit lens — "lexical task heads rely heavily on projecting component outputs into the vocabulary space using the logit lens technique, which is known to be unreliable and heuristic."
> - Thank you for pointing this out. We agree that the logit lens can sometimes be unreliable, however, we argue that this makes our finding that such lexical task behavior exists several across models and tasks even more surprising. More importantly, beyond simply decoding to task-descriptive tokens, we find that these lexical task heads have measurable causal effects on model behavior (see Figure 4 and Appendix G).
>
> > Pretrain vs. Instruction models  — "do lexical task heads emerge in pretraining or after instruct finetuning?"
> - Thank you for the question about the emergence of lexical task heads across training stages. We conduct additional analysis on the pretrained models across three model families: meta-llama/Llama-3.1-8B, Qwen/Qwen2.5-7B, and google/gemma-2-9b. We summarize key findings below and link the **figures [here](https://github.com/anonymous-conf-acount/icml_submission/blob/6e7663a7b8372bb5c14d91389eab7da2fd90e741/results/Compare_w_base_models.pdf)**: 1) Lexical task heads exist in base models. However, for most of the tasks, there are less lexical task heads than the instruction fine-tuned models (Fig. 1–3). Lexical task heads in base models and instruction models largely overlap (Fig 4–6). The causal effect of the lexical task heads is comparable to instruction models (Fig. 7–9). We will include these analyses in the final version of the paper and we believe readers will find them informative.
>
> > Practical implications — "it still seems fails to imply which prompts will fail or giving direction for a better prompt design"
> - While we do not give directions for better prompt design, our preliminary analysis of code generation suggests the potential of applying our findings in lexical task representations to real-world practical settings. Given that the lexical task representation is less strong in failure cases and activating lexical task heads can repair them, our work lays the foundation for future studies to build practical tools for automatic failure detection and intervention. We will incorporate these points in the Discussion section.

---

> > ### Author Rebuttal · Reviewer_2g3w · 2026-04-04
> >
> > I appreciate the response and additional experimental results done by the authors, which mostly addressed my concerns, so I decided to raise my score to 5 accept as a support.

---

### Official Review · Reviewer_qhuG · 2026-03-17

**Soundness:** 4
**Presentation:** 3
**Significance:** 3
**Originality:** 3
**Overall Recommendation:** 5
**Confidence:** 4

**Summary:**

The authors present a study of both the variation of performance given variation in prompts (in LLMs) as well as the study of task heads (and the study of those in conjunction).

Specifically, the authors identify the concept of "lexical task representation", which is a way of grounding the representation of a task in vocabulary (as opposed to a vector space). Thereby the authors are able to identify the lexical task heads that are activated for a given task.

The study presents a very detailed and in-depth analysis of findings of the inner workings of different prompting techniques and offer an explanation for the it's variation and how to influence it.

**Compliance With Llm Reviewing Policy:**

Affirmed.

**Final Justification:**

Please see rebuttal acknowledgement.

**Key Questions For Authors:**

1) How would lexical task vectors be extended to more complicated scenarios.

2) How can composite tasks be studied? (e.g. where perhaps more than one atomic task head needs to be activated)

**Limitations:**

Yes.

**Strengths And Weaknesses:**

### Strenghts:

A) Very well written and clear study

B) Interesting insights and very logically presented findings (with good evidence)

C) Novel and interesting idea of studying inner workings of LLMs.

### Weakness:

A) Perhaps the lenght is more suitable for a journal paper

B) The concept of lexical task vector is somewhat vague and model dependent... and it's hard to tell how this would generalise to more complicated tasks (e.g. `agentic' tasks).

---

> ### Author Rebuttal · Authors · 2026-03-31
>
> Thank you for your positive review and helpful questions. We provide a detailed response below.
>
> > Perhaps the length is more suitable for a journal paper
> - Even though our analysis and results are rich and comprehensive, our paper is within the page limit and structured as a standard conference paper.
>
> > How can composite tasks be studied? (where more than one atomic task head needs to be activated)
> - As a proof of concept, we test the lexical task representation mechanisms on three tasks: Landmark-Country-Capital, Park-Country-Capital, and Product-Company-Ceo, which are two-hop compositional tasks studied in prior interpretability work (Khandelwal & Pavlick, 2025). We replicate key findings in our paper and link the **figures [here](https://github.com/anonymous-conf-acount/icml_submission/blob/24e7b023f4a26d67dc28acdb62960826fc0aa2c7/results/Compositional_tasks.pdf)**: 1) Behavioral variability (linked Fig. 1),  2) Identification of lexical task heads for each of the two composing tasks. For example, for the Park-Country-Capital task, we separately detect _country_ and _capital city_ heads (linked Fig. 2), 3) Causal effect: activating these heads recovers 12%--50% of the performance (linked Fig. 3).
>
> > ... it's hard to tell how this would generalise to more complicated tasks (e.g. `agentic' tasks)... How would lexical task vectors be extended to more complicated scenarios?
> - As a proof of concept, we use Python and JavaScript code generation tasks. We describe the main results below and linked the **figures / tables [here](https://github.com/anonymous-conf-acount/icml_submission/blob/e0c328e6b3329bb47364d7bb1d2e7aaaedc28e6e/results/code_generation/Code_generation.pdf)**. We detect and quantify _Python_ and _JavaScript_ lexical task heads (linked Fig. 1 and Table 1) and subsequently use them to steer the model to generate code in a specific programming language. For example, when ``Python’’ heads are activated, the model switches from generating JavaScript to Python code. This steering effect depends on the activation strength (scaling factor) applied (see linked pp. 2–3, Table 2–3). Our results across 164 coding problems from the HumanEval-X dataset (Zheng et al., 2024) show that lexical task heads remain functional even in sophisticated, long-form generation contexts.
> - While our extension to compositional and long-form generation offers preliminary evidence of the generalizability of the lexical task representations, an exhaustive characterization of this generality remains a promising avenue for future research.

---

> > ### Author Rebuttal · Reviewer_qhuG · 2026-04-06
> >
> > Thank you for responding to our review.
> >
> > Again we encourage the authors to submit a longer journal paper given the amount of results and work.
> >
> > We hope the meta-reviewers will give a high score to this paper, as it represents a solid contribution.

---

### Official Review · Reviewer_5iv6 · 2026-03-20

**Soundness:** 2
**Presentation:** 1
**Significance:** 2
**Originality:** 3
**Overall Recommendation:** 3
**Confidence:** 2

**Summary:**

This paper compares how the Instruction-based vs Example-based prompts affect the responses and performance of one LLM (i.e., Llama-3.1-8B). They found that these two different prompts share similar mechnism, and difference was casued by the activation strength.

**Compliance With Llm Reviewing Policy:**

Affirmed.

**Final Justification:**

The authors have addressed my concerns over model evalution but not presentation.
Thus, I incresed my score from 2 to 3 only for model evalution.

**Key Questions For Authors:**

Why you need 51 Figures? Which are the key Figures? Do you really intend or expect reviewers or readers to read your work carefully?

**Limitations:**

Yes

**Strengths And Weaknesses:**

The paper has 44 pages and 51 Figures. All the Figures are well-presentated and there is almost no typo in the main text. The Introduction part, especially the Figure 1, is well-illutrated and easy to understand.

After spending several hours to read this paper, as someone who not working directly in this area, I got lost in this paper. I think the authors need to simply their work and make their findings easier to understand. For example, after mentioning Figure 1, the authors mentioned Fig. 46 in L94, Page 2. I thougt it is a typo at first, then I clicked the link and found that they indeed indicate the 46th Figure. Before mentioning Figure 3, the authors talked about Fig. 16 and Fig. 45 (L231-214, Page 4). You simply can not present your paper like this. The Supplementary Figure should also be presented in order and directly relevant. For a real human reviewer, I found it is painful to jump randomly among your 51 Figures. In addtion, the authors only test a toy LLM as a testbed to analyze the mechanism. I think they need to scale up to larger open-source model.

Some typos:
L105: Davidson et al. later found. Which year?
L360: Our analysis XXX that some failures.

---

> ### Author Rebuttal · Authors · 2026-03-31
>
> Thank you, but this review does not accurately reflect the work. Specifically:
> - The review says our work focuses on one toy LM. This is not true, we analyze six models across three model families.
> - The review says our contribution is to compare how example-based vs. instruction-based prompts affect performance. This is also not accurate, our contribution primarily centers around the discovery of a specific mechanism – lexical task heads – which we characterize in detail.
> - The review focuses on the length and number of figures in the paper. The main content of the paper is the standard length and is self contained. The many additional figures are provided in the appendix, they are not necessary for assessing the contribution, and they serve to demonstrate that our findings generalize across tasks and models.
>
> We provide a more detailed response to the reviewer below.
>
> > ...of one LLM (i.e., Llama-3.1-8B)... the authors only test a toy LLM as a testbed to analyze the mechanism. I think they need to scale up to larger open-source model.
> - We employ six models across three model families: Llama-3.1 models (8B and 70B), Gemma-2 models 9B and 27B), and Qwen2.5 models (7B and 32B), as summarized in Appendix A.1. For comparison, closely related studies utilize models of comparable or smaller scales: Todd et al. used Llama-2 models of 7B, 13B and 70B parameters, and Davidson et al. used Llama-3.2 models (1B and 3B) and Llama-2 (7B). Thus, our model selection represents the upper bound of current standards in the field of mechanistic interpretability.
> - In response to other reviewers, we have also replicated our results on more recent Qwen3 models, which are Mixture of Expert (MoE) models and one of them is a reasoning model: Qwen3-30B-A3B-Instruct-2507 and Qwen3-30B-A3B-Thinking-2507. We highlight three key results and link the **figures [here](https://github.com/anonymous-conf-acount/icml_submission/blob/6e7663a7b8372bb5c14d91389eab7da2fd90e741/results/Qwen3_models.pdf)**: 1) We characterize the behavioral variability of the models (linked Figs. 1 & 2).  2) We locate the lexical task heads (see linked Figs. 3 & 4 for the quantification of these heads). 3) We confirm the causal effect of these heads. Activating these heads recover at least 10% and as much as 90% of the performance across tasks (linked Figs. 5 & 6).
>
> > Why you need 51 Figures? Which are the key Figures?
> - The main paper contains 9 figures. The other figures in the appendix exist specifically to demonstrate replication across all 6 models and 17 tasks. We view the comprehensive appendix as a strength that speaks to the rigor and thoroughness of our work, not a limitation.
>
> > Do you really intend or expect reviewers or readers to read your work carefully?
> - While our analysis is comprehensive, the manuscript adheres strictly to standard structural and length requirements. The main claims are fully supported within the primary text; the appendix serves as a supplemental resource for readers seeking extended methodological depth or seeking to replicate the results across tasks and models.
>
> > …after mentioning Figure 1, the authors mentioned Fig. 46 in L94, Page 2.
> - The paragraph in question begins by introducing function vectors as a key neighboring work. Figure 46 is referenced there to highlight that lexical task heads and function vector heads are disjoint. The figure is in the appendix because it is supplementary to the main narrative, and the main conclusion on lexical task heads remains fully self-contained without it.
>
> > Before mentioning Figure 3, the authors talked about Fig. 16 and Fig. 45 (L231-214, Page 4).
> - Fig. 16 shows the number of lexical task heads identified for each of the two prompting styles (instruction-based and example-based) across all 6 models. Fig. 45 shows the per-task distribution of these attention heads across layers, for all 17 tasks. Both figures provide supplementary context for Section 2.3.1, which is about the overlap of these attention heads across prompting styles.
>
> > Some typos: L105: Davidson et al. later found. Which year? L360: Our analysis XXX that some failures.
> - Thank you for catching these. We will correct both in the final version.
>
> We're open to any additional suggestions. If you are more confident in our work and think we have addressed your concerns, would you consider raising the score for our paper?

---

> > ### Author Rebuttal · Reviewer_5iv6 · 2026-04-02
> >
> > Thank you for the clarifications. I concede my mistake regarding the model evaluation—the inclusion of six models across three model families resolves my concern about the scale of the experiments. Accordingly, I have raised my score to a "Weak Reject."
> >
> > That said, I still find the presentation of this paper inadequate for a conference format. Another reviewer (qhuG) explicitly noted that the length and style are more suited for a journal. Rather than taking this shared critique as an opportunity to improve the manuscript's flow, the authors' response to both of us essentially boiled down to: "we are within the page limit."
> >
> > The core issue is not the page count, but the fragmented reading experience caused by heavy reliance on 40+ appendix figures to support early claims. Since the authors seem unwilling to make an effort to address two separate reviewers' concerns regarding the presentation, my score will remain a Weak Reject.

---

> > > ### Author Response · Authors · 2026-04-08
> > >
> > > Thank you for the feedback. While our paper is self-contained (the supplemental figures are not required to assess the main contributions or claims), we recognize that the presentation could be improved. We will streamline the supplemental figure references so that we point to whole appendices at the end of a paragraph rather than individual figures in the appendix. This will be more aligned with the standard practice and should alleviate this and other reviewers’ concerns.

---

### Official Review · Reviewer_px9g · 2026-03-20

**Soundness:** 2
**Presentation:** 2
**Significance:** 2
**Originality:** 3
**Overall Recommendation:** 4
**Confidence:** 3

**Summary:**

In this paper, the authors analyze the prompt sensitivity of LLMs. They first discover these heads by projecting task-specific semantics into the vocabulary space with explicit natural language words. Then, by comparing instruction-based and example-based (few-shot) prompting, the authors discover that both styles leverage a shared internal mechanism: Lexical Task Heads. Next, they found that the failure across different prompts is primarily driven by the activation strength of these heads. Crucially, the authors employ interventions such as activation patching/boosting to fix failing model outputs, improving the performance, and linking these internal representations to the model's behavior.

**Compliance With Llm Reviewing Policy:**

Affirmed.

**Final Justification:**

See acknowledgement

**Key Questions For Authors:**

See above.

**Limitations:**

yes

**Strengths And Weaknesses:**

Strength:
1. The problem of prompt sensitivity is a very important and annoying issue of LLMs.
2. The idea of the task head is novel and intuitive
3. The analysis is diverse and demonstrates the results effectively.

Weaknesses:

1. Task domain scope is limited. The tasks evaluated appear to be primarily those with clear, word-level semantic labels (e.g., classification, simple transformations). It remains unclear whether this "lexical projection" mechanism holds for more abstract or complex reasoning tasks (e.g., mathematical logic or multi-step planning) where the task cannot be easily mapped to a single "task word."

2. The paper is not very clear. First, many technical details are unclear, such as the method for identifying the task head and the motivation that this head exists. Similarly, retreival head is not very clear as well. Second, the target of the paper is to analyze why prompts lead to different performance; it is not very motivated to show that few-shot and instruction share the same task head because we do not know how this sharing is important for the research problem. In other words, even if we do not know this, we can still conclude that both prompts rely on the strength of the activation in the task head. Third, the introduction is vague and fails to address the main research gap/methods to solve the problem. Overall, the writing can be improved to make it easier to read.

3. The conclusion is overstated. Errors of LLMs can vary, and the understanding of the task itself is just one type of basic error. It is overclaiming to state that the performance difference of the samples is caused by task heads, since it only covers the task understanding types.  Besides, the finding is to find the relationship between the lexical task head and performance, but not the direct proof that the lexical task head leads to lower performance.

4. Limited usage of the findings. Although enhancing the activation head can increase the performance, it has not been widely tested to demonstrate the usefulness of the findings. Maybe a discussion session on how this finding is important can give more insights and make it critical.

---

> ### Author Rebuttal · Authors · 2026-03-31
>
> Thank you for the feedback. We provide a detailed response below.
>
> > Task domain scope is limited…whether this 'lexical projection' mechanism holds for more abstract or complex reasoning tasks
> - Our choice of tasks is consistent with recent interpretability work (Hendel et al., Todd et al., Davidson et al.). We use these simple tasks to isolate specific mechanisms in a complex system, which is a necessary first step before evaluating generalization to more tasks.
> - That said, we like the idea of multi-step tasks. As a proof of concept, we test the lexical task representation mechanisms on three tasks: Landmark-Country-Capital, Park-Country-Capital, and Product-Company-Ceo, which are two-hop compositional tasks studied in prior interpretability work (Khandelwal & Pavlick, 2025). We replicate key findings in our paper and link the **figures [here](https://github.com/anonymous-conf-acount/icml_submission/blob/24e7b023f4a26d67dc28acdb62960826fc0aa2c7/results/Compositional_tasks.pdf)**: 1) Behavioral variability (linked Fig. 1),  2) Identification of lexical task heads for each of the two composing tasks. For example, for the Park-Country-Capital task, we separately detect _country_ and _capital city_ heads (linked Fig. 2), 3) Causal effect: activating these heads recovers 12%--50% of the performance (linked Fig. 3).
> - We also expand the task scope to long-form code generation tasks (see response to [reviewer 2g3w](https://openreview.net/forum?id=1kmTTgPJAm&noteId=apVgMx9K9S)).
>
> > …it is not very motivated to show that few-shot and instruction share the same task head
> - We demonstrate that lexical task heads are shared by the two prompting styles because, critically, it distinguishes between two mechanisms: (1) models utilize different circuits to represent a task in different prompting styles, or (2) models engage the same set of attention heads to represent a task regardless of prompting styles. Demonstrating the sharing of attention heads provides evidence for the `circuit reuse’ in LLMs, which is an actively studied area in interpretability (see Related Work, pp. 8, ll. 426–439). Importantly, while previous literature studies circuit reuse across tasks, our work shows that circuits can also be reused across different prompting styles for the same task.
>
> > …the finding is to find the relationship between the lexical task head and performance, but not the direct proof that the lexical task head leads to lower performance
> - The paper also provides causal evidence. Specifically, we demonstrate that activating identified lexical task heads successfully steers the model to generate correct answers for prompts that previously elicited incorrect ones (pp.6,  Fig. 7).
>
> > …retrieval head is not very clear
> - Since retrieval heads have been extensively studied in prior work <cite>, we deliberately focus our main analysis on novel findings of the lexical task heads, delegating the specifics of retrieval heads to the Appendix. There, we detail our identification criteria (appendix J.1), provide comprehensive quantification (appendix J.2), and analyze their layer-wise distribution (appendix J.3). Furthermore, we show that retrieval heads are reused across prompting styles (appendix J.4).
>
> > Errors of LLMs can vary, and the understanding of the task itself is just one type of basic error
> - We acknowledge that task representation is one of several factors contributing to prompt sensitivity. While we do not suggest that lexical task heads are the sole driver of all performance, our study targets on how lexical task heads can explain certain failure modes. To ensure this distinction is clear, we explicitly state that ``prompt sensitivity is a complex phenomenon, likely driven by many mechanisms beyond those explored in this paper'' ( pp.8, ll. 399–401).  We also encourage future research to study other processes, such as `task detection’ (see pp. 8, ll. 393–400). We think this framing positions our findings as a key piece of a larger puzzle rather than an exhaustive explanation.
>
> > Limited usage of the findings…Maybe a discussion session on how this finding is important
> - As suggested, we will include the following points in the Discussion section to:
>   - Summarize the significance: While previous literature primarily focused on prompts that lead to the correct answer, our work investigates failure cases and prompt sensitivity.
>   - Clarify the scope: This work focuses on how one aspect of task processing – lexical task representations –  can explain prompt sensitivity. We thoroughly investigate 17 simple tasks, and provide a proof of concept of the generalizability of our findings by analyzing compositional and long-form generation tasks.
>   - Provide insights for future work: Future studies can build on our findings to investigate A) task processes beyond task representation,  B) broader generalizability of complex tasks in real-world settings, C) practical tools to detect and repair under-performing prompts.

---

> > ### Author Rebuttal · Reviewer_px9g · 2026-04-04
> >
> > Thanks for the response. I will increase my score to 4. I encourage the authors to incorporate the discussion in the paper and add more experiments to extend the scope.

---

### Official Review · Reviewer_HeaT · 2026-03-24

**Soundness:** 3
**Presentation:** 3
**Significance:** 2
**Originality:** 2
**Overall Recommendation:** 4
**Confidence:** 4

**Summary:**

This paper investigates the mechanistic underpinnings of "prompt sensitivity" in LLMs. The authors aim to determine whether different prompting styles—specifically instruction-based vs. example-based (few-shot) prompts—invoke distinct or shared internal circuits. Through a combination of Logit Lens analysis and causal interventions (Activation/Path Patching), the paper identifies a specific class of attention heads termed Lexical Task Heads (LTHs). These heads represent tasks by projecting task-descriptive semantic information (e.g., "opposite," "cities") directly into the vocabulary space.

The study demonstrates that LTHs are largely shared across different prompting styles and that behavioral variance is primarily a function of LTH activation strength. Furthermore, the authors claim to explore a central domain of LLM interpretability by showing that model failures are often the result of either insufficient LTH activation or interference from competing "off-target" task representations. The findings are validated across several model families (Llama-3.1, Gemma-2, Qwen-2.5).

**Compliance With Llm Reviewing Policy:**

Affirmed.

**Final Justification:**

Given the efforts the authors made and the contributions of this paper, I would recommend a weak accept.

**Key Questions For Authors:**

**Key Questions to Authors**
- **Generalization to Complex Tasks**: How do LTHs behave in tasks requiring complex logic or long-form generation? Is there evidence of "distributed LTHs" for tasks that are not easily named with a single word (e.g., "summarization" vs. "creative writing in the style of X")? How would the mechanism interact with Chain-of-Thought reasoning?
- **Generalization to SOTA LLMs**: Could you elaborate on the applicability of your findings to the absolute SOTA LLMs or reasoning-specialized models (e.g., Qwen3 or o1-like models)? Do these models rely more heavily on LTHs, or do they employ alternative circuits for task representation?
- **Interaction with Function Vectors**: Your work contrasts with Davidson et al. (2025), who found that function vectors for instruction prompts use different heads. Your lexical task heads are shared. Could you provide more insights into the functional relationship between these two types of representations?
- **Practical Applications**: Based on your findings regarding LTH activation and failure modes, could you provide concrete examples or potential algorithms for how this understanding could be used to automatically detect and repair under-performing prompts in a real-world setting? This would significantly enhance the paper's impact.

**Limitations:**

See weakness.

**Strengths And Weaknesses:**

**Strengths**

- **Important and Timely Problem**: The paper addresses the fundamental and practical challenge of prompt sensitivity, moving beyond empirical observation to provide a mechanistic explanation. This is highly relevant for building reliable and predictable AI systems.
- **Rigorous Methodology**: The use of causal intervention methods (activation patching, path patching) provides strong evidence for the functional role of the identified components, moving beyond mere correlation.
- **Comprehensive Evaluation**: The experiments span multiple model families (Llama, Gemma, Qwen), sizes (7B to 70B), and a diverse set of tasks, strengthening the generalizability of the findings.

**Weaknesses**
- **Limited Task Complexity**: The evaluation is restricted to "simple tasks" (e.g., attribute extraction, antonyms). It remains unclear if LTHs exist or function similarly for complex, multi-step reasoning (Chain-of-Thought) or creative tasks where the "task" cannot be easily mapped to a single semantic cluster.
- **Limited Analysis in SOTA Models**: While the experiments utilize strong recent models (Llama-3.1, Qwen-2.5, Gemma-2), the field is evolving rapidly. The paper lacks evaluation on the absolute latest architectures or "reasoning-specialized" models (e.g., models with explicit reasoning/thinking capabilities) where task representation might be more distributed or procedural. It is worth exploring whether the correlation between LTH activation and performance strengthens in more capable models, or if newer architectures (e.g., Qwen3) exhibit different mechanisms.
- **Heuristic Dependency**: The definition of LTH relies on a predefined list of task-descriptive terms (Appendix A.2.1). While sensitivity analyses are provided, the choice of these terms is inherently subjective and could influence which heads are identified, potentially introducing bias into the results.
- **Lack of Practical Applications**: While the paper explains why prompts fail, it stops short of providing a practical algorithm for automatically optimizing prompts or dynamically intervening in the residual stream to improve reliability in production environments. Bridging the gap between mechanistic understanding and practical application would significantly enhance the paper's impact.

---

> ### Author Rebuttal · Authors · 2026-03-31
>
> Thank you for the feedback. We provide a detailed response below.
>
> > How do LTHs behave in tasks requiring complex logic or long-form generation?
> - Our choice of tasks is consistent with recent interpretability work (Hendel et al., Todd et al., Davidson et al.). We leverage these simple tasks to isolate specific mechanisms in a complex system, which is a necessary first step before evaluating generalization to more scenarios.
> - That said, we like the suggestion on long-form generation tasks. As a proof of concept, we use Python and JavaScript code generation tasks. We describe the results below and provide the **figures \ tables [here](https://github.com/anonymous-conf-acount/icml_submission/blob/e0c328e6b3329bb47364d7bb1d2e7aaaedc28e6e/results/code_generation/Code_generation.pdf)**: We detect and quantify _Python_ and _JavaScript_ lexical task heads (linked Fig. 1 and Table 1) and subsequently use them to steer the model to generate code in a specific programming language. For example, when ``Python’’ heads are activated, the model switches from generating JavaScript to Python code. This steering effect depends on the activation strength (scaling factor) applied (see linked pp. 2–3, Table 2–3). Our results across 164 coding problems from the HumanEval-X dataset (Zheng et al., 2024) show that lexical task heads are functional even in complex, long-form generation contexts.
>
> > …if newer architectures (e.g., Qwen3) exhibit different mechanisms.
> - We replicate our results on Qwen3 models, including a reasoning model: Qwen3-30B-A3B-Instruct-2507 and Qwen3-30B-A3B-Thinking-2507. We highlight three key results and link the **figures [here](https://github.com/anonymous-conf-acount/icml_submission/blob/6e7663a7b8372bb5c14d91389eab7da2fd90e741/results/Qwen3_models.pdf)**: 1) Behavioral variability (linked Figs. 1 & 2),  2) Identification of lexical task heads (linked Figs. 3 & 4), 3) Causal effect: activating these heads recover 10%–90% of the performance across tasks (linked Figs. 5 & 6).
>
> >  How would the mechanism interact with Chain-of-Thought reasoning?
> - Thank you for this suggestion; exploring the interplay between lexical task representations and Chain-of-Thought (CoT) mechanisms could be a compelling direction. For instance, studying whether the internal lexical task representations shift dynamically during reasoning—and whether these shifts align with the model’s CoT token outputs —could provide a window into the underlying computations. These computations may not be faithfully represented in the external CoT tokens, which is an active area of research (Arcuschin et al., 2025, Paul et al., 2024).
> - While a definitive study of these dynamics is beyond the current scope of the paper, our results show that lexical task mechanisms are involved in reasoning models, laying the groundwork for future work to study models’ internal processing in reasoning models.
>
> > The definition of LTH relies on a predefined list of task-descriptive terms.
> - We acknowledge that the predefined task-descriptive terms could be a source of bias in the Limitation section (see pp.8, ll. 411–416). As an attempt to address this, we leverage several frontier LLMs (GPT-5.4 Mini, Claude 4.6 Sonnet, and Grok 4.2 Beta) to generate task-descriptive terms. We find that the lexical task heads identified with those terms are comparable to the originally identified ones (see **figures [here](https://github.com/anonymous-conf-acount/icml_submission/blob/6e7663a7b8372bb5c14d91389eab7da2fd90e741/results/Compare_task_descriptive_terms.pdf)** for comparison). Thus, we conclude that our predefined task-descriptive terms encompass relatively comprehensive task descriptions and are sufficient to locate lexical task heads that have causal effect (pp. 5, Fig. 4) on model performance.
>
> > Your work contrasts with Davidson et al. (2025)…Could you provide more insights into the functional relationship between these two types of representations?
> - We agree that our work is closely related to but critically distinct from Davidson et al. (2025). We compared these works in the Introduction (see pp. 2, ll. 98–108 of the left column, and ll. 80–98 of the right column). We highlight that lexical task representations differ from function vectors in that they generalize across prompting styles, revealing a shared encoding that captures the core meaning of a task, rather than being specific to surface-level structure of a given prompting style.
>
> > …how this understanding could be used to automatically detect and repair under-performing prompts in a real-world setting?
> - Our preliminary analysis of code generation provides a concrete demonstration that lexical task representations extend to real-world domains. Given our findings that the lexical task representation is less strong in failure cases and activating lexical task heads can repair them, our work lays the foundation for future studies to build practical tools for failure detection and intervention.

---

> > ### Author Rebuttal · Reviewer_HeaT · 2026-04-04
> >
> > Thank you for your detailed response and the additional experiments conducted during the rebuttal period. I appreciate the effort to address the concerns regarding task complexity and model generalization. While the new results on Qwen3 and code generation are encouraging, several critical concerns remain unresolved, preventing me from raising my score at this stage.
> >
> > 1. Presentation and Narrative Logic
> >
> > As noted by other reviewers, the main text relies heavily on pointing the reader to figures and results in the Appendix to support core arguments. While I understand this was not a page-limit violation, it severely disrupts the narrative flow and readability. The main paper should be self-contained in presenting the complete logical arc of your findings. The authors' indication that they do not plan to restructure the main text is concerning, as it suggests a lack of focus on refining the conceptual clarity of the contribution. A tighter integration of key results into the main text is necessary to improve the paper's accessibility and impact.
> >
> > 2. Limited Practical Motivation (Circular Dependency)
> >
> > Regarding the practical applications, the proposed "repair" mechanism via activation patching still suffers from a fundamental circular dependency. To repair a failing prompt, one needs access to the "correct" activations (e.g., from a successful prompt or ground truth). In a real-world scenario, if a user already possesses the correct prompt or answer, the model's failure is moot. The paper currently lacks a practical pathway for diagnosis or intervention that does not rely on prior knowledge of the correct output. I would be interested to see if the authors can propose a practical application, such as a failure prediction metric based on LTH activation patterns that does not require ground-truth patching.
> >
> > 3. Generalization vs. "Cherry-picking"
> >
> > The code generation results (Python vs. JavaScript) show a performance recovery range of 10%–90%. This wide variance raises a concern about "cherry-picking" or the limited scope of the mechanism.
> > - Task Nature: Python vs. JavaScript is essentially a "vocabulary/style switching" task, which aligns perfectly with the definition of LTHs (outputting task-descriptive tokens). However, does this mechanism hold for tasks requiring deep logical reasoning (e.g., "solve a math problem" vs. "summarize a math problem")?
> > - Variance Analysis: The 10% lower bound suggests that LTH intervention is ineffective in certain scenarios. Does this imply that LTHs are only functional for tasks with distinct lexical signatures, and fail for tasks where the "task description" is abstract or procedural?
> >
> > I am keeping my score unchanged for now, but I remain open to revising it if the authors can solve my concerns.

---

> > > ### Author Response · Authors · 2026-04-08
> > >
> > > Thank you for the feedback. We attempt to address your concerns in detail below.
> > >
> > > > Presentation and Narrative Logic… A tighter integration of key results into the main text is necessary to improve the paper's accessibility and impact.
> > >
> > > - While our paper is self-contained (the supplemental figures are not required to assess the main contributions), we recognize that the presentation could be improved. We will streamline the supplemental figure references so that we point to whole appendices at the end of a paragraph rather than individual figures in the appendix. This will be more aligned with the standard practice and should alleviate this and other reviewers’ concerns.
> > >
> > > >  Generalization vs. "Cherry-picking". The code generation results (Python vs. JavaScript) show a performance recovery range of 10%--90%.
> > >
> > > - We would like to clarify that the code generation task is not just a "style switching" task. The completion of this task has two criteria: 1) The model needs to solve a coding problem by generating code. 2) The model needs to output code in a particular programming language. Critically, this task is distinct from merely translating existing code to another programming language. Thus, our task actually requires deep logical reasoning similar to "solving a math problem" as the reviewer mentioned.
> > > - We would like to point out that we did not report a "recovery range of 10%--90%" for the code generation task. Instead, we reported that the "steering effect depends on the activation strength (scaling factor) applied", in that there exists a scaling factor that achieves the optimal causal effect.
> > >
> > > > Limited Practical Motivation (Circular Dependency)... The paper currently lacks a practical pathway for diagnosis or intervention that does not rely on prior knowledge of the correct output.
> > >
> > > - We believe there is a misunderstanding and would like to clarify that our approach does NOT have a circular dependency. 1) The identification of lexical task heads does not rely on "correct activations". Critically, we use the Logit Lens technique to identify relevant heads, which do not depend on "prior knowledge of the correct output’".  For example, to locate _Python_ heads for the code generation task, one can use simple, _neutral_ prompts like "Write Python code to solve a problem", which is disentangled from the correctness of the model’s output. 2) To repair a failing prompt, one can patch the scaled activation of identified lexical heads cached from the neutral prompts, rather than relying on "ground-truth patching".  Notably, we use patching to validate the causal effect of lexical task heads, not as a proposed intervention with practical use.
> > > - In summary, our findings are generalizable across several distinct task types and model architectures that we studied. We think our work will lay the mechanistic foundation to inspire future studies to build practical tools for failure detection and intervention.
> > >
> > > If we have addressed your previous and current concerns, we hope you consider raising the score.

---

### Decision · Program_Chairs · 2026-04-30

**Decision:**

Accept (regular)

**Comment:**

This paper received overall positive evaluations, including two Weak Accept, two Accept, and one Weak Reject recommendations. The reviewer consensus is supportive of acceptance.

The authors have addressed most of the concerns raised during the rebuttal phase. While Reviewer 5iv6 continues to express some concerns regarding the presentation and clarity of the paper, these issues appear to be addressable through revision and do not outweigh the overall strengths of the work.

I encourage the authors to further improve the presentation, clarity, and organization of the manuscript in the final revision. With these improvements, the paper will be further strengthened.

Based on the overall reviewer consensus and my assessment, the final decision is Accept.